Corrected: Author correction

# c-Met activation leads to the establishment of a TGFβ-receptor regulatory network in bladder cancer progression

Wen Jing Sim[1,20], Prasanna Vasudevan Iyengar [2,3,20], Dilraj Lama[4], Sarah Kit Leng Lui[2], Hsien Chun Ng[1], Lior Haviv-Shapira[5], Eytan Domany[5], Dennis Kappei [2,6], Tuan Zea Tan [2], Azad Saei [6,7], Patrick William Jaynes[2], Chandra Shekhar Verma[4,8,9], Alan Prem Kumar [2,10,11,12], Mathieu Rouanne[13,14], Hong Koo Ha[15], Camelia Radulescu[16], Peter ten Dijke[3], Pieter Johan Adam Eichhorn [2,10,17,18] & Jean Paul Thiery [1,6,19]

Treatment of muscle-invasive bladder cancer remains a major clinical challenge. Aberrant HGF/c-MET upregulation and activation is frequently observed in bladder cancer correlating with cancer progression and invasion. However, the mechanisms underlying HGF/c-MET-mediated invasion in bladder cancer remains unknown. As part of a negative feedback loop SMAD7 binds to SMURF2 targeting the TGFβ receptor for degradation. Under these conditions, SMAD7 acts as a SMURF2 agonist by disrupting the intramolecular interactions within SMURF2. We demonstrate that HGF stimulates TGFβ signalling through c-SRC-mediated phosphorylation of SMURF2 resulting in loss of SMAD7 binding and enhanced SMURF2 C2-HECT interaction, inhibiting SMURF2 and enhancing TGFβ receptor stabilisation. This upregulation of the TGFβ pathway by HGF leads to TGFβ-mediated EMT and invasion. In vivo we show that TGFβ receptor inhibition prevents bladder cancer invasion. Furthermore, we make a rationale for the use of combinatorial TGFβ and MEK inhibitors for treatment of high-grade non-muscle-invasive bladder cancers.

[1] Institute of Molecular and Cell Biology, A*STAR, Singapore 138672, Singapore. [2] Cancer Science Institute of Singapore, National University of Singapore, Singapore 117599, Singapore. [3] Department of Cell and Chemical Biology and Oncode Institute, Leiden University Medical Center, 2333 ZC Leiden, The Netherlands. [4] Bioinformatics Institute (A*STAR), 30 Biopolis Street, 07-01 Matrix, Singapore 138671, Singapore. [5] Department of Physics of Complex Systems, The Weizmann Institute of Science, Rehovot, Israel. [6] Department of Biochemistry, Yong Loo Lin School of Medicine, National University of Singapore, Singapore 117596, Singapore. [7] Genome Institute of Singapore, A*STAR, Singapore 138672, Singapore. [8] Department of Biological Sciences, National University of Singapore, 14 Science Drive, Singapore 117543, Singapore. [9] School of Biological Sciences, Nanyang Technological University, 60 Nanyang Drive, Singapore 637551, Singapore. [10] Department of Pharmacology, Yong Loo Lin School of Medicine, National University of Singapore, Singapore 117597, Singapore. [11] Cancer Program, Medical Science Cluster, Yong Loo Lin School of Medicine, National University of Singapore, Singapore, Singapore. [12] Curtin Medical School, Faculty of Health Sciences, Curtin University, Perth, WA, Australia. [13] Department of Urology, Hôpital Foch, Université Versailles-Saint-Quentin-en-Yvelines, Université Paris-Saclay, Suresnes, France. [14] INSERM Unit 1015, Laboratoire de Recherche Translationnelle en Immunologie (LRTI), Gustave Roussy, Université Paris-Saclay, Villejuif, France. [15] Department of Urology, Pusan National University Hospital, Pusan National University School of Medicine, Busan 602—739, Korea. [16] Department of Pathology, Hôpital Foch, Université Versailles-Saint-Quentin-en-Yvelines, Université Paris-Saclay, Suresnes, France. [17] School of Pharmacy and Biomedical Sciences, Faculty of Health Sciences, Curtin University, Bentley 6102, Australia. [18] Curtin Health Innovation Research Institute and Faculty of Health Sciences, Curtin University, Bentley, WA 6102, Australia. [19] Guangzhou Regenerative Medicine and Health, Guangdong Laboratory, 510530 Guangzhou, China. [20] These authors contributed equally: Wen Jing Sim and Prasanna Vasudevan Iyengar. Correspondence and requests for materials should be addressed to P.J.A.E. (email: Pieter.Eichhorn@curtin.edu.au) or to J.P.T. (email: bchtjp@nus.edu.sg)

Bladder cancer is a highly prevalent disease ranking eleventh most common cause of cancer-related deaths worldwide[1]. Initial tumourigenesis originates within the urothelium layer lining the inner surface of the bladder with more-progressive tumours invading the surrounding muscularis propia eventually leading to metastatic disease. Approximately 75% of bladder cancers initially present as non-muscle-invasive bladder cancer (NMIBC). Diagnosis and primary clinical management is achieved by transurethral resection of the bladder tumour (TURBT) possibly followed by intravesical immunotherapies, primarily bacillus Calmette-Guérin (BCG)[2–4]. However, superficial bladder tumours have a high recurrence rate and patients require intensive costly management. The remaining 25% of bladder cancers are muscle-invasive bladder cancers (MIBC). For those patients progressing on intravesical treatments or patients presenting MIBC the standard of care remains cystectomy with or without chemotherapy[5].

Epithelial–mesenchymal transition (EMT) is a multistep process that was first recognised as a critical event in the shaping of embryos during gastrulation and subsequently during organogenesis[6,7]. Indeed, the significant parallels in cell plasticity between embryonic development and carcinoma progression has led to the idea of EMT as a central driver of epithelial-derived tumour malignancies[8]. Furthermore, the transition from an epithelial state toward a more mesenchymal-like state has been seen as a fundamental requirement for cellular invasion and metastasis[7,9–11]. The observation that EMT occurs at the invasive front of the primary tumour has advocated the role of microenvironment signals generated by the surrounding stromal cells as inducers of EMT[12–14]. These signals consist of an array of cytokines and growth factors including hepatocyte growth factor (HGF), epidermal growth factor (EGF), transforming growth factor-β (TGFβ) and insulin-like growth factor-1 (IGF-1)[15]. HGF has been found to be significantly elevated in the serum and urine of patients with MIBC and its expression is correlated with poorer overall survival[16,17]. Furthermore, elevated expression levels of activated c-Met receptors identified in MIBC enhances the invasive and metastatic potential of bladder carcinoma cells[18–22].

The canonical TGFβ receptor (TβR) pathway has also been demonstrated to be a critical regulator of EMT, invasion and metastasis[23–26]. TGFβ ligand activation of the pathway is conducted through the type II (TβRII) and type I (TβRI) receptors. Upon activation of the receptors, intracellular signalling is initiated through the phosphorylation of receptor SMADs (SMAD2/SMAD3) permitting oligomerization with the co-SMAD, SMAD4[27]. Upon entry into the nucleus the R-SMAD/SMAD4 transcription factor complex regulates the expression of hundreds of genes, including a number of transcription factors critical for EMT[23]. To ensure external cues generate desired intercellular responses, a number of inhibitory feedback loops exist to limit unwanted hyperactivation of the pathway. For TβR signalling, SMAD7, a transcriptional target of the SMAD complexes, acts as scaffold to recruit the E3 ligase SMURF2 to the TβR complex to facilitate receptor degradation and attenuate TGFβ signalling[28–31]. Furthermore, SMAD7 can also function as an agonist for SMURF2 activity. To limit unwanted activity toward its substrates the N-terminal C2 domain of SMURF2 interacts with its C-terminal HECT domain, inhibiting ubiquitin thioester formation of its catalytic cysteine residue. The binding of SMAD7 to the WW3 domain of SMURF2 abrogates the inhibitory intramolecular interactions between the C2 and HECT domains, facilitating SMURF2 ubiquitin ligase activity[32].

The elevated levels of HGF in patients with MIBC compared with patients with superficial bladder cancer suggests that HGF has a role in inducing EMT and invasion in bladder cancer progression[18,20]. We and others have previously reported on the role of HGF in EMT induction[33,34]. However, the precise mechanism through which c-MET activation induces EMT remains undetermined. Here we elucidate the mechanism through which activated c-Met promotes downstream TβR signalling enhancing EMT and invasion in vitro, and tumour progression in bladder carcinoma orthotopic transplantation model.

## Results

**HGF/c-Met confers invasive properties to bladder carcinoma cells.** To identify the mechanisms through which HGF and its cognate receptor, c-MET, induces EMT in bladder cancer we treated the rat bladder carcinoma cell line NBT-II, with either HGF or EGF and analysed changes in cellular morphology with EMT marker expression (Fig. 1a, Supplementary Fig. 1a). HGF and EGF treatment decreased intercellular adhesion as exhibited by loss of desmosomes and E-cadherin junctional complexes while increasing vimentin expression[15,33–36]. NBT-II cell motility was assessed using time-lapse microscopy (Supplementary Fig. 1b). Treated NBT-II cells travelled further while displaying an increase in velocity compared with control cells (Supplementary Fig. 1c, d). To investigate the phenotypic changes following HGF exposure, images of NBT-II cells were taken at 5 min intervals for 48 h (Fig. 1a). Following HGF treatment, coherent sheets of epithelial NBT-II carcinoma cells progressively dissociated over time, leading to establishment of single mesenchymal-like migratory cells as early as 9 h (Fig. 1a). Furthermore, tracking of cells at early time points showed that NBT-II cells engaged into locomotion within 2–4 h following HGF induction (Fig. 1b, c). HGF-treated cells decreased E-cadherin expression, dissociated from one another and exhibited an increase in vimentin (Fig. 1d). These effects were annulled upon treatment with the c-MET inhibitor JNJ38877605 (Supplementary Fig. 1e). Collectively, these results indicate that HGF induces a rapid EMT response in NBT-II cells.

**c-Met activation drives TβR-dependent SMAD2 phosphorylation.** To understand the intracellular responses upon HGF treatment, we probed a phospho-specific antibody microarray with lysates from control or HGF-treated NBT-II cells 2 h post treatment (Fig. 2a, Supplementary Data 1). Intriguingly, HGF upregulated the phosphorylation of R-SMADs at both the linker residues (Ser204; pSMAD2L) and at the C-terminal motif (Ser467; pSMAD2C). pERK levels were also unexpectedly reduced[37]. Notably, SMAD2 C-terminal phosphorylation can only occur through TGFβR activation, revealing a potential cross-activation of the TβR receptors by HGF[38]. This was validated as both pSMAD2L and pSMAD2C correspondingly increased in a HGF dose-dependent manner (Fig. 2b, Supplementary Fig. 2a). NBT-II cells treated with HGF in a time-dependent manner demonstrated a rapid induction of pERK at 5 mins with an analogous increase phospho-SMAD2L (Fig. 2c, Supplementary Fig. 2b)[39]. Notably, ERK phosphorylation diminished at later time points, a result in line with our phosphoprotein array. Importantly, SMAD2C phosphorylation was upregulated only 1 h after HGF induction (Fig. 2c, Supplementary Fig. 2b). Also c-SRC activation occurred at 15 mins after HGF treatment and corresponded with downregulation of pERK and upregulation of pSMAD2C (Fig. 2c, Supplementary Fig. 2b). Co-treatment with the c-Met inhibitor JNJ38877605 blocked HGF activation of both the phosphorylated isoforms of SMAD2 indicating that HGF-mediated phosphorylation of SMAD2 is dependent upon active c-Met signalling (Fig. 2d). Next, we sought to determine whether HGF-induced pSMAD2C was directly dependent upon TβR activity. To this end, we performed a nuclear cytoplasmic fractionation assay to identify the localisation of pSMAD2 in the

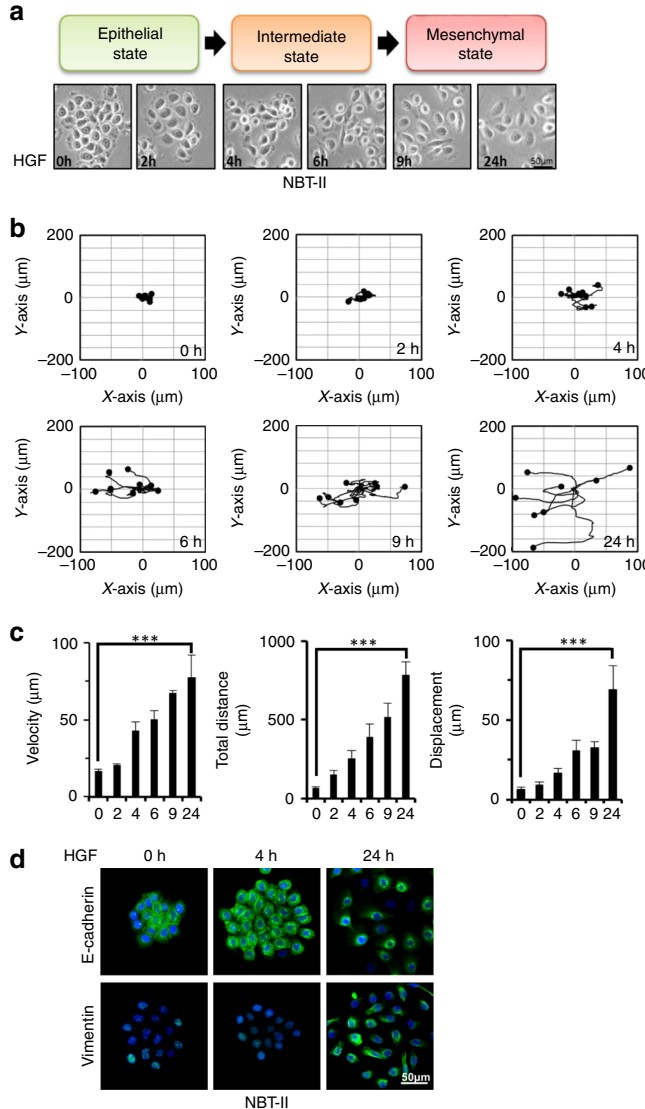

**Fig. 1** HGF-induced EMT in NBT-II carcinoma cells. **a** Phase contrast images of NBT-II cells at 0, 2, 4, 6, 9 and 24 h post HGF induction. Scale bars: 50 μm. **b** Cell tracks of NBT-II cells at 0, 2, 4, 6, 9 and 24 h post HGF induction. Data shown are representative of three independent and reproducible experiments. **c** Bar charts display velocity (left panel), total distance (middle panel) and displacement (right panel) of NBT-II cells over 24 h. Bars represent mean ± SD of three independent experiments. A two-tailed Student's $t$ test compares the treated versus untreated cell populations, ***$P < 0.001$. **d** Representative immunofluorescence images of NBT-II stained with E-cadherin and Vimentin at 0, 4 and 24 h post HGF. Scale bars: 50 μm

absence and presence of the TβR inhibitor A83-01[40]. HGF treatment enhanced nuclear pSMAD2 localisation, an effect that was partially reversed upon the co-addition of A83-01 (Fig. 2e, f). Furthermore, NBT-II cells expressing a kinase-inactive TβRI resulted in the cytoplasmic retention of SMAD2 following HGF exposure, indicating that TβR activation is required for HGF-induced pSMAD2C (Fig. 2g). Next, we sought to investigate the impact of SMAD2 on HGF-induced EMT. siRNA-mediated knockdown of SMAD2 prevented the dispersion of HGF-treated cell colonies (Supplementary Fig. 2c, d). As the Activin receptors (ACVR2B and ALK4) have previously been shown to play a role in the induction of SMAD2 we utilised soluble Activin receptor

type 2B-Fc ligand traps or pan TGFβ-neutralising antibodies to observe if ALK4 or ALK5 is involved in HGF-induced pSMAD2C. While Activin inhibition minimally decreased pSMAD2C levels, neutralising TGFβ completely abolished HGF-mediated pSMAD2C (Supplementary Fig. 2e). Similar observations were made with catalytically inactive mutants of ALK4 and ALK5 (Supplementary Fig. 2f). Co-treatment with A83-01 prevented HGF-induced EMT as observed by the reconstitution of desmosomes and the concomitant loss of the mesenchymal marker vimentin (Supplementary Fig. 3a). In addition, HGF-induced migration was attenuated upon treatment with A83-01 (Supplementary Fig. 3b, c). These results confirm the role of the canonical TβR pathway in HGF-induced EMT and motility.

Next, we analysed the RNA expression profiles of NBT-II cells following HGF treatment at 2, 4, 6, 9, 24 and 48 h. We noted an early upregulation of 229 genes 2 h post HGF treatment, the majority of which remained upregulated up to 9 h post treatment (Supplementary Fig. 4a). Interestingly, a number of described TGFβ-regulated genes were upregulated by HGF as early as 2 h, indicating that transcription of these genes may be TGFβ pathway-dependent (Supplementary Fig. 4b). Indeed, analysis of a TβR/BMPR signalling pathway profiler qRT-PCR array indicated that HGF induced the expression of 28 TβR target genes (Supplementary Fig. 4c, Supplementary Data 2). To explore the functional processes of these 229 early transcribed genes, we performed pathway enrichment analysis using Enrichr[41]. TGFβ signalling showed the two highest combined enrichment scores with six different TβR signalling gene sets, demonstrating significance (Fig. 2h, Supplementary Data 3). Furthermore, comparison of our transcriptional signature in NBT-II cells following HGF treatment indicated a significant enrichment score to either MSigdb v5.0 Hallmark TGFβ or EMT signatures (Supplementary Fig. 4d). To further confirm the role of the TGFβ pathway in our observations we analysed the effect of A83-01 on HGF-induced transcription. Co-treatment with A83-01 reversed gene expression of a subset of genes associated with TGFβ signalling as determined by Enrichr (Supplementary Fig. 4e). In particular, A83-01 diminished HGF-induced PAI-1 mRNA and protein levels (Supplementary Fig. 4f, g). Taken together, these results suggest that HGF induces an early TβR expression signature required for EMT in bladder cancer.

**HGF/c-MET driven c-SRC inhibition of SMURF2 ligase activity.** To uncover novel repressors of EGF, HGF and IGF-induced EMT we previously performed a high content screening assay where we identified compounds targeting c-SRC as an antagonist of this process[34]. Follow-up analyses demonstrates a near-complete inhibition of cell scattering when cells were treated with the c-SRC inhibitor AZD0530 and HGF compared with HGF alone (Fig. 3a). Furthermore, co-treatment with AZD0530 blocked HGF, EGF, or IGF-induced EMT as observed by the reconstitution of desmosomes and the concomitant loss of vimentin (Supplementary Fig. 5a, b). Correspondingly, co-transfection of NBT-II cells with siRNA targeting c-SRC enhanced the presence of E-cadherin at cell–cell junctions and decreased vimentin expression compared with cells treated with HGF alone (Supplementary Fig. 5c, d).

Next, we sought to analyse the intercellular responses in NBT-II cells upon treatment with the SRC inhibitors AZD0530 and PP1. Treatment with either inhibitor resulted in a similar protein expression pattern to TβR downregulation with TβR inhibitors A83-01 and LY2157299, displaying comparable decreases in pSMAD2C and upregulation of pERK (Fig. 3b). c-SRC is frequently activated in human cancer and has been shown to play a role in EMT, resulting in tumour invasion and metastasis,

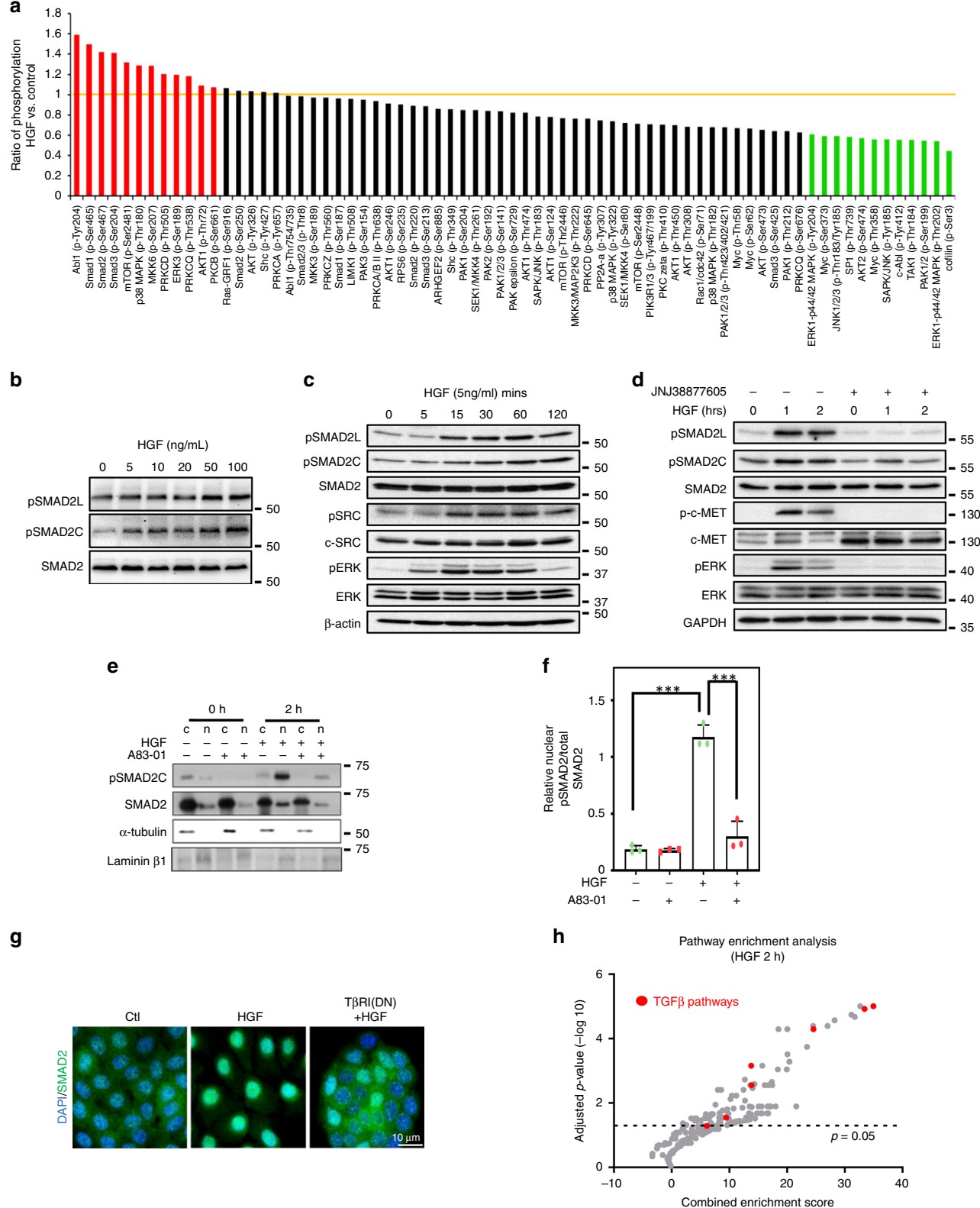

but no evidence has directly demonstrated a mechanism describing c-SRC regulation of pSMAD2[42–45].

Both SMURF2 and its orthologue NEDD4 harbour a C2 and HECT domain whereby the C2 domain binds to the HECT domain, resulting in a closed conformation of the protein, and loss of enzymatic activity[32]. Recently, it has been demonstrated that c-SRC phosphorylates a number of tyrosine residues on

NEDD4, disrupting the C2-HECT complex, leading to catalytic activation of ligase[46]. Similarly, the binding of SMAD7 to SMURF2 antagonises C2-HECT interaction permitting activation of SMURF2, leading to eventual degradation of the TβR complex and loss of TβR signalling[32]. Although, the phosphorylation of NEDD4 by c-SRC resulted in the activation of NEDD4, a mechanism counter-intuitive to our proposed observations, we

**Fig. 2** c-Met and TGFβ signalling in HGF-mediated EMT of NBT-II carcinoma cells. **a** Bar graph displays ratio of phosphorylated proteins at 2 h post HGF compared with control. **b** Western blot analysis of NBT-II cells treated with escalating doses of HGF at various concentrations for 2 h. Lysates are probed with indicated antibodies. **c** Western blot analysis of NBT-II cells treated with 5 ng/ml of HGF. Lysates were collected at denoted time points and probed with indicated antibodies. **d** Western blot analysis of NBT-II cells treated with 5 ng/ml of HGF in the absence or presence of c-MET inhibitor JNJ38877605. Lysates were collected at denoted time points and probed with indicated antibodies. **e** Western blot analysis of nuclear (n) and cytoplasmic (c) fractions isolated from NBT-II cells treated with HGF for 2 h with or without TβRI inhibitor (A83-01, 8 μM). Lysates were collected at denoted time points and probed with indicated antibodies. **f** Bar chart displays ratio of pSMAD2 band intensity in nuclear (n) fractions compared with total SMAD2 (nuclear + cytoplasmic) fractions isolated from NBT-II cells treated with HGF for 2 h with or without TβRI inhibitor (A83-01, 8 μM). Bars represent mean ± SD of three independent experiments. A two-tailed Student's $t$ test compares the treated cell populations, **$P < 0.01$, ***$P < 0.001$. **g** Immunofluorescence images of SMAD2-stained cells transfected with either control vector or TβRI-dominant negative (DN) mutant construct with or without HGF. Scale bars: 10 μm. **h** Scatter plot of 245 pathways combined enrichment score ($x$ axis) and $p$ value in $-\log 10$ scale ($y$ axis) from Enrichr. Pathways related to TGFβ are denoted in red. Dotted line indicates $p = 0.05$. The adjusted $p$ value is the Benjamini–Hochberg corrected $p$ value from hypergeometric test

investigated whether c-SRC could form a complex with SMURF2. Indeed, c-SRC bound to SMURF2 under both ectopically expressed and physiological conditions (Fig. 3c, Supplementary Fig. 5e, f). Furthermore, HGF treatment enhanced the binding of c-SRC to SMURF2 (Fig. 3d). Next, we tested if c-SRC or other SRC family members phosphorylated SMURF2. Both c-SRC and LCK enhanced tyrosine phosphorylation of SMURF2 but not LYN, FYN, or YES (Fig. 3e). Moreover, overexpression of a constitutively active mutant of c-SRC (Y530F) enhanced tyrosine phosphorylation compared with its wild-type counterpart (Supplementary Fig. 5g). In contrast, co-transfection with either kinase-dead or inactive mutants of c-SRC (K298M and Y419F) or the addition of the SRC inhibitors PP2 and Dasatinib inhibited SMURF2 tyrosine phosphorylation (Fig. 3f, Supplementary Fig. 5g).

To identify the putative phosphorylation sites on SMURF2 regulated by c-SRC, we immunoprecipitated SMURF2 from SILAC (Stable Isotope Labelling with Amino Acids in Cell Culture)-labelled HEK293T cells in the absence or presence of c-SRC followed by quantitative mass spectrometry analysis. We identified five residues, which are phosphorylated (Ser44, Thr249, Tyr314, Ser384 and Thr466) on SMURF2. However, only Tyr314 phosphorylation was identified with SILAC ratios clearly different from the ~ 1:1 ratios observed in unmodified SMURF2 protein and the other candidate residues (Fig. 3g, Supplementary Fig. 6a, b). Analysis of SMURF2 phosphorylation by c-SRC, utilising a SMURF2 mutant harbouring a phenylalanine mutation at Tyr314 significantly reduced tyrosine phosphorylation of SMURF2 (Fig. 3h). However, low levels of SMURF2 tyrosine phosphorylation were still detected, indicating other sites on SMURF2 may be phosphorylated by c-SRC. As NEDD4 is phosphorylated by c-SRC at a number of different residues (Tyr43, Tyr365, Tyr366 and Tyr585) we performed sequence alignment with NEDD4 and SMURF2 and identified Tyr434, corresponding to Tyr585 in NEDD4, as a candidate phosphorylation site for c-SRC (Fig. 3g)[46]. Generation of a double phenylalanine mutant at Tyr314 and Tyr434 (henceforth referred to as FF) completely inhibited c-SRC-mediated phosphorylation of SMURF2 (Fig. 3h). To further confirm the phosphorylation of SMURF2 by c-SRC, we generated phospho-specific antibodies targeting both loci. In Fig. 3i, j c-SRC enhanced phosphorylation of Tyr314 and Tyr434, an effect, which was abrogated in the corresponding SMURF2-phenylalanine-mutants.

SMURF2 is a negative regulator of TGFβ activity. Therefore, to determine whether these residues are critical for SMURF2-mediated function, we co-transfected a TGFβ-responsive luciferase reporter (CAGA-Luc) with either wild-type SMURF2 or the SMURF2 (FF) mutant. As expected, ectopic expression of SMURF2 and the SMURF2 (FF) mutant significantly down-regulated CAGA-Luc activity (Fig. 3k). In contrast, ectopic expression of either SMURF2 phospho-mimetic mutants

SMURF2 (Y314E) or SMURF2 (Y434E) partially inhibited the ability of SMURF2 to downregulate the reporter (Fig. 3l). However, transfection of a SMURF2 mutant with glutamic residues at both tyrosine residues (Y314E/Y434E, referred to as EE) completely abrogated SMURF2-mediated downregulation of luciferase activity (Fig. 3l). Collectively, these results indicate that the tyrosine kinase c-SRC phosphorylates SMURF2 at Tyr314 and Tyr434 blocking SMURF2-mediated downregulation of the canonical TGFβ pathway.

**Phosphorylation of Tyr314 and Tyr434 by c-SRC inhibits SMAD7 binding and enhances C2-HECT domain interaction.** As c-SRC phosphorylation of NEDD4 led to enhanced activity, we sought to understand what effect c-SRC phosphorylation has on SMURF2. Ectopic expression of c-SRC completely abolished SMURF2 auto-ubiquitination, a mark of SMURF2 ligase activity (Fig. 4a). In addition to the C2 and HECT domains, SMURF2 contains three WW domains, which recognise PPXY (PY) motifs on SMURF2 target proteins or adapters, such as SMAD7. Further investigations have shown that the PY motif of SMAD7 directly interacts with the WW3 (297–330aa) domain of SMURF2[47,48]. As c-SRC phosphorylates SMURF2 at Tyr314 in the WW3 domain, we sought to explore if phosphorylation of Tyr314 altered SMAD7 binding. The structure of WW3 domain of SMURF2 complexed with a SMAD7 peptide incorporating the PY motif (PDB ID: 2LTZ) provides a model to investigate this effect. We generated an electrostatic potential surface of WW3 domain, which, reveals that the PY motif of SMAD7 tracks along a hydrophobic region of the domain, where Tyr314 resides (Fig. 4b). We subjected this complex structure to two independent molecular dynamics simulations differing in the chemical state of Tyr314. Our simulations reveal that upon Tyr314 phosphorylation, SMAD7 peptide is no longer able to form a complex with the WW3 domain (Fig. 4b–d). The addition of the negative charge through phosphorylation at Tyr314 attracts the adjacent arginine residues R306 and R321, forming a salt-bridge interaction collectively altering the chemical and physical nature of the binding interface in the WW3 domain, resulting in the prevention of SMAD7 binding (Fig.4c, d). Indeed, co-immunoprecipitation of SMAD7 with SMURF2 was diminished in SMURF2 (Y314E) mutants, mimicking c-SRC phosphorylation (Fig. 4e). Analogously, co-expression of c-SRC decreased the binding of SMAD7 to SMURF2 and to the SMURF2 catalytically inactive mutant (SMURF2 C/A), but not with the SMURF2 isoforms Y314F and Y434F (Fig. 4f, g). These results indicate that phosphorylation of Tyr314 within the WW3 domain of SMURF2 inhibits the binding of SMAD7 to SMURF2.

As indicated, c-SRC phosphorylation of NEDD4 at Tyr585 disrupts the C2-HECT domain interaction, resulting in catalytic activation. To understand the role of the matching Tyr434

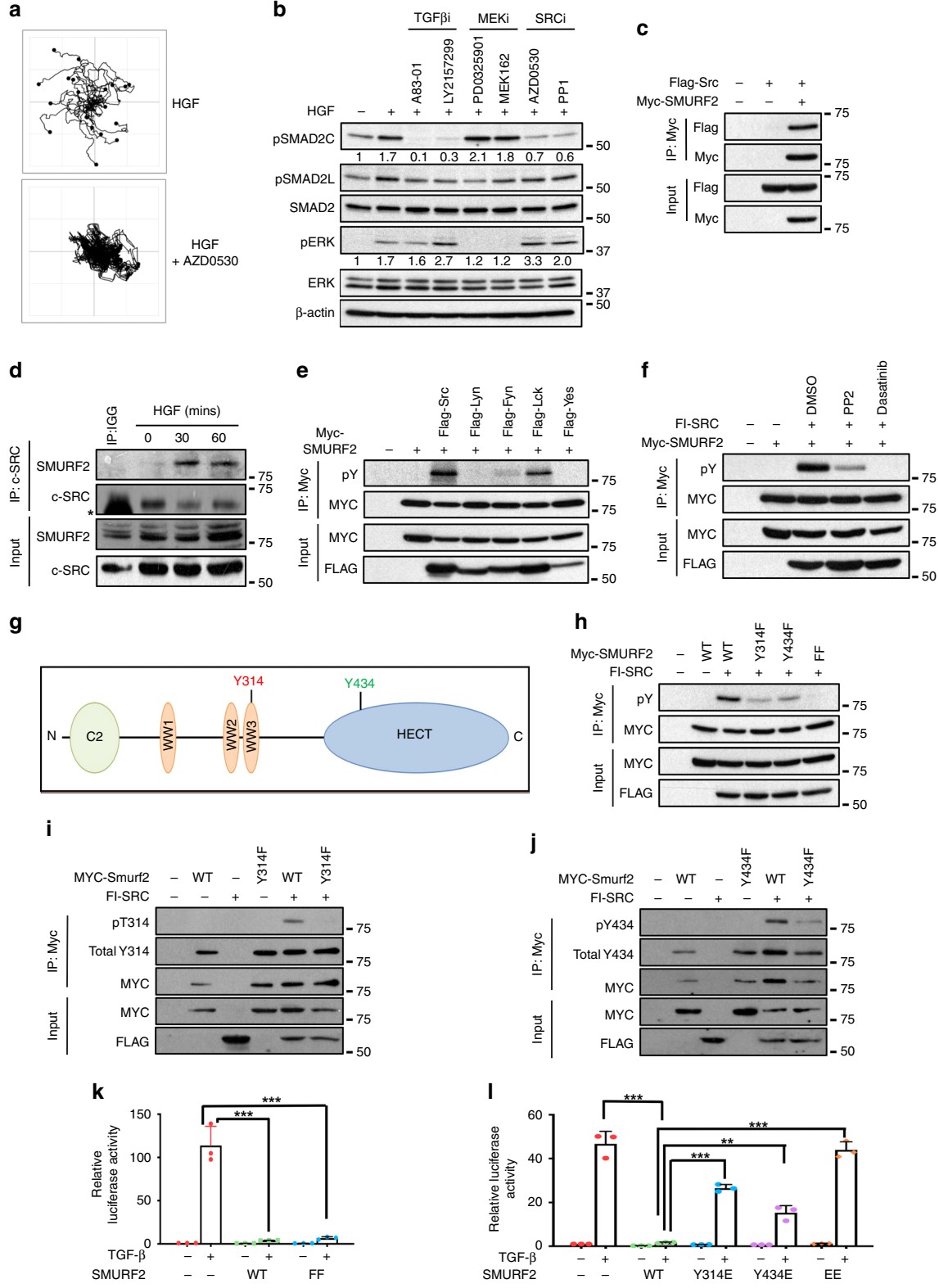

phosphorylation site in SMURF2, we modelled the complex state structure of C2-HECT domains (Fig. 5a) (See methods section for description of model generation; Supplementary Fig. 7a–e). Although, Tyr434 does not directly overlap with the C2 domain, we observed that a highly charged segment within the C2 domain formed by residues $^{82}$HKKIHKK$^{88}$ is located in proximity to Tyr434 (Fig. 5a, b). In an unphosphorylated state, Tyr434 does not form any significant interaction with this motif. However, in

contrast to NEDD4, tyrosine phosphorylation at Tyr434 enhances the binding between C2 and HECT domains. We observe that phosphorylation at Tyr434 changes the local electrostatic potential of the HECT domain region from one which is relatively positive towards one being more negative (Fig. 5b, c). This generates a favourable environment for the positively charged motif of C2 to come into closer contact with the newly formed negatively charged region, surrounding Tyr434

**Fig. 3** c-SRC phosphorylates SMURF2 at Tyr314 and Tyr434. **a** Cell tracks of HGF-treated NBT-II cells at 24 h in presence or absence of the c-SRC inhibitor AZD0530 (1 μM). **b** Western blot analysis of NBT-II cells treated with 5 ng/ml of HGF and A83-01 (8 μM), or LY2157299 (1 μM), PD0325901 (1 μM), MEK162 (1 μM), AZD0530 (1 μM), or PP1 (1 μM). Lysates were collected at 90 min post HGF treatment and probed with indicated antibodies. **c** 293 T cells were transfected as indicated. Lysates were immunoprecipitated with anti-MYC. Whole cell extracts were probed with the indicated antibodies. **d** 293 T cells treated with HGF for indicated time points. Lysates were immunoprecipitated with anti-SMURF2. Whole cell extracts were probed with indicated antibodies. * indicates IgG band. **e** 293 T cells were transfected as indicated. Lysates were immunoprecipitated with anti-MYC affinity resin. Whole cell extracts were probed with the indicated antibodies. pY signifies tyrosine phosphorylation. **f** 293 T cells were transfected as indicated. After 24 h cells were treated with c-SRC inhibitors PP2 (1 μM) or dasatinib (1 μM). Twenty-four hours later, cells were lysed and immunoprecipitated with anti-MYC. Whole-cell extracts were probed with the indicated antibodies. **g** Schematic of c-SRC tyrosine phosphorylation sites on SMURF2. Red denotes identification by mass spectrometry. Green denotes identification through sequence alignment. **h** 293 T cells were transfected as indicated. After 48 h, cells were lysed and immunoprecipitated with anti-MYC. Whole-cell extracts were probed with the indicated antibodies. **i** 293 T cells were transfected as indicated. Whole-cell extracts were probed with the indicated antibodies. **j** 293 T cells were transfected as indicated. Whole-cell extracts were probed with the indicated antibodies. **k** 293 T cells were co-transfected with a CAGA-luciferase reporter and either SMURF2 WT, or SMURF2 (FF). Cells were stimulated with TGF-β overnight and luciferase activity was measured. **l** 293 T cells were co-transfected with a CAGA-luciferase reporter, SV40-Renilla and either SMURF2 WT, or SMURF2 Y314E, Y434E, or the double mutant EE. Cells were stimulated with TGFβ overnight and luciferase activity was measured. For **k** and **l** bars represent mean ± SD of three independent experiments. A two-tailed Student's *t* test compares the treated populations, **$P < 0.01$, ***$P < 0.001$

permitting a stronger interaction between Lys87 and phosphorylated Tyr434 (Fig. 5d). The other basic residues, H86 and K88, embedded in the motif, though not as close to Y434 as compared with K87 may still form a favourable interaction as electrostatic interactions have significant long-range effects. This favourable interaction between the phosphotyrosine and lysine/histidine residues could be an important factor in the enhancement of the binding between C2 and HECT domains of SMURF2 in the phosphorylated state of Tyr434. Thus, the proposed structural model of the C2-HECT complex suggests a favourable interaction between the phosphotyrosine and lysine/histidine residues, which could be an important factor in the enhancement of the binding between C2 and HECT domains of SMURF2 when Tyr434 is phosphorylated. However, this may not be the only optimum binding geometry and there could be other similar modes of interactions between the two domains in the ensemble of structures.

Our results indicate that c-SRC phosphorylation of SMURF2 at Tyr314/Tyr434 prompts C2-HECT domain interaction, maintaining SMURF2 in a closed inactive conformation. Accordingly, auto-ubiquitination of SMURF2 was diminished by both Y314E and Y434E mutations (Fig. 5e). In contrast, SMURF2 Y434F showed a similar demonstrable increase in incorporated ubiquitin levels compared with the SMURF2 FF29/30AA, a mutant that inhibits the binding between the C2/HECT domains rendering SMURF2 catalytically active (Fig. 5f)[32]. Furthermore, mutation of Tyr314 and Tyr434 to phenylalanine completely annulled the ability of c-SRC to downregulate SMURF2 activity (Fig. 5g). Collectively our studies reveal a mechanism in which HGF-induced c-SRC phosphorylation of SMURF2 at Tyr314 and Tyr434 functions to negatively regulate E3 ligase activity by maintaining complex formation between the C2 and HECT domains.

**HGF/c-MET utilise c-SRC/SMURF2 axis as a bimodal switch to regulate TGFβ and MAPK kinase signalling.** SMURF2 can function as both a tumour suppressor or an oncogene[30]. Recently, it has been shown that SMURF2 ubiquitinates and degrades the E3 ligase β-TrCP, resulting in the stabilisation of K-RAS[49]. As ERK phosphorylation was rapidly downregulated at 2 h post HGF treatment, we hypothesised that c-SRC-mediated inhibition of SMURF2 through phosphorylation at Tyr314 and Tyr434 may act as a negative feedback loop to limit MAPK hyperactivation. In contrast, loss of HGF-mediated c-SRC phosphorylation would lead to SMURF2 activation and a decrease in TβR stability and overall downregulation of TGFβ signalling. As expected, HGF addition enhanced membrane localisation of TβRI (Fig. 6a).

Moreover, HGF-induced TβR stability was significantly decreased in cells treated with the c-SRC inhibitor AZD0530 (Fig. 6b, Supplementary Fig. 7f–h). Next, we analysed the effect of SMURF2 or the relevant SMURF2 mutants on TβRI stability. Co-transfection of wild-type SMURF2 or SMURF2(FF) decreased TβRI levels (Fig. 6c). In contrast, ectopic expression of SMURF2 (EE) completely diminished the ability of SMURF2 to degrade TβRI (Fig. 6c). In addition, ectopic expression of c-SRC stabilised TβRI in both the presence and absence of SMURF2 (Fig. 6d). The ability of SMURF2 and SMURF2(FF) to downregulate TβRI stability was blocked by the addition of the proteasome inhibitor, MG132, indicating that tyrosine phosphorylation of SMURF2 blocks SMURF2 downregulation of TβR potentially enhancing pSMAD2 signalling (Fig. 6e). Indeed, both SMURF2 and SMURF2(FF) repressed SMAD2 phosphorylation, but not the SMURF2(EE) mutant (Fig. 6f).

Next, we evaluated the consequence of SMURF2 phosphorylation on MAPK activation. As expected, expression of SMURF2 or SMURF(FF) decreased β-TrCP levels (Fig. 6g)[49]. Furthermore, and in complete contrast to TβR signalling, both SMURF2 and SMURF2(FF) enhanced MAPK activation whereas SMURF2(EE) abrogated downstream MAPK activation (Fig. 6h). Collectively, these results suggest that SMURF2 acts as a bio-modal switch between MAPK and TGFβ activation and suppression, an effect regulated by HGF-induced c-SRC activation (Fig. 6i).

**TβR suppression inhibits HGF-induced EMT and invasion.** c-Met expression level correlates with tumour grade and poor prognosis in human bladder cancer patients[18–20]. To further investigate the potential role of the HGF/c-Met axis and EMT in bladder cancer we probed the TCGA Bladder Cancer data set (BLCA $n = 408$) against our established EMT signature[50]. A high EMT score indicates a sample, portends toward a more mesenchymal-like phenotype, and vice versa. Importantly, HGF, c-Met and p-c-Met expression all correlated with the EMT signature (Fig. 7a). Furthermore, stratification of bladder cancer patients into two groups based on the HGF expression determined that patients with high HGF expression had significantly reduced overall survival (Fig. 7b).

Recently, we performed a meta-analyses of 2411 patients from 19 bladder cancer cohorts and identified six major molecular subtypes, including two predominant subtypes denoted as Mesenchymal-like (MES) and Squamous-cell carcinoma-like (SCC)[51]. Cross analysis of the subtypes with our established EMT signature indicated that both MES and SCC subtypes had a more pronounced EMT signature compared with other subtypes. Importantly, there was a significant subtype association with SCC

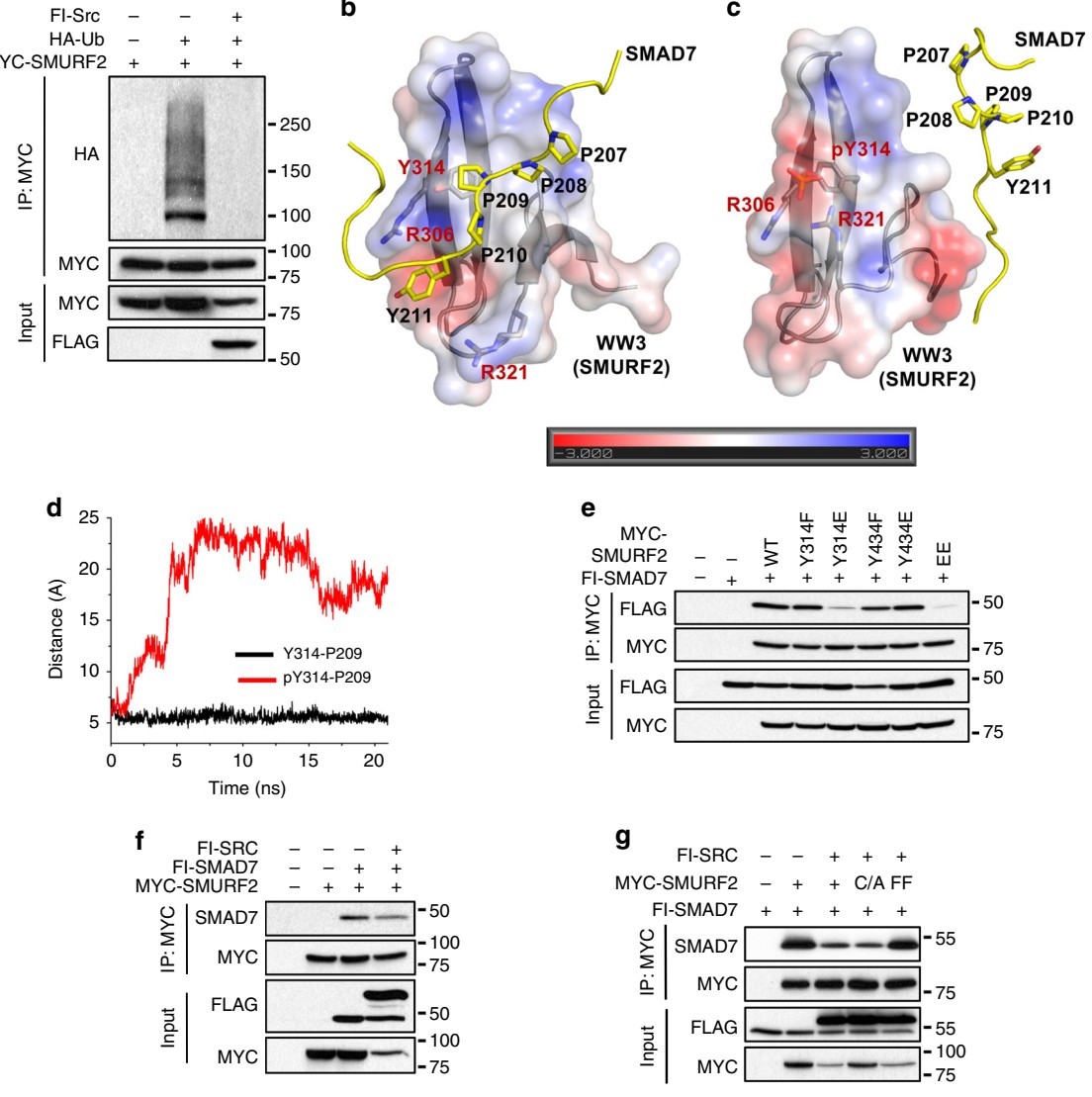

**Fig. 4** c-SRC phosphorylation of SMURF2 inhibits SMAD7 binding. **a** 293 T cells transfected with MYC-SMURF2, Flag-SRC, control vector and HA-tagged ubiquitin. Following immunoprecipitation of MYC-SMURF2 lysates were resolved by SDS–PAGE and probed with indicated antibodies. **b**, **c** Representative structure from MD simulation of WW3 domain of SMURF2 in complex with a peptide from SMAD7 in **b** unphosphorylated state of Y314 and **c** phosphorylated state of Y314. The WW3 domain is shown in electrostatic surface and ribbon representation, whereas the SMAD7 peptide is shown in yellow colour in ribbon. The PY motif of SMAD7 and Y314 along with the adjacent arginine residues of WW3 domain is shown in stick representation. **d** Time series plot (in angstrom unit) of the centre of mass distance between Y314 of WW3 domain and P209 from SMAD7 peptide in the phosphorylated (red colour) and unphosphorylated (black) states of Y314 during the MD simulation of WW3-SMAD7 complex. **e** 293 T cells were transfected as indicated with MYC-tagged SMURF2 or corresponding mutants and FLAG-tagged SMAD7. After 48 h, cells were lysed and immunoprecipitated with anti-MYC affinity resin. Whole-cell extracts were probed with the indicated antibodies. **f** 293 T cells were transfected as indicated with MYC-tagged SMURF2, Flag-tagged SMAD7 and/or Flag-tagged c-SRC. After 48 h, cells were lysed and immunoprecipitated with anti-MYC affinity resin. Whole-cell extracts were probed with the indicated antibodies. **g** 293 T cells were transfected as indicated with MYC-tagged SMURF2, MYC-tagged SMURF2(C/A) or MYC-tagged SMURF2(FF), Flag-tagged SMAD7 and/or Flag-tagged c-SRC. After 48 h, cells were lysed and immunoprecipitated with anti-MYC affinity resin. Whole-cell extracts were probed with the indicated antibodies

tumours and distant metastasis[51]. Evaluation of SMURF2, HGF and c-Met expression in the SCC subtype ($n = 272$) indicates a significant correlation with expression and EMT score (SMURF2 $R = 0.18$, $p = 0.003$; HGF $R = 0.22$, $p = 0.0003$; c-Met $R = 0.17$, $p = 0.005$).

Next, we sought to determine whether p-c-SRC correlated with pSMAD2C in a collection of bladder cancer patient-derived samples. Immunohistochemical analysis revealed that p-c-SRC expression co-expressed with pSMAD2C in all of the histo-subtypes tested including papillary, urothelial carcinoma with squamous differentiation and sarcomatoid (Supplementary

Fig. 8a). Furthermore, p-c-SRC expression correlated with nuclear pSMAD2C in 2/2 papillary urothelial carcinoma and 5/6 urothelial carcinoma with squamous differentiation bladder carcinoma and neuro-endocrine bladder carcinoma (Supplementary Fig. 8b, c). This finding confirmed our hypothesis that c-SRC activity regulates TGFβ activity in bladder cancer.

Next, we probed RNA data sets of bladder cancer cell lines ($n = 21$) and scored them against the spectral EMT signature (Fig. 7c, Supplementary Data 4)[50]. As certain bladder cancer subtypes display a mesenchymal phenotype we focused our attention on cell lines displaying the highest EMT scores to reflect

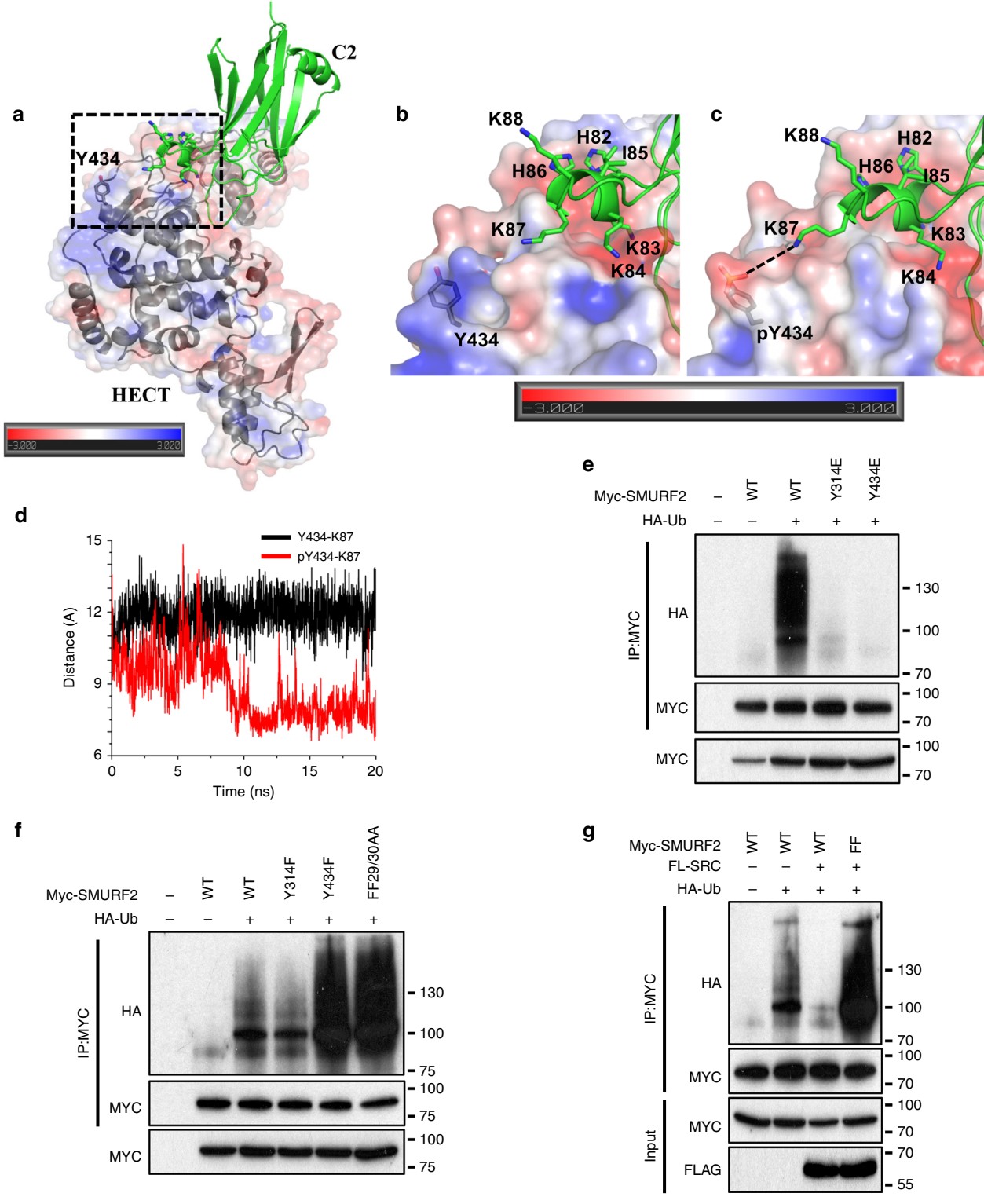

what is observed in clinical settings. Importantly, HGF treatment stimulated c-Met activation in T24, J82 and UMUC3 cell lines an effect abrogated upon the addition of the c-Met inhibitor JNJ38877605 (Supplementary Fig. 9a). Moreover, HGF treatment rapidly induced downstream SMAD2 linker and C-terminal phosphorylation in T24, J82 and UMUC3 cell lines (Supplementary Fig. 9b). pSMAD2C phosphorylation was dependent upon c-SRC activation as treatment with the c-SRC inhibitors AZD0530 and PP1 decreased the phosphorylation levels of SMAD2C (Supplementary Fig. 9c–e). An increase in the pERK was also observed upon treatment with AZD0530 (Supplementary Fig. 9c–e). Furthermore, ectopic expression of SMURF2 or SMURF2(FF) in UMUC3 cells increased pERK levels, while SMURF2(EE) enhanced pSMAD2C (Supplementary Fig. 9f).

As HGF leads to TGFβ-mediated EMT in NBT-II cells, we explored if the TGFβ pathway was essential for tumourigenesis, EMT and invasion in bladder cancer. To this end, we utilised the cell lines UMUC3 and T24, the two cell lines which displayed the highest and lowest levels of activated c-Met, respectively. In both cell lines HGF exposure enhanced invasion as observed using a

**Fig. 5** c-SRC phosphorylation of SMURF2 inhibits SMURF2 ligase activity. **a** Representative structure from MD simulation of the C2-HECT domain complex. The HECT domain is shown in electrostatic surface and ribbon representation while the C2 domain is shown in ribbon representation in green colour. The region that encloses Y434 in HECT and the [82]HKKIHKK[88] motif in C2 is highlighted within a box (**b**, **c**). **b** Magnified representation of highlighted region in HECT without the ribbon for clarity. **c** Representative structure from MD simulation of the phosphorylated Y434, highlighting the equivalent region in HECT and C2. The electrostatic potential molecular surface shown here (and in Fig. 4b, c) was created using the APBS plugin through the Pymol molecular visualisation software. A colour gradient from blue to red represents the range of surface potential kT/e values from strongly positive (+3.0) to strongly negative (−3.0). **d** Time series plot (in angstrom unit) of the minimum distance between Y434 (OH or P atoms) of HECT and K87 (NZ atom) of C2 in the Phosphorylated (red colour) and Unphosphorylated (black) state of Y434 during the MD simulation of HECT-C2 complex. **e** 293 T cells transfected with wild-type MYC-SMURF2 or indicated SMURF2 mutants and HA-tagged ubiquitin. Following immunoprecipitation of MYC-SMURF2 lysates were resolved by SDS–PAGE and probed with indicated antibodies. **f** 293 T cells transfected with WT MYC-SMURF2 or indicated SMURF2 mutants and HA-tagged ubiquitin. Following immunoprecipitation of MYC-SMURF2 lysates were resolved by SDS–PAGE and probed with indicated antibodies. **g** 293 T cells transfected with wild-type MYC-SMURF2 or indicated SMURF2 mutants, HA-tagged ubiquitin and/or Flag-tagged c-SRC. Following immunoprecipitation of MYC-SMURF2 lysates were resolved by SDS–PAGE and probed with the indicated antibodies

transwell migration assay an effect diminished upon co-treatment with JNJ38877605 indicating that HGF-mediated invasion in UMUC3 and T24 cells is dependent upon active c-Met receptor signalling (Supplementary Fig. 10a–d). Correspondingly, treatment of UMUC3 cells with JNJ38877605 limited anchorage-independent growth (Supplementary Fig. 10e, f). As HGF enhanced invasion in UMUC3 cells we sought to understand if this function might be dependent on increased EMT. Indeed, UMUC3 cells treated with HGF led to a decrease in the epithelial marker E-cadherin, an effect abolished upon the addition of JNJ38877605 (Supplementary Fig. 10g). Similarly, treatment with the TβR inhibitor A83-01 in UMUC3 cells enhanced the levels of cortical actin indicating that TβR suppression reverts UMUC3 cells to a more epithelial state (Supplementary Fig. 10h). Taken together, these results suggest that TGFβ signalling is required for HGF-mediated EMT induction, and invasion in bladder cancer cell lines.

As, HGF induces c-SRC to phosphorylate and inhibit SMURF2 leading to increased TβR signalling, we transfected UMUC3 and T24 cells with either WT SMURF2, SMURF2(EE) or SMURF2 (FF) to see if the corresponding c-SRC phosphorylation site mutants of SMURF2 play a role in HGF-mediated invasion and EMT. As can be seen in Supplementary Figs. 11A, 10C, SMURF2 (EE) expression enhanced the invasion capacity of both UMUC3 and T24 cells compared with cells expressing WT SMURF2 an effect further enhanced in cells treated with HGF. In contrast, the invasion capacity of UMUC3 and T24 cells expressing SMURF2 (FF) was diminished (Supplementary Fig. 11a–d). Furthermore, ectopic expression of SMURF2(FF) enhanced the levels of β-catenin at cell–cell contacts while concomitantly decreasing overall levels of vimentin, indicating that TβR suppression by SMURF2 leads cells to acquire a more epithelial state (Supplementary Fig. 11e).

Next, we sought to analyse the role of corresponding SMURF2 mutants on targeted inhibition of MAPK and TGFβ pathways. It would be expected that SMURF2(FF) mutants, which downregulates TGFβ signalling but upregulates the MAPK pathway, would be sensitive to MAPK inhibition. In contrast, SMURF2(EE) mutants, which downregulates MAPK signalling but upregulates the TGFβ pathway, would similarly be sensitive to inhibitors that target the TGFβ pathway. As can be seen in Supplementary Figs. 12A, B, UMUC3 cells transfected with SMURF2(FF) were significantly more sensitive to the MEK inhibitor PD0325901 than WT SMURF2 or untreated SMURF2(FF) expressing cells in forming colonies under anchorage-independent conditions. Moreover, UMUC3 cells expressing SMURF2(EE) were significantly more sensitive to TGFβ inhibition with LY2157299 than WT SMURF2-expressing cells. Interestingly, both SMURF2(FF) and SMURF2(EE) mutants displayed lower colony formation than WT SMURF2, indicating that constitutive activation of both TGFβ and MAPK signalling is required for anchorage-independent growth

(Supplementary Fig. 12a–d). Further analysis of the intercellular responses indicated that treatment with PD0325901 completely abolished SMURF2(FF) pERK levels while correspondingly LY2157299 treatment completely abolished pSMAD2C levels in SMURF2(EE) cells (Supplementary Fig. 12e, f).

To further investigate if SMAD2C phosphorylation is critical in bladder tumorigenesis, UMUC3 cells stably expressing SMAD2 WT or a SMAD2 mutant (SMAD2 dnC), containing alanine residues within the respective C-terminal phosphorylation motif, were subcutaneously injected into mice. The presence of SMAD2 C-terminal mutations caused an 80% reduction in overall tumour growth formation thus underscoring the role for TGFβ signalling in bladder cancer tumourigenesis (Supplementary Fig. 13a–c).

Based on these results we used an orthotopic mouse model of bladder cancer whereby we could analyze the role of c-MET-mediated invasion. We choose to utilise UMUC3 cells in this setting as they expressed the highest levels of activated c-MET. To detect the growth of UMUC3 tumours in the mouse bladder, UMUC3 carcinoma cells expressing a mammalian codon-optimised firefly luciferase2 was generated permitting bioluminescence detection. UMUC3-luc2 cells were inoculated into trypsin pre-treated bladders of 6–8-week-old mice and the presence of tumours was confirmed a week later using the bioluminescence in vivo imaging system. When tumours reached a bioluminescence intensity of $5e^8$ photons/sec mice were paired and treated intraperitoneally with either dimethyl sulfoxide (DMSO) or A83-01 or LY2157299. Bladder tumours sporadically shed cells into the urine. Therefore, to analyze the effect of A83-01 on cellular homoeostasis urine was collected and placed in cell culture medium. Cells collected from tumour bearing mice treated with A83-01 clustered together initiating cell–cell contacts while those from control-treated mice conferred a similar morphology to parental UMUC3 cell line, indicating the effective targeting of the tumour with A83-01 (Supplementary Fig. 14a). Notably, TGFβ inhibitor-treated mice displayed a significant decrease in tumour size (Supplementary Fig. 14b–e). After 2 weeks of treatment, bladders were collected and H&E staining was performed. Importantly, tumours isolated from control-treated mice exhibited a near-complete breakdown of the submucosa layer displaying characteristics of blood and lymphatic invasion similar to the phenotype observed in later stage bladder cancers (Fig. 7d, e, Supplementary Fig. 14f). In contrast, mice treated with either the TβR inhibitors A83-01 or LY2157299, tumour formation remained superficial with minimal invasion of the submucosal layer (Fig. 7d, e, Supplementary Fig. 14f). The overall percentage of bladder wall invasion in A83-01 or LY2157299 treated mice was 14% and 20.6%, respectively, compared with 89% in control-treated mice. Furthermore, half of control-treated mice displayed limited but significant lung metastasis

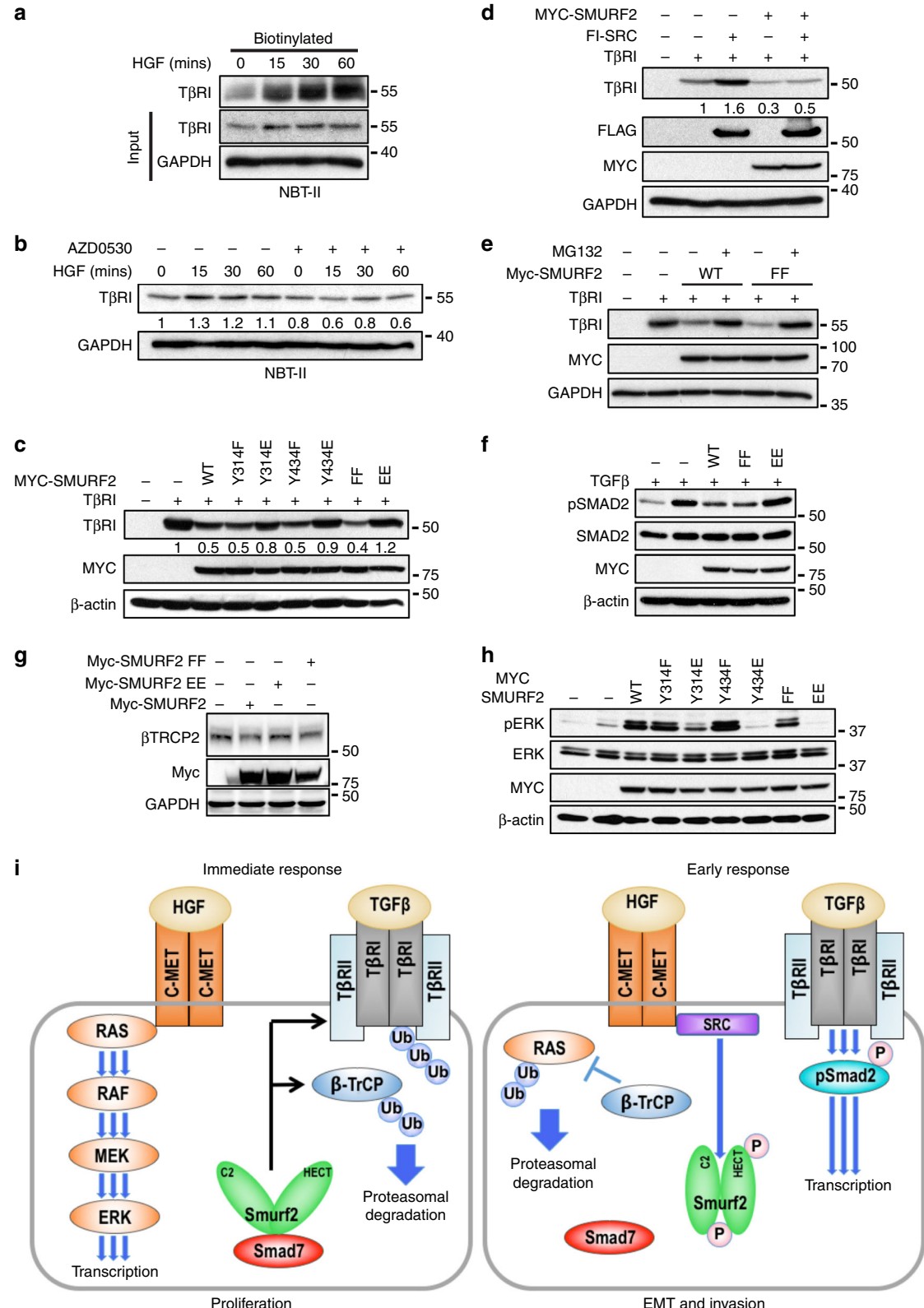

(four out of eight) while no lung metastasis was detected in TβR inhibitor-treated mice (Supplementary Fig. 14g–j). Importantly, however, the remaining tumour maintained a high tumour proliferation capacity. Taken together, these data indicate that TβR inhibition induces a change in cell morphology limiting the invasion capacity, but not the proliferation, of a subpopulation of bladder carcinoma cells.

**Combined inhibition of MAPK and TGFβ signalling pathways suppresses HGF-mediated proliferation and invasion.** As observed, TGFβ inhibition resulted in a significant decrease in tumour size but did not completely eliminate the tumour, therefore we hypothesise that TβR inhibition may result in the loss of negative feedback mechanisms in a subpopulation of the tumour, resulting in a compensatory activation of MAPK

**Fig. 6** SMURF2 acts as a bimodal switch between TβR and MAPK activation. **a** NBT-II cells were treated with HGF for indicated time points and subsequently with biotin for 30 mins at 4 °C. Following immunoprecipitation with NeutrAvidin resin lysates were resolved by SDS–PAGE and probed with indicated antibodies. **b** NBT-II cells were treated with HGF (5 ng/ml) and/or AZD0530 (1 μM) and lysed at the indicated time points. Whole-cell extracts were probed with the indicated antibodies. **c** 293 T cells were transfected as indicated. Whole-cell extracts were probed with the indicated antibodies. TβRI in relation to β-actin was quantified using ImageJ. **d** 293 T cells were transfected as indicated. Whole-cell extracts were probed with the indicated antibodies. TβRI in relation to β-actin was quantified using ImageJ. **e** 293 T cells were transfected as indicated. After 48 h, cells were treated overnight with MG132 (2 μM). Whole-cell extracts were probed with the indicated antibodies. **f** 293 T cells were transfected as indicated. After 48 h, cells were treated with TGF-β (2.5 ng/ml) overnight. Whole-cell extracts were probed with the indicated antibodies. **g** 293 T cells were transfected as indicated. Whole-cell extracts were probed with the indicated antibodies. **h** 293 T cells were transfected as indicated. Whole-cell extracts were probed with the indicated antibodies. **i** Schematic of HGF-induced TGFβ activation and corresponding downregulation of MAPK pathway

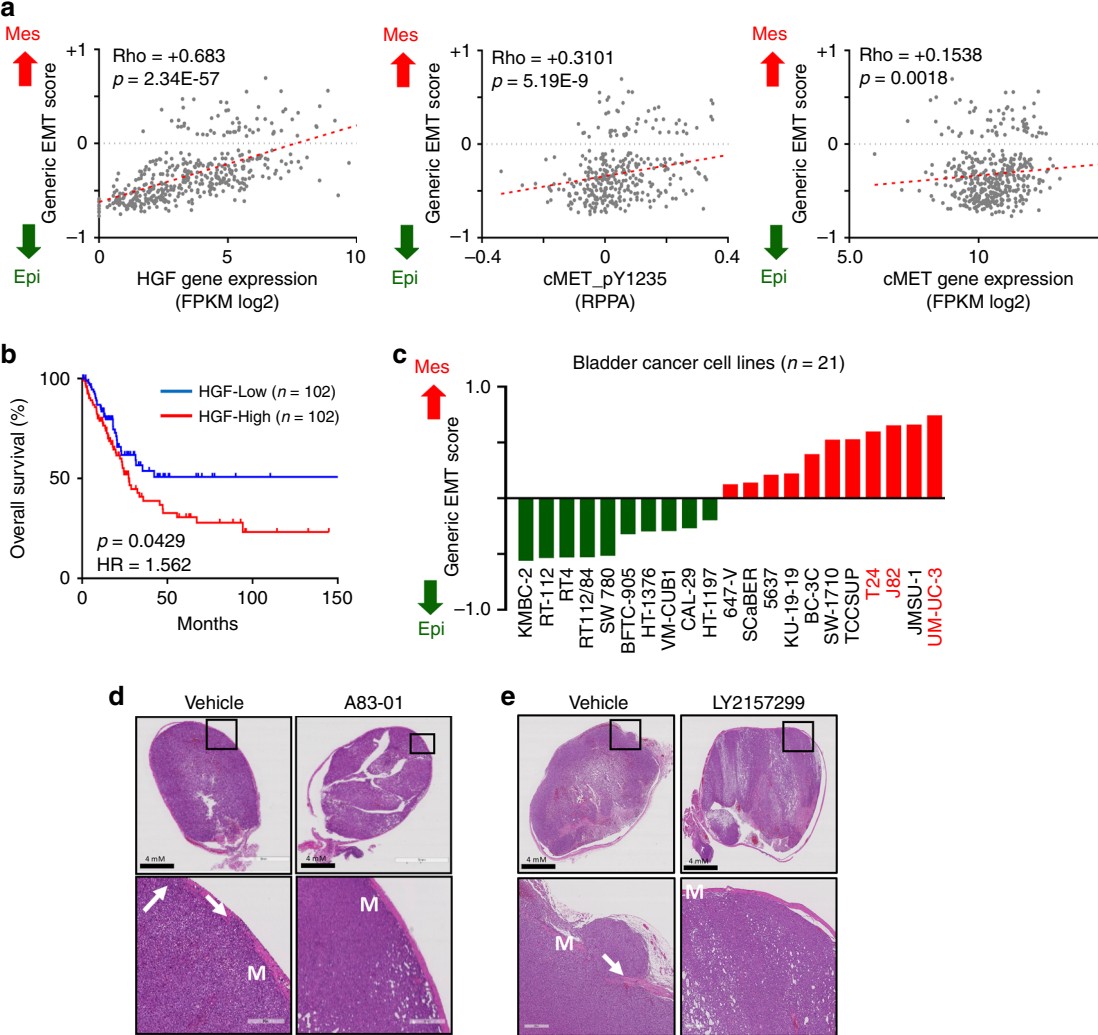

**Fig. 7** HGF-induced invasion is abrogated by TβR inhibition. **a** EMT scores of TCGA Bladder Cancer data set (BLCA $n = 408$) compared with HGF expression (left panel; $R = 0.683$, $p = 2.3E\text{-}57$), phospho c-Met expression (middle panel; $R = 0.3101$, $p = 5.19E\text{-}9$) and c-Met expression (right panel; $R = 0.1538$, $p = 0.0018$). The $p$ value is computed by Spearman correlation coefficient test. **b** Kaplan–Meier curves showing probability of overall survival of bladder cancer patients with higher copy number of HGF is significantly less than those with lower level of HGF (BLCA $n = 408$, $p = 0.0429$, HR = 1.562). The $p$ value is computed by log-rank test. **c** EMT scores of 21 bladder cancer cell lines defined as either epithelial (green) or mesenchymal (red). **d**, **e** Analysis of bladder tumours formed from intraluminally implanted UMUC3 cells in the absence or presence of TβR inhibitors. Treatment was started when bioluminescence intensity reached ∼5e$^8$ photons/sec. Mice were treated for 2 weeks and tumour volumes were measured twice a week. M = muscularis propria. **d** H&E staining of representative tumour samples. Daily administration of vehicle control only (control; $n = 5$) or A83-01 (50 mg/kg; $n = 5$). T-stages at end of experiment in control-treated mice indicated T3-3 mice, T2-2 mice. All A83-01 treated mice were T1. Scale bars: 4 mm. **e** H&E staining of representative tumour samples. Daily administration of vehicle control only (control; $n = 3$) or LY2157299 (80 mg/kg; $n = 3$). T-stages at end of experiment indicated T3-1 mice, T1-2 mice. All LY2157299 treated mice were T1. Scale bars: 50 mm

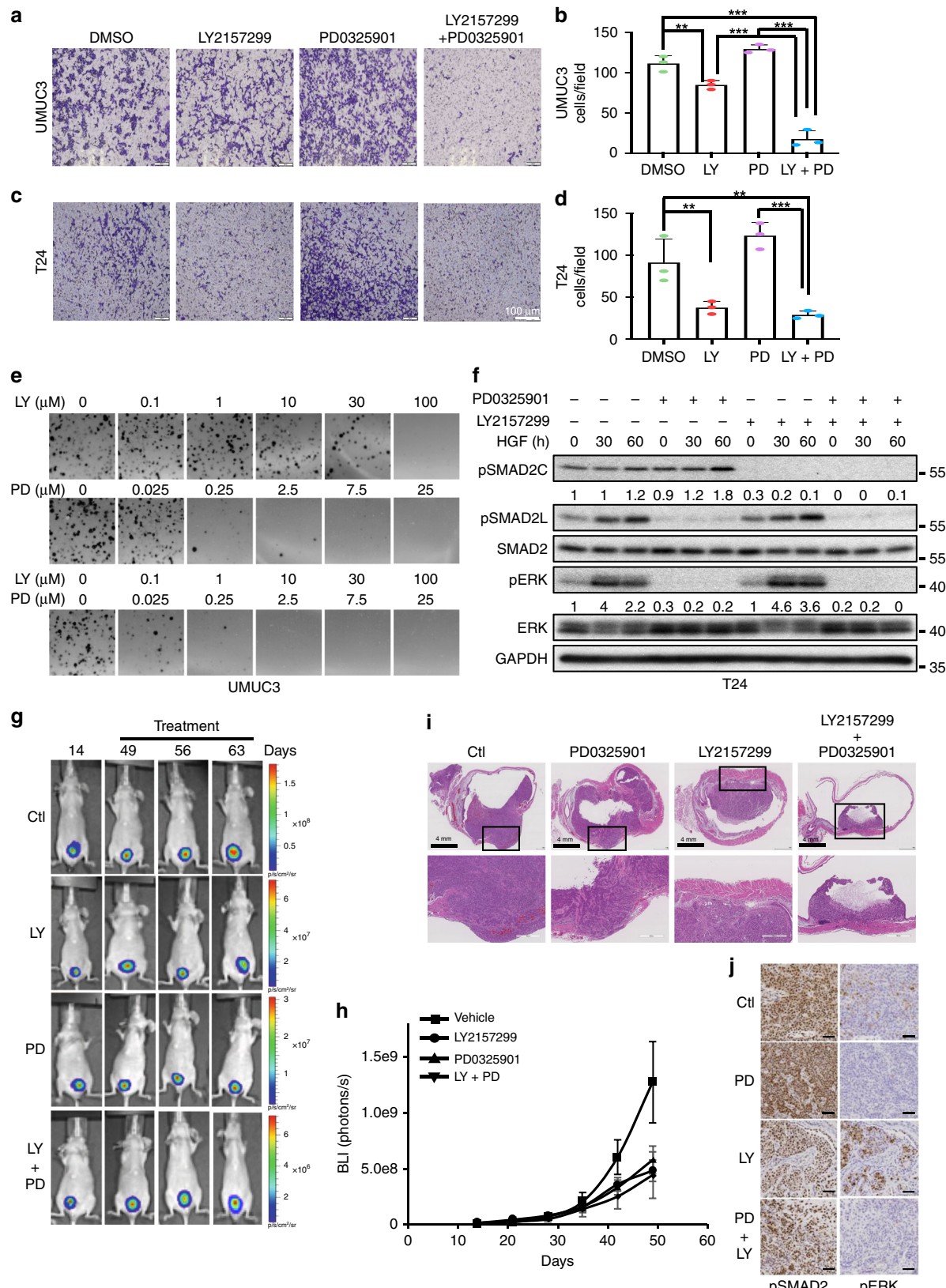

signalling. Furthermore, it would be predicted that loss of MAPK signalling would induce TβR activation. To this end, we treated both UMUC3 and T24 cells with either LY2157299 or PD0325901, or the combination, and interrogated the invasion capacity of both cell lines. Treatment with the TGFβ inhibitor LY2157299 diminished the invasion capacity of both cell lines

(Fig. 8a, c). In contrast, PD0325901 partially enhanced the invasion capacity of both cell lines potentially owing to the activation of compensatory mechanisms, resulting in the activation of TGFβ signalling. However, dual inhibition of both TGFβ and MAPK signalling nearly completely abolished the invasion capacity of these cell lines (Fig. 8a–d). Next, we sought to

**Fig. 8** HGF-induced invasion and proliferation is abrogated by TβR/MEK inhibition. **a** Transwell assay of UMUC3 cells treated with LY2157299 (1 μM) or PD0325901 (1 μM) or the combination for 16 h prior to fixation and crystal violet staining. Scale bars: 100 μm. **b** Graph represents average number of migrated cells taken from four different random fields from panel. Data are mean ± SD of triplicate samples from a representative experiment performed three times. Student's *t* test compares the treated populations, *P < 0.05, ***P < 0.001. **c** Transwell assay of T24 cells treated with LY2157299 (1 μM) or PD0325901 (1 μM) or the combination for 16 h prior to fixation and crystal violet staining. **d** Graph represents average number of migrated cells taken from four different random fields from panel. Data are mean ± SD of triplicate samples from a representative experiment performed three times. Student's *t* test compares the treated populations, **P < 0.01, ***P < 0.001. **e** Soft agar growth assay of UMUC3 cells treated with either LY2157299, PD0325901 or both. **f** Western blot analysis of T24 cells treated with 5 ng/ml of HGF in the presence or absence of LY2157299 (1 μM), PD0325901 (1 μM) or both. Lysates were collected at denoted time points and probed with indicated antibodies. pSMAD2C in relation to total SMAD2 and pERK in relation to total ERK was quantified using ImageJ. **g** Bioluminescent images of mice injected luciferase tagged T24 cells intramurally in the bladder. Daily administration of vehicle control only (control), PD0325901 (25 mg/kg), LY2157299 (80 mg/kg) or the combination started when bioluminescence intensity reaches ~ 2e$^8$ photons/sec. Bioluminescence was measured once a week (*n* = 5/treatment group). Scale bars = p/sec/cm$^2$/sr. **h** Quantitative analysis of bioluminescent signal intensity over days after injection. ***P value < 0.001. **i** H&E staining of representative tumour samples from **g**. Scale bars: 4 mm. **j** Tumours generated in **g** were analysed by IHC for pSMAD2 and pERK. Representative images are displayed. Original magnification ×400. Scale bars: 50 μm

determine whether the dual inhibition of MAPK and TGFβ signalling correlated with diminished anchorage-independent growth. Although both LY2157299 and PD0325901 were effective at reducing growth in soft agar of UMUC3 cells, combinatorial treatment significantly inhibited colony forming ability compared with either compound alone (Fig. 8e). Similar effects were observed using A83-01 and PD0325901 (Supplementary Fig. 15). To further assess the cross-activation of TGFβ and MAPK signalling with MEK and TβR inhibitors, respectively, we treated T24 cells with either LY2157299 or PD0325901 or the combination in the presence of HGF and observed the intercellular responses. Consistent with our previous results, treatment of T24 cells with the MEK inhibitor PD0325901 upregulated pSMAD2C, whereas treatment with the TβR inhibitor LY2157299 enhanced pERK levels (Fig. 8f). Importantly, treatment with the combination of LY2157299 and PD0325901 completely abolished pSMAD2C and pERK levels (Fig. 8f).

As most bladder cancers eventually invade the surrounding smooth muscle, we sought to generate an orthotopic bladder cancer model by injecting bladder cancer cells intramurally to gain insight into the potential anti-tumour invasive activity of MEK and TβR inhibition in vivo. T24 cells were generated to stably express a mammalian codon-optimised firefly luciferase2 permitting bioluminescence detection. Following tumour formation, mice were paired and treated by oral gavage with either vehicle or LY2157299 or PD0325901 or a combination of the two. Similar to our previous observations using our intravesical inoculation model, the addition of LY2157299 decreased tumour size and inhibited the invasion of established tumours through the submucosa and muscular layers, however, as before, remaining tumours maintained a high level of proliferative capacity (Fig. 8g–i). In contrast, although MEK inhibitor treated tumours decreased in overall size, tumour invasion remained equivalent to control-treated tumours (Fig. 8g–i). Treatment with the combination MEK and TβR inhibition not only abrogated tumour cell invasion but the remaining tumour size was decreased compared with treatment with either inhibitor alone (Fig. 8g–i). Furthermore, in vivo pharmacodynamics assessment shows that treatment with PD0325901 enhances pSMAD2, whereas treatment with LY2157299 enhances pERK an effect, which is circumvented when mice are treated with the combination (Fig. 8j). These set of experiments confirm that the combination of both drugs is required to effectively inhibit invasion and proliferation.

## Discussion

The invasive potential of carcinoma cells is determined by a multitude of factors, including the ability of cells to undergo EMT. Furthermore, it has now become apparently clear that the surrounding tumour microenvironment can promote pro-invasive phenotypes. The recent observation that EMT occurs primarily within the invasive front of the tumour underlies the role of the microenvironment in this setting[12–14]. HGF/c-MET signalling has been shown to promote tumour progression, particularly influencing tumour invasiveness and metastatic potential[52,53]. HGF and soluble Met levels are highly expressed in the serum of bladder cancer patients, positively correlating with disease progression[16,54]. Although some multi-kinase inhibitors that target Met have received regulatory approval for the treatment of bladder cancer overall survival rates in MIBC remain relatively unchanged. The goal of the present study was to identify the mechanism underlying the function of HGF/c-Met in bladder tumourigenesis.

Recently, we performed a meta-analysis of 2411 bladder cancer patients, allowing to identify six major molecular subtypes including two molecular subtypes, MES and SCC, which exhibit high EMT score[51]. To further understand the role of EMT in bladder cancer we utilised an in vitro-based model, which can undergo EMT in short-term cultures and compared real-time changes in cell plasticity with transcriptomics following HGF treatment. This allowed a comparison between gene expression in cells undergoing phenotypic EMT. HGF treatment and concomitant c-Met activation results in the activation of a number of downstream pathways including PI3K, RAC/PAK, MAPK, etc. We also observe a rapid upregulation of pERK following HGF stimulation. Interestingly, sustained pERK was not observed with phosphorylation levels, declining over time until baseline pERK levels were reached. Accompanying the decline in pERK, we observed an increase in p-c-SRC, and pSMAD2L pSMAD2C. The central dogma suggests that pSMAD2C can only occur downstream of TGFβ-like receptors, indicating that HGF may stimulate TβR activity. Accordingly, inhibition of TGFβ pathway activation using either the TβR kinase inhibitors or knockdown of SMAD2 inhibited HGF-induced scattering and loss of EMT marker expression, suggesting that TGFβ activity is critical for HGF-induced EMT and invasion. Furthermore, transcriptomic analysis revealed that HGF treatment induces a subset of genes corresponding to known TGFβ and EMT signatures.

In advanced cancers, TGFβ functions as an oncogenic factor enhancing proliferation, invasion and metastasis[55,56]. Our study reveals that HGF/c-MET requires a functional TβR signalling component for this to occur. Recently, it has been demonstrated that TGFβ family ligands correlate with a more aggressive phenotype in bladder cancer cell lines and deregulated TGFβ signalling leads to enhanced migration and invasiveness of bladder cancer cells[57]. We demonstrate that in orthotopic mouse models, inhibition of TβR signalling completely reverses the invasion capacity of the bladder cancer cell line UMUC3. However, we

noted that loss of TβR signalling had modest impact on tumour regression. Further in vitro and in vivo analysis revealed a crosstalk between MAPK and TβR signalling, whereby inhibition of either MAPK or TGFβ would result in hyperactivation of the other. Importantly, combined inhibition of both MAPK and TβR resulted in a decrease in both clonogenic capacity and invasive growth capability. For these reasons, we favour the initial clinical development of this combination in bladder cancer patients with aggressive T1 tumours.

The primary function of c-SRC is to couple receptor activation with the cytoplasmic signalling machinery, thereby regulating a number of fundamental cellular processes. c-SRC has long been shown to be positive regulator of EMT but the mechanistic actions of c-SRC in this setting remain vague[35]. To prevent uncontrolled ubiquitination of its substrates, SMURF2 ligase activity is obstructed through the formation of intramolecular interactions between the C2 and HECT domains[32]. The C2-HECT formation is antagonised upon the binding of SMAD7, resulting in the activation of SMURF2 and ubiquitination of SMURF2 substrates including the TβR complex and β-TrCP[28,49]. β-TrCP targets K-RAS for proteasome-mediated degradation. The net result of SMURF2 activation is downregulation of TGFβ signalling and concomitant upregulation of MAPK activity through the stabilisation of K-RAS. We show that HGF induces c-SRC tyrosine phosphorylation of SMURF2 at Tyr314 and Tyr434, resulting in the inhibition of SMAD7 binding and enhancement of electrostatic interactions between the HECT domain and C2 domain, respectively, resulting in downregulation of SMURF2 ligase activity. There are a number of direct implications to these results. First, SMURF2 inhibition by c-SRC upregulates the TβR pathway by stabilising TGFβR complex a function we define to be critical for HGF-induced EMT and invasion in vitro and tumour progression in vivo in bladder cancer. Second, the observation that c-SRC phosphorylation of SMURF2 results in the downregulation of MAPK activity reveals the identification of a novel adaptive response mediated by a c-SRC-SMURF2 axis to downregulate MAPK signalling following HGF stimulation. This is unsurprising as a number of negative feedback loops have already been identified, which regulate this kinase cascade[58–60]. Taken together, our results indicate that HGF induces c-SRC-mediated phosphorylation of SMURF2 whereby SMURF2 acts as a temporal bimodal switch between MAPK and TGFβ signalling. This places SMURF2 at a key node in the regulation of either a pro-proliferative or invasive phenotypes.

As c-SRC is activated downstream by a number of RTKs, it would be unsurprising if loss of c-SRC activity by EGFR, HER2 or FGFR inhibitors would not have a deleterious effect on downregulation of the MAPK pathway as SMURF2 would remain active and as a consequence K-RAS would remain stable. It has recently been shown that EGFR activation results in c-SRC-mediated phosphorylation of NEDD4, suggesting that a similar outcome may occur with SMURF2. Therefore, future clinical studies may be required to incorporate MEK inhibitors with either EGFR, FGFR or HER2 inhibitors in relevant patient settings to observe complete downregulation of MAPK signalling. Furthermore, our observations suggest that c-SRC inhibitor monotherapy may function to prevent invasion and metastasis in high-grade T1 stage bladder cancers particularly in BCG refractory patients. In summary our findings suggest that TβR activation is a critical factor in c-MET induced bladder cancer invasion and further studies into the potential use of TβR and MAPK inhibitors is warranted.

## Methods

### Cell lines and culture conditions
The transitional carcinoma cell lines NBT-II, T24, J82, UMUC3 and human embryonic kidney cells HEK293T were obtained from the American Type Culture Collection (ATCC; catalogue no. CRL-1655 and CRL-1749). Cells were routinely cultured in Dulbecco's modified eagle medium (DMEM, Gibco, Life Technologies) supplemented with 10% fetal bovine serum (FBS, HyClone Thermo Scientific), and 100 units/ml penicillin–streptomycin (1 × pen–strep, Invitrogen). Cultured cells were regularly tested for mycoplasma contamination. Cells were grown in a 5% $CO_2$ atmosphere incubator at 37 °C. When appropriate HGF (5 ng/ml, Calbiochem), TGF-β1 (2.5 ng/ml, R&D), SB431542 (5 μM, Tocris), A83-01 (8 μM, Tocris), A83-01 is a small molecule inhibitor specifically targeting TβRI (ALK5), Activin receptor type-1B/ALK4 and Activin receptor type-A/ALK-7, the three of which contain highly structurally related kinase domains, PD0325901 (1 μM, Selleck Chemicals), LY2157299 (1 μM, Selleck Chemicals), AZD0530 (2 μM, Selleck Chemicals), JNJ38877605 (4 μM, Selleck Chemicals) or MG132 (2 μM; Sigma) was added. All cell lines were independently genotyped and validated (please refer to source data).

### EMT inhibition and cell tracking
EMT induction with HGF and inhibitory effects of selected compounds were validated by NBT-II epithelial colony time-lapse microscopy. NBT-II cells were plated onto 12-well plate (BD) at a low density of 200 cells per well in 2 ml of medium. Cells were allowed to grow and form epithelial colonies for a period of 4 days. The cultures were then refreshed with medium containing with or without 5 ng/ml HGF or compound 2 μM AZD0530 (Selleck Chemicals) prior to time-lapse imaging. Time-lapse microscopy of individual cell colonies was performed using a video microscope incubator system (Axiovert-200M, Carl Zeiss). Time-lapse images were taken at 5 min intervals for 24 h. Tracking of cells was performed with the particle tracking function in Metamorph software (Metamorph).

### Transient transfection
HEK293T cells were transfected using calcium chloride and 4-(2-hydroxyethyl)-1-piperazineethanesulfonic acid (HEPES)-buffered saline (pH 6.95). Sixteen hours after transfection, cells were washed twice with 1 × phosphate-buffered saline (PBS) and replenished with fresh media. The cells were grown for a further 24 h before processing for cell lysis. UMUC3 cells were transfected using lipofectamine 3000 according to manufacturer's protocol.

### Stable transfection
NBT-II were transiently transfected with a plasmid harbouring a kinase-dead mutation K295R or constitutive active mutation Y527F in c-SRC or kinase-dead TGFβR I using lipofectamine 3000, according to manufacturer's protocol (Invitrogen). UMUC3 were stably transfected with a plasmid harbouring either dominant negative SMAD2 having point mutation S/T phosphorylation sites in the linker region (T220V, S245A, S250A, S255A) or deletion of four amino acids at the carboxyl terminal region (deposited by Rik Derynck; Addgene plasmid 12637) followed by selection for single clones by supplementing 1 μg/ml puromycin (Invitrogen) to the media. pLNCX chick c-SRC K295R was a gift from Joan Brugge (Addgene plasmid # 13659); pLNCX chick c-SRC Y527F was a gift from Joan Brugge (Addgene plasmid # 13660); pCMV5B-TGF-beta receptor I K232R was a gift from Jeff Wrana (Addgene plasmid # 11763); pCMV5 Flag-SMAD2 EPSM was a gift from Joan Massague (Addgene plasmid # 14933); LPCX SMAD2 deltaSSMS was a gift from Rik Derynck (Addgene plasmid # 12637).

### Immunofluorescence
NBT-II cells were seeded at a density of 100 cells per well on Ibidi eight-well μ-slides and grown for 4 days to form colonies. Cells were pretreated with 5 ng/ml of HGF (Calbiochem) and then fixed with 4% paraformaldehyde and stained with mouse anti-rat Desmoplakin (clone DP2.15, Millipore), mouse anti-human E-cadherin (clone 36/E-cadherin (RUO), BD), mouse anti-human Vimentin (clone V9, Abcam) and mouse anti-Smad2/3 (clone 18Smad2/3, BD) primary antibodies. A goat anti-mouse antibody conjugated to Alexa Fluor 488 (Invitrogen) was used as a secondary antibody. Fluorescent images were captured with a Leica DM6000b inverted microscope (Leica Microsystems) and monochrome coolsnap HQ2 camera (Photometrix).

### Immunohistochemistry
Immunohistochemistry was performed by AMPL A*STAR. In brief, samples are paraformaldehyde fixed and paraffin embedded, cut on microtome (6 μm thickness) and deparaffinised, rehydrated and heated in Antigen Retrieval Citrate Solution, pH 6 (Abcam) before staining. Sections were incubated with Dual Endogenous Enzyme Block (Dako, Glostrup, Denmark), and were treated with Protein Block Serum-Free (Dako). Slides were incubated at room temperature for 1 h with rabbit monoclonal anti-human pERK (T202/Y204) antibody (clone D13.14.4E, dilution 1:400), rabbit monoclonal anti-human pSmad2 (S465/467) antibody (clone 138D4, dilution 1:50), rabbit monoclonal anti-human pSmad2 (S245/250/255) antibody (CST #3104, dilution 1:50) and rabbit monoclonal anti-human pSrc (Y416) antibody (clone D49G4, dilution 1:100) (followed by DAB substrate kit (Abcam) and counterstained with hematoxylin. Slides were scanned with Amperio slide scanner (Leica) and analysed by pathologists at AMPL and at the Hopital Foch (Paris, France). Declaration Number for the cohort of Bladder Cancer tissue samples used in this study No. DC-2017-2942 ministry of research, division of bioethics, Head of Project Rouanne Mathieu.

**Phospho-specific protein microarray analysis**. NBT-II cells were seeded at a density of $0.5 \times 10^6$ cells per 10 cm cell culture-treated dish (NUNC) and grown for 4 days to form colonies. Cells were either treated with or without 5 ng/mL of HGF (Calbiochem) for 2 hours before harvesting. Cells were harvested by incubating cells with dissociation buffer (Gibco) for 10 min and centrifuged at $300 \times g$ for 5 min. Cell pellets were snap frozen with liquid nitrogen and sent to Fullmoon Biosystems (Sunnyvale, CA, USA) for processing, hybridisation and phospho-array detection of the TGF-β Signalling Phospho-Specific Antibody Array (Fullmoon Biosystems, Sunnyvale, CA, USA). Fluorescence intensity was scanned with an Axon GenePix 4400 A microarray scanner (Axon, Sunnyvale, CA, USA) at 760 nm wavelength. Each slide was analysed by the software GenePix Pro 7126 and phosphorylation ratio was calculated from the raw data with phospho value/unphospho-value (provided by Fullmoon Biosystems).

**Microarray data analysis**. HGF-treated NBT-II bladder carcinoma cells at 0, 2, 4, 6, 9, 24, 48 h were subjected to microarray gene expression profiling on Affymetrix Rat Gene 1.0 ST platform. The data had been deposited in Gene Expression Omnibus (GEO) with the accession GSE104740. The preprocessing of microarray was performed using Partek software version 5.0 (Partek Incorporated). Probeset intensities were normalised using RMA algorithm. Log2 probe intensities of below 6 were deemed low expression and were filtered off from the analysis. Significant differentially expressed genes were chosen with criteria of FDR 0.01 and a minimum fold change of 1.2.

**Mass spectrometry analysis**. Samples were treated by in-gel digestion prior to MS analysis. In brief, samples were reduced in 10 mM dithiothreitol (DTT) for 1 h at 56 °C followed by alkylation with 55 mM iodoacetamide (Sigma) for 45 min in the dark. Tryptic digest was performed overnight in 50 mM ammonium bicarbonate buffer with 2 µg trypsin (Promega) at 37 °C. Peptides were desalted on StageTips and analysed by nanoflow liquid chromatography on an EASY-nLC 1200 system coupled to a Q Exactive HF mass spectrometer (Thermo Fisher Scientific). Peptides were separated on a C18-reversed phase column (25 cm long, 75-µm inner diameter) packed in-house with ReproSil-Pur C18-QAQ 1.9-µm resin. The column was mounted on an Easy Flex Nano Source and temperature controlled by a column oven (Sonation) at 40 °C. A 105-min gradient from 2 to 40% acetonitrile in 0.5% formic acid at a flow of 225 nl/min was used. Spray voltage was set to 2.4 kV. The Q Exactive HF was operated with a TOP20 MS/MS spectra acquisition method per MS full scan. MS scans were conducted with 60,000 and MS/MS scans with 15,000 resolution. The raw files were processed with MaxQuant version 1.5.2.8 with preset standard settings for SILAC labelled samples and the re-quantify option was activated[61]. Carbamidomethylation was set as fixed modification while methionine oxidation, protein N-acetylation and phosphorylation (STY) were considered as variable modifications. Search results were filtered with a false discovery rate of 0.01.

**Statistical analysis**. Pathway enrichment analyses were performed using signature from Molecular signature database (Msigdb v5.0) and ssGSEA projection method implemented in R 3.3.1 bioconductor package GSVA 1.20.0. P value is computed by two-tailed unpaired t test.

**Nuclear cytoplasmic fractionation assay**. NBT-II cells were seeded at a density of $0.5 \times 10^6$ cells per 10 cm cell culture-treated dish (NUNC) and grown for 4 days to form colonies. Cells were pre-treated with 5 ng/ml of HGF (Calbiochem) alone or HGF with 8 µM A83-01 (Tocris Bioscience) before harvesting. Cells were trypsinized and the number of cells in each sample was quantified. Equal number of cells was pelleted for each sample. Fractionation was performed using Qproteome cell compartment kit (Qiagen), according to manufacturer's protocol.

**Site directed mutagenesis**. These PCR reactions were performed using Quik-Change XL kit by Agilent Technologies (catalogue no. 200517-4) according to manufacturer's instructions. The presence of mutants was confirmed by Sanger sequencing.

**Modelling the HECT: C2 domain complex of SMURF2**. The NMR structure of the C2 domain (PDB ID: 2JQZ) and the X-ray crystal structure of the HECT domain (PDB ID: 1ZVD) of SMURF2 protein were downloaded from the Protein Data Bank[29,32]. The lowest energy state model (Model 1) from the ensemble of C2 structures was selected for the study. The effective length considered for the C2 domain was 131 residues (10–140) and for the HECT domain was 371 residues (369–739), residue numbering as per Uniprot ID: Q9HAU4. These two independently solved structures of C2 and HECT (Fig. S7A, B) were used to model the HECT-C2 domain complex using the programme HADDOCK through its web portal[62]. The work by Mari et al. and Wiesner et al.[32,63] where they have mapped the potential binding interface and residues involved in interaction between the HECT and C2 domains of SMURF2, were used as references to specifically define the active (F403, E407, Y454 for HECT; F29, F30, R31, L57 for C2) and passive site residues (R399, E401, I402, M409, Y453, Y482, I489, E666, Q673, S678 for HECT; K27, L32, T56, K84, H86, K88 for C2) for docking. A total of 10 structure clusters

were generated after docking of the two domains. The representative structure from the top cluster with the best score as computed by the scoring scheme used in HADDOCK was selected as an initial model (Fig. S7C). It has also been previously suggested by Wiesner et al.[32] that the C-lobe in HECT could rotate around the well-known flexible linker connecting the N- and C-lobes in order to facilitate a more favourable conformation of the two lobes for C2 interaction as compared with the state observed in the crystal structure of HECT (PDB ID: 1ZVD). So, as a further optimisation to our initial docked model, the C-lobe was rotated around the linker (628GLGKID632) by gradually changing their backbone dihedral angles in steps. This was done till the region from C-lobe, which has the residues predicted to be involved in interaction with C2 was in close proximity to the docked C2 domain (Fig. S7D). Subsequently, the complex structure was refined by energy minimisation to obtain the final representative docked HECT-C2 complex state structure of Smurf2 (Fig. S7E).

**System setup for molecular dynamics simulations**. The NMR structure of SMURF2 WW3 domain in complex with a SMAD7-derived peptide (PDB ID: 2LTZ) was downloaded from the Protein Data Bank[47]. The lowest energy state model (Model 1) from the ensemble of structures was selected for simulation. The effective length for WW3 domain was 37 residues (297–333, residue numbering as per Uniprot ID: Q9HAU4) and for the SMAD7 peptide was 15 residues (203–217, residue numbering as per Uniprot ID: O15105). The refined HECT-C2 complex state obtained from docking (Fig. S7E) was used as a starting state for simulation of this system. The phosphorylated state of Y314 in WW3-SMAD7 complex and Y434 in HECT-C2 complex was respectively modelled in these structures which collectively resulted in a total of four systems for simulation. The N- and C-terminal of the separate chains in all the complexes was capped with ACE and NME functional groups. The complexes were then placed in a cuboid box whose dimensions were set by ensuring a minimum distance of 8 Å between any solute atom and the edge of the box. TIP3P water model was used for solvation, and all the systems were neutralised by adding an appropriate number of counter ions using the TLEAP module of AMBER14[64,65].

**Molecular dynamics simulations**. MD simulations were performed in AMBER14 using PMEMD module and employing the all-atom ff99SB force field[66]. The amber library files for Phosphotyrosine were downloaded from http://research.bmh.manchester.ac.uk/bryce/amber wherein the force field parameters for the modified amino acid were derived from the work by Craft and Legge[67]. Each of the four solvated systems were energy minimised using the steepest descent and conjugate gradient algorithms, heated to a temperature of 300 K over 30 ps in NVT ensemble, equilibrated for another 500 ps and finally subjected to the production run for 20 ns. Both the equilibration and production simulations were done in NPT ensemble. In the simulation of HECT-C2 complexes, harmonic positional restrain of 5 Kcal/mol was applied to all the backbone (N, CA, C and O) heavy atoms. Langevin dynamics with a collision frequency of $1.0 \, ps^{-1}$ were used for temperature regulation and a weak-coupling with relaxation time of 1 ps was employed to maintain the pressure at 1 atm[68–70]. Periodic boundary condition was appropriately applied, and the Particle Mesh Ewald (PME) method was used for calculating long-range electrostatic interactions[71]. All bonds involving hydrogen atoms were constrained using the SHAKE algorithm with an integration time-step of 2 fs[72].

**Western blots and antibodies**. Cells were lysed with radioimmunoprecipitation assay (RIPA) buffer supplemented with Complete Protease inhibitor cocktail (Roche) (RIPA: 50 mM Tris pH 8.0, 150 mM NaCl, 1% NP-40, 0.5% sodium deoxycholate, 0.1% sodium dodecyl sulfate (SDS), 1 mM sodium orthovanadate, 50 mM sodium fluoride, 100 mM β-glycero phosphate)[31]. Whole cell extracts were then separated on 7–12% SDS–PAGE (polyacrylamide gel electrophoresis) gels and transferred to 0.45 µM PVDF membranes (Millipore). Membranes were blocked with bovine serum albumin for all antibodies except phospho-SMAD2, which was blocked in milk and probed with specific antibodies at 4 °C overnight. Blots were then incubated with a horseradish peroxidase-linked second antibody and detected using chemiluminescence (Pierce). The following antibodies were used; anti-E-cadherin (BD), anti-FLAG and anti-α-tubulin (Sigma), anti-HA (Y11, Santa Cruz biotech), anti-Myc (9E10 or A14, Santa Cruz Biotech), anti-phospho-SMAD2 (S465/467, Cell Signaling), anti-Laminin-b1 (Abcam), anti-SMAD2 (L16D3, Cell Signaling), anti-β-actin (Sigma), anti-TGF-β receptor I (V-22, Santa Cruz Biotech), anti-phospho-tyrosine (clone 4G10, Millipore, 05–321), anti-Phospho-Src (Tyr 416, 2101 S, Cell Signaling), anti-c-Src (B-12, Santa Cruz Biotech), anti-phospho-ERK (P-p44/42 MAPK T202/Y204, 9101, Cell Signaling), anti-Total ERK (p44/42 MAPK, 9102, Cell Signaling), anti-p38 (Cell Signaling), p-c-MET Tyr1234/1235 (Cell Signaling) and anti-phospho-SMAD2 (S245/250/255, 3104, Cell Signaling). All antibodies were used at a dilution of 1:1000. The pan TGFβ neutralising antibody (ID11) was purchased from Genzyme, the activin ligand trap soluble ACVR2B-Fc was a kind gift from Olli Ritvos (University of Helsinki, Finland).

**Generation of phospho-specific antibodies**. Phospho-specific antibodies to Tyr314 and Tyr434 of SMURF2 were generated by Biomatik, USA. Rabbit polyclonal antibodies were raised against specific peptide sequences. For Tyr314, peptide sequence corresponding to CEIRNTATGRV(pY)FVDHN was used to raise

antibodies against the phosphorylated form and CEIRNTATGRVYFVDHN for the unphosphorylated form. Similarly, for Tyr434, peptide sequence corresponding to CLWKRLMIKFRGEEGLD(pY)GGVAR was used to raise antibodies against the phosphorylated form and CLWKRLMIKFRGEEGLDYGGVAR for the unphosphorylated form. The antibodies were received in lyophilised form, which was further dissolved using in ddH2O, except antibodies for the unphosphorylated Tyr434, which could only be dissolved in 10% DMSO. All antibodies were used at a dilution of 1:1000.

**Immunoprecipitation and In vivo deubiquitination assay**. For co-immunoprecipitation experiments, cells were lysed in ELB (250 mM NaCl, 0.5% NP-40, 50 mM HEPES, pH 7.3) supplemented with protease inhibitors. Cell lysates were incubated for overnight with the indicated antibodies, then conjugated to protein A or protein G sepharose beads (GE Healthcare), washed three times in ELB buffer, boiled in sample buffer and separated out on SDS–PAGE gels. When appropriate cell lysates were immunoprecipitated with ANTI-FLAG M2 affinity resin (Sigma). Wild-type Myc-tagged SMURF2 (5 μg) or corresponding relevant mutants were co-transfected with HA-ubiquitin (3 μg). After 72 h MG132 (2 μM) was added, incubated overnight and cells were lysed in ELB buffer. SMURF2 was immunoprecipitated with Myc antibodies and bound proteins were resolved by SDS–PAGE gel for further processing by immunoblotting[73].

**Biotin labelling of TGF-β receptor**. Cell surface biotinylation was performed using Pierce Cell surface protein isolation kit (catalogue no. 89881) according to the manufacturer's instructions. NBT-II cells were grown on 15 cm plates treated with HGF (5 ng/ml), media was removed at the indicated time points and cells were biotinylated at 4 °C for 30 min. Thereafter, cells were lysed, protein estimation was performed and equal amounts of lysates were incubated with neutravidin beads (from above kit) for 1 h. The beads were subsequently washed and boiled in sample buffer with 50 mM DTT and loaded onto an SDS–PAGE gel for further processing by immunoblotting.

**Subcutaneous model**. Xenografts were generated by injecting 0.1 ml of $2.5 \times 10^6$ of UMUC3 (wild-type or transfected with either Smad2dnL or Smad2 dnC construct) cells subcutaneously into the dorsal flanks of female BALB/c nude mice (6–8 weeks old). Animal body weight and physical signs were monitored during the experiments. Tumour size was measured weekly by Vernier calliper. The tumour volume was calculated, with the formula: (length × width²)/2. In vivo xenograft models, the mice were gavaged with either 50 mg/kg of A83-01 or 12.5 mg/kg of PD0325901 or a combination of 50 mg/kg of A83-01 and 12.5 mg/kg of PD0325901 daily when tumours reached 50 mm³.

**Mouse orthotopic bladder cancer model**. All animal work adhered to the Agency of Science Technology and Research (A*STAR), Institutional Animal Care and use Committee (IACUC), guidelines on animal use and handling. Six to eight weeks old-female balb/c nude (nu/nu) mice were used. General anaesthesia was induced in the mice with inhaled isoflurane and maintained via nosecone (4% for induction, 1–2% for maintenance). Sterile ophthalmic ointment was applied to the animal's eyes and a heating platform was used to maintain body heat. The urine of mice was removed by applying gentle pressure on the bladder. Using a lubricated 24-gauge intravenous catheter (Braun) connected to a 1 ml syringe (BD) with the needle stylet removed, the mouse urethra was catherised. For bladder instillation: 100 μl of PBS (GIBCO) was instilled into the bladder and removed to wash the lumen. In total, 100 μl 0.25% trypsin (Sigma) was instilled in the lumen for a maximum of 30 min followed by three PBS washes. Subsequently, 100 μl of $5 \times 10^6$ UMUC3 cells stably transfected with pGL4.51 luciferase 2/CMV vector (Promega) using Lipofectamine 2000 (Invitrogen) and selected by 1 mg/ml neomycin (Sigma) were instilled into the bladder and the catheter and syringe left in place for 3 h. For the first experiment (A83-01) 50 mice were inoculated with 10 mice forming tumours. Mice were then separated into vehicle control ($n = 5$) or treated with A83-01 ($n = 5$) in gavage solution (DMSO:Saline, 3:2, v/v) (80 mg/kg daily for 2 weeks). Similarly, in the second experiment (LY2157299) 45 mice were inoculated with six mice forming tumours. Mice were then separated into vehicle control ($n = 3$) or treated with LY2157299 ($n = 3$) in gavage solution (0.5% hydroxyl propylnmethyl-cellulose + 0.2% Tween 80) (80 mg/kg daily for two weeks). For bladder wall injection: 0.5 cm slit was made on the abdomen of mice under anaesthesia to expose the bladder. The mice were catherised with PBS to maintain the bladder at about 85% full. In all, 10 μl of $1 \times 10^6$ T24-luc2 cells prepared as described for UMUC3 were injected into the bladder wall using a 31-gauge insulin syringe (BD) under stereoscope (Leica) with × 3.5 magnification and the slit was sutured after procedure. Tumour implantation was assessed by bioluminescence imaging with IVIS Spectrum Imaging System 200 (Xenogen). Imaging was performed 10 min after an intraperitoneal injection of 150 mg/kg firefly D-Luciferin in 200 μl of PBS. Signal intensity was quantified in photons per second per region of interest. Respective drugs were administered when bioluminescence intensity reached ~2e⁸ photons/sec. A83-01 administered in gavage solution (DMSO: Saline, 3:2, v/v) (80 mg/kg daily for two weeks). LY2157299 administered in gavage solution (0.5% hydroxyl propyl methyl-cellulose + 0.2% Tween 80) (80 mg/kg daily for 2 weeks).

PD0325901 administered in gavage solution (1% polysorbate 80) (25 mg/kg daily for 2 weeks).

**Transwell migration assay**. For transwell migration assay, UMUC3 cells or T24 cells were grown in 10 cm dishes to 80% confluency and serum-starved for 24 h. In all, $5 \times 10^5$ cells were suspended in 1% FBS supplemented DMEM in the absence or presence of JNJ38877605 and seeded on the upper chambers. The lower chambers were filled with 5% FBS supplemented DMEM in the presence or absence of HGF. Migrated cells were stained after 16 h. The cells were fixed in ice-cold methanol for 10 min and stained with crystal violet solution. The inner side of the upper chamber was wiped with cotton swabs. Migrated cells were then visualised through brightfield microscope and pictures were taken at four random sites and quantified.

**Soft agar**. The soft agar assay consisted of three layers. 0.6 mL bottom layer of 0.6% agar in 10% supplemented DMEM was dispensed into each well of a 24-well plate and allowed to solidify at room temperature for half hour. UMUC3 cells with or without transient transfection were trypsinized and counted. 0.5 mL middle layer of 0.36% agar and $5 \times 10^5$ UMUC3 carcinoma cells in 10% supplemented RPMI was plated above the bottom layer and allowed to solidify at room temperature for half hour. The top layer consisted of 0.5 mL/well of 10% FBS supplemented DMEM in the presence or absence of small molecular weight compounds and incubated at 37 °C in a humidified incubator with 5% CO₂ for 2 weeks and stained with MTT (Abcam). The plates were scanned and images from three replicates were quantified.

**Luciferase assays**. Luciferase assays were performed using the Dual luciferase system (Promega). CAGA-luciferase vector (300 ng) was transfected in the presence of SMURF2 (1 μg) or relevant mutants, and CMV-Renilla (0.25 μg). After 72 h TGF-β1 (2.5 ng/ml) was added in DMEM (0% FCS) and luciferase counts were measured 12 h later using a Sirius Luminometer (Berthold).

**RT2 profiler**. The expression profile of 84 TGFβ pathway-related genes was determined using a 96-well format TGFβ signalling pathway RT² Profiler PCR array (Qiagen-PARN-035) according to the manufacturer's instructions. The array also included six housekeeping genes and three RNA as internal controls. qPCR were run on an ABI 7900HT qPCR instrument (Applied Biosystems, UK) equipped with SDS 2.3 software, using RT² SYBR Green/ROX qPCR master mix. Data analysis was done by the $2^{-\Delta\Delta Ct}$ method on the manufacturer's Web portal. http://pcrdataanalysis.sabiosciences.com/pcr/arrayanalysis.php.

**Quantitative real-time PCR**. Cells were collected, washed twice in PBS and RNA was isolated using GeneJet RNA extraction kit (Thermo Scientific) qRT was performed using Taqman probes (USP26) from Applied Biosystems according to manufacturer's recommendations. Reactions were carried out on a ABI 7900 or 7500 FAST sequence detector (Perkin Elmer). Relative mRNA values are calculated by the ΔΔCt method. GAPDH or 18 S were used as internal normalisation controls where specified. The following QRT primers were used: PAI-1-5′-AAGG-CACCTCTGAGAACTTCA-3′, 5′-CCCAGGACTAGGCAGGTG-3′.

**Reporting summary**. Further information on research design is available in the Nature Research Reporting Summary linked to this article.

## Data availability

The molecular dynamics simulation trajectories and the coordinates files of the molecular models reported in the paper are available from the Figshare repository (https://figshare.com/articles/_/9044549). The NBT-II transcriptomic data are available from the GEO repository with the accession GSE104740. The mass spectrometry proteomics data have been deposited to the ProteomeXchange Consortium via the PRIDE [1] partner repository with the dataset identifier PXD014736. All other source data are available from the corresponding authors on reasonable request and/or are included with the manuscript (as figure source data or Supplementary Information).

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

## Acknowledgements

We appreciate the assistance in technical support and facility usage from the Histopathology Facility/Advanced Molecular Pathology Lab (AMPL), IMCB; and Dr. R. A. Jackson for editorial assistance. We thank Dr. Adelene Sim for her assistance during C2-HECT model building. This work was funded by research grants from Institute of Molecular and Cell Biology, Agency for Science, Technology and Research (A*STAR) core funding and Joint Council Office Development Programme 1234e00018, the National Research Foundation Singapore and the Singapore Ministry of Education under its Research Centers of Excellence initiative and the Ministry of Education Academic Research Fund Tier 1 grants (T1-2013 Sep-10) and (T1-2014 Oct-08). We thank A*STAR-BMRC for computing support. P.V.I. thanks the support from the European Union's Horizon 2020 research and innovation programme under the Marie Skłodowska-Curie Individual Fellowship for project number 786880. P.t.D is supported by Cancer Genomics Centre Netherlands.

## Author contributions

W.J.S., P.V.I., D.L., S.K.L.L., H.C.N., A.S., P.W.J., M.R., H.K.H., C.R., participated in experimental design, implementation and interpretation. D.K. performed the mass spectrometry work. D.L. and C.S.V., performed the dynamic modelling. P.t.D. supervised experiments performed at LUMC. A.P.K. supervised experiments performed at CSI. L.H.G., E.D., T.Z.T. performed the bioinformatics analysis. P.J.A.E. and J.P.T. conceived the project, W.J.S., P.V.I., P.J.A.E. and J.P.T. interpreted the results and wrote the paper.

## Additional information

**Competing interests:** C.S.V. is the founder of Sinopsee Therapeutics, a biotech company developing molecules for therapeutic purposes; the current work has no conflict with the company. J.P.T is a consultant/advisory board member for Aim Biotech Singapore, ACT Genomic Taipei, Biosyngen PTE ltd and CSO of BioCheetah Ltd Singapore. The authors declare no further conflicts of interest.

