## [Peer Review File · Nature Communications]

Editorial Note: Parts of this Peer Review File have been redacted as indicated to remove third-party material where no permission to publish could be obtained. This information is available upon request.

Reviewers' comments:

Reviewer #1, Expertise: Tgfbeta (Remarks to the Author):

The authors here explore the role of HGF- Met signaling in driving a muscle invasive phenotype in bladder cancer. They report that HGF induces TGFbeta signaling that in turn mediates epithelial-mesenchymal transition. They then go on to show that HGF regulates EMT and enhanced motility via TGFbeta-Smad signaling. At the mechanistic level they show this is mediated by HGF-induced Src activation that functions to inhibit Smad7-Smurf2 assembly and Smurf2 catalytic activity, thereby promoting TGFbeta receptor signaling. Overall, this is a very compelling study that describes a novel way in which HGF induces epithelial cell scattering, EMT and cell motility. Also, of note the mechanistic studies linking Src-dependent phosphorylation to Smurf2 functions are particularly elegant. Overall, I think a an exciting study that would be substantially improved by addressing the following issues:

1) The cell line used in these studies is an odd choice as it is derived from a chemically induced rat bladder cancer. It's unusual for a high grade bladder cancer to display a epithelial phenotype as these cells do. Also, the entire model is based on NBT-II. It is important to test key aspects of the model in different cell lines, in particular the key lines pursued in Figure 6.

2) The pathways proposed by the authors focus on TGFbeta. But based on the close similarity of Activin signaling and the fact that Activin is expressed in these cells, it's entirely possible this is mediated by Activin signaling or a combination of the two. To allow the conclusion that HGF is signaling through TGFbeta specifically, the authors should test if activin receptors are expressed and if knockdown of ALK5 and/or ALK4 interferes with HGF-induced EMT and migration.

3) The interaction of Src with Smurf2 is intriguing. It is important to demonstrate this with endogenous proteins in NBT-II cells. Also they need to test if HGF stimulation leads to endogenous Smurf2 tyrosine phosphorylation and ideally at the mapped sites given that the authors have p-specific antibodies.

4) The structural modeling is very nice. In Fig. 4a, testing if Src suppression of auto-Ub is rescued by the Y314F mutant is an important control to include. In Fig. 4F the demonstration that Src interferes with Smad7-Smurf2 interaction is not convincing in particular as Smurf2 steady state levels drop upon Smad7 co-expression. The authors need to explore this more thoroughly by using catalytically dead Smurf2 which should be more stable, as well as comparing to a Y->F mutant which should resist Src-dependent antagonism.

5) Results at line 404 should be Fig. 4K I presume. Here the evidence that Y434E inhibits activity needs to be improved. Given the very strong effect of Y434F, the activities should be compared separately so that the conclusion that Y434E indeed suppresses Smurf2 activity can be better documented. To better solidify the model the authors could also test the C2 mutant or C2 deletion that fails to inhibit the HECT domain.

6) Figure 5A and 5B tests effects of Smurf2 mutants and Src on TBRI stability. Here the experimental design could be improved substantially because the experiment does not assess turnover of the TGFbeta heteromeric receptor which is what the Smurf2-Smad7 complex targets. It just looks at TBRI and so the reported effects are modest at best when compared to effects on Smad2 activation. Minimally the authors need to confirm that stability is indeed affected by testing Smurf2 catalytic mutants and showing that TBRI loss is blocked by proteasome/lysosome inhibitors.

7) The activation of Erk by expressing Smurf2 is quite nice. This would be substantially improved and provide convincing support for the model if the authors showed that the Smurf2 mutants regulated beta-TRCP turnover as predicted from the model.

8) The studies in the bladder cancer models are important but a lot of the data needs substantive improvement. In particular:

-correlation of Met expression is not convincing. There is almost no change in expression and only one cell line has epithelial character. Assessment using a more accurate method across all the cell lines or at least a good chunk of them is necessary to make this point.

-it was difficult to follow the rest of this section as the references to figures and panels was jumbled.

-the "enhanced" E-cadherin referred to for Fig. 6e was completely unsupported by the figure. IF to assess MET would be important to include to substantiate this conclusion.

9) The pharmacologic interventions in Figure 6 show loss of invasive phenotype by inhibiting TBRI. This is interesting but it's unclear any of this relates to the mechanisms proposed above and specifically with respect to HGF signaling. Does Met inhibition also inhibit invasive capacity and most importantly, does a Smurf2 Src phosphorylation site mutant also inhibit this phenotype. This would provide strong support that the proposed molecular mechanism is important for the in vivo invasive phenotype.

10) Similar concerns arise in Figure 7 where the combined treatment data provides impressive results but is it linked to the mechanisms proposed in the earlier part of the paper? Expressing Smurf2 mutants that affect TBR1 versus pERK would really increase the impact of these studies. For example, tumours harbouring Smurf2^{YY-EE} (which prevents Erk activation, but allows for Smad2 activation) would presumably be highly sensitive to just TBRI inhibitors.

Minor issues:

Use of standard "µm" for distances (eg Fig. 1) would be less confusing than µM which stands for micromolar.

The plotting of results of the p-SMAD2C studies in Fig. 2E are problematic as the treatment of the cells with A83-01 leads to significant loss of total SMAD2 in the nucleus, so the ratio plot of 2E doesn't tell us anything. Plotting nuclear pSMAD2 to total SMAD2 (n+c) is a more meaningful analysis. In Figure 2I, what does >9 h mean? This should be a specific timepoint based on the description of the experiment.

Reviewer #2, Expertise: HGF, metastasis (Remarks to the Author):

The authors have made the interesting and intriguing observation that HGF stimulation leads to the activation of the TGFBR pathway, including SMAD2 phosphorylation and early gene expression, leading to EMT. It is further shown that this is src dependent. Then it is shown that src phosphorylates smurf2 on Y314 and 434, leading to smurf2 inactivation and increased stability of TBRI.

How exactly HGF stimulates TGFBR pathway remains unclear. There are several missing experiments to really prove the implication of the discovered src-smurf2 pathway in HGF dependent EMT. Results shown in figures 3, 4 and 5 are not related to HGF. It is acceptable that a number of biochemical experiments were done through overexpression of constructs. However, key results should have been verified in the context of the HGF stimulation and where possible on the endogenous molecules. For example, does HGF stimulation increase smurf2 phosphorylation on Y314 and 434? Is this inhibited by src inhibitor and dominant negative form? Is HGF dependent EMT (in vitro assay) affected in cells transfected with smurf2 Y314/Y434 various mutant? Also, if the hypothesis presented here is valid, HGF should lead to an increased stabilization of TBRI and this should be inhibited by src inhibition. This has not been investigated. Control with c-met inhibition (or knock down) is missing in the in vitro and in vivo experiments using UM-UC3-3 cells

to prove that the EMT phenotype of these cells results from c-met activity and that the tumorigenesis is c-met dependent.

Although generally the experiments are of high quality, a number of experiments need repeats and statistical analyses (see specific comments).

Some of the experiments should be repeated in presence of c-Met pharmacological inhibition / siRNA knock down to verify the requirement of c-Met in this system, at least in Figure 2.

Specific comments

The EMT figures are not clear (S1A, S2B, S4A): The methods indicate that cells were grown for 4 days and then treated and imaged by time lapse for 24h. It is therefore assumed the IF pictures are taken at 5 days. How was the confluence of the cells at 4 days? The untreated cells appear very confluent on the pictures. In the case cells were already quite confluent at day 4, how can the cells be so sparse upon growth factor stimulation? Have some cells died? In the case the cells were subconfluent at day 4, then, the control cells have continued proliferating until confluence at day 5 while the cells treated with the growth factor did not proliferate but would have gone EMT instead? It would be good to clarify this.

Line 210: the statement "while unexpectedly decreasing the levels of pERK" is not correct according to Figure S2A. HGF does increase ERK1/2 phosphorylation, strongly, although the level goes down with time (but sustained activation is still seen).

Line 225: The requirement of ERK1/2 for SMAD2 phosphorylation should be assessed with MEK inhibitor treatment.

Line 240: specify what is A83-01

Line 421: clarify that it is co-transfection of SMURF2 with TBRI and define clearly what is TRBI.

Figure 1C: statistics (from at least 3 independent experiments) is missing.

Figure 2B: quantification from at least 3 independent experiments and statistical analyses should be provided.

Figure 2D: is this really laminin beta1 or rather lamin beta?

Figure 2E: statistics (from at least 3 independent experiments) is missing.

Figure 2G: quantification from at least 3 independent experiments and statistical analyses should be provided.

Figure S4A: EGF, HGF and IGF should appear more clearly on the figure. AZD0530 is more effective on EMT with increasing doses. What does this mean? Are doses of 2 and 8 microM still specific to src? Controls should be provided including siRNA src.

Figure 3B: quantification from at least 3 independent experiments and statistical analyses should be provided. The drugs (MEKI and SRCI) should have been added on cells + and - HGF, to clearly monitor the influence HGF dependent effect on the signalling analysed.

Figure 3C: can the co-IP be done with the endogenous proteins to show it is not an "artificial" interaction?

Figures 3K and L: quantification from at least 3 independent experiments and statistical analyses should be provided.

Figure 5A and 5B: quantification from at least 3 independent experiments and statistical analyses should be provided.

Figure 6A: pictures should be provided, at least for the cells shown in Figure 6C.

Figure 6C: the blot quality is not good. Cells should have been treated with a c-met inhibitor to control for the P-c-met band. Moreover, additional experiments to show that c-met inhibition / siRNA knock down reverts the EMT phenotype should have been done. Which c-met P-site was investigated?

Figure 6E: quantification from at least 3 independent experiments and statistical analyses should be provided. Confocal pictures should be provided.

Page 12 of the manuscript: there are errors in figure numbers. It is not Figure 6 but Figure 4

Reviewer #3, Expertise: Bladder cancer (Remarks to the Author):

The authors investigate the role of HGF in bladder cancer progression. The key finding is that HGF activates also TGF-B signaling pathway, and combined blockade of MEK and TGF-B pathways is necessary to inhibit HGF effects in bladder cancer.

The clinical context is poorly established. Reference 21 hardly justifies studying c-Met in bladder cancer. What do we know about c-Met from TCGA, for example? Better references include: PMID: 25816892. Also, the following paper suggests MET is down-regulated in aggressive bladder cancer: PMID: 24853099. How do the authors reconcile this? In the Results (Line 440) the authors state that c-Met correlates with tumor grade and poor prognosis– please provide reference for this.

How does HGF activate TGF-B pathway? There is a section heading “c-MET activation drives TBR-dependent SMAD2 phosphorylation” – but c-MET has not really been studied here. We know only that HGF induces TGFB signaling, but how the two pathways link is not determined. c-MET activation is even in the title, but the actual role of c-MET is not established. It is likely a correct assumption, but should be demonstrated. Assuming that it is indeed through c-MET, how is TGFB pathway activated by c-Met? This key question is not addressed.

There is no correlation in this paper to patient tumors - it is primarily in vitro work with some basic cell line xenograft work. There needs to be some link back to patient samples to provide clinical context.

There is no logic to the use of cell lines in this paper. One major underlying flaw is the use of a carcinogen (BBN) – induced rat bladder tumor cell line NBT-II without any mention that it is a rat model. This undermines the entire foundation of the HGF-MET signaling. This work needs to be carried out in a panel of human bladder cancer cell lines. UC3 and T24 are introduced later, but there is random use of one or the other, with no rhyme or reason. Fig 7 mixes UC3, T24 and NBT-II arbitrarily. In Figure 6 EMT is scored in a nice panel of cells lines, but it would be important to show c-Met expression and phosphorylation in a broader spectrum of these cell lines (not just the four selected lines in Fig 6b).

This paper is like two papers in one. After establishing a link between HGF and TGFB signaling, the investigators jump abruptly (Fig 3-5) into c-SRC signaling and SMURF2 without any logical transition. The rationale for these experiments needs to be clearer in the results (building on text in Introduction), and the transition needs to be smoother. The work on c-SRC and SMURF2 is elegant, innovative and valuable – but it does not appear to have anything to do with the rest of the paper, and especially nothing to do with the bladder cancer models. Can the investigators demonstrate relevance of SMURF2 in bladder cancer? The authors jump equally abruptly back to

HGF-induced EMT/invasion in bladder cancer (Figs 6 and 7) in the latter part of the paper, picking up the story started in Fig 1 and 2.

The investigators use A83-01 and LY2157299 to inhibit TGF β signaling – without justification of differences. Why one versus the other for different assays? They do not use one to validate the other, which would make sense, but instead interchange the two arbitrarily. In the middle of Fig 7 they switch between two TGF β inhibitors.

Figure 6:

- Fig 6a: Need to define in legend how EMT scores were measured.
- In Fig 6d, what happens to c-Met phosphorylation under same conditions?
- Fig 6F-G – This is not really IHC but rather H&E (same in Fig 7F).
- It is easy to find a location of muscle invasion in one tumor and an area of no invasion in another – this hardly establishes an effect of the inhibitors. Please include number of mouse tumors established for each condition and categorize T-stage for each mouse (ie Ta/T1/T2/T3 etc). Comment if there were any nodal metastases. In Fig 6G one can see the entire bladder which is more convincing – in F one sees only part of interface with bladder wall. It is unclear how the investigators determined in lower right panel of F that the tumor is separate from the urothelium. The black line appears to be drawn arbitrarily through tumor.
- The difference in pSMAD2 in panel H is subtle – much less than the summary numbers indicated in bar graph below (looks like 80-90% positive staining).
- Were results with both inhibitors (bar graphs for F and G) really identical?

In Fig 6 (lines 460-462) the authors claim: “Western blot analysis in UM-UC-3 cells indicates that loss of TBR signaling through treatment with A83-01 results in enhanced E-cadherin expression indicating that TBR suppression reverts UM-UC-3 cells to a more epithelial state (Fig. 6e)” - but there is no E-cadherin expression in either condition on this Western blot. If true, the authors should show also in the xenografts. Without either, a main premise of the authors is undermined.

In Fig 7F in the LY panel, the blow up should be of the middle of the tumor front where invasion would be more likely – and not way off at the edge of the tumor.

Fig 7G: “tumour invasion” implies depth of invasion (lamina propria, muscularis propria, etc), but the variables entered are G1, G3 and G4. This implies “grade”. But it would be unusual for the grade of a tumor to change with drug therapy – especially the drastic changes described here. Also, bladder cancer is usually graded G1, G2 and G3 – there is no grade 4. The images in F do not reflect any difference in grade. Why is there a single tumor size for each condition – only a single mouse?

The investigators jump around between intravesical and subcutaneous mouse models. Why use any subcutaneous model if able to do orthotopic? In 7F the authors state T24 injected into the bladder wall of nude mice – but this implies intramural and not intraluminal inoculation. Which was used? Only the intravesical inoculation of bladder cancer cells is described in the methods for the orthotopic model. This is a very tricky model and it is hard to get good tumor take. It is surprising that the authors have made this work for both T24 and UC3. Have the authors verified that both cell lines are genotypically what they are supposed to be? This is not indicated in the methods – T24 is actually not even listed in the methods. Please indicate how many mice were inoculated, what proportion had tumors, and please show growth curves over time. The main advantage of this model is that drugs can be administered intravesically. Here they are administered systemically, so intramural injection would have been a more robust model.

The main endpoint reported in the mouse model is depth of bladder wall invasion. The authors show sample images, but do not quantify this adequately. How many were muscle invasive? What about pT1 or pT3? Showing only part of the bladder wall can also be deceiving. Invasion into lamina propria is still early and not late stage bladder cancer, as the authors suggest. Please show an example of lymphovascular invasion – was this determined with IHC for a vessel wall marker

(e.g. CD31) or just H&E?

Figs 6 and 7: The legend says that tumor volume was measured twice weekly for the T24 orthotopic xenografts – but these results are not shown. Since this is a luciferase-transduced cell line, presumably the authors mean that bioluminescence was measured. Please show the growth curves rather than sample mice, which can be quite deceiving. It is again critical to show how many mice were used.

The investigators focus on EMT – yet show no in vitro correlates for invasion, and nothing related to metastasis.

The authors emphasize the clinical relevance of their orthotopic bladder cancer model. They should remove this text. This model is not clinically relevant. It conforms to the “seed and soil” hypothesis, so it is biologically relevant, but it is far from clinically relevant.

Would it not be useful to knockout SMAD2 to show its importance in HGF effects?

Minor points:

Figure 2: There is not enough evidence to support the conclusion that SMAD2L but not SMAD2C is likely to be mediated by ERK. The differences are subtle.

Fig 2E: A83-01 reverted only a small proportion of the pSMAD2 activation.

Figure 3 – Please indicate what pY is in legend.

Fig 7a-c: There is no convincing evidence here that TGF β inhibitor leads to increased ERK phosphorylation in either cell line.

Line 458-490: Note errors in this section referring to Fig 3 rather than Fig 6.

Fig 7: Why are tumors decreased in size with inhibitors if same or increased proliferation?

Reviewer #4, Expertise: molecular modelling and dynamics (Remarks to the Author):

Sim et al. study HGF/c-MET regulation with a focus on the influence of phosphorylation of SMURF2 at tyrosine residues and its influence on SMAD7 binding and how it enhances the interaction of SMURF2 and the C2-HECT domain. The authors use molecular docking and MD simulations to study the influence of Tyr phosphorylation in a WW3 domain of SMURF2 to bind a SMAD7 peptide. The authors use molecular docking to build a model of the HECT:C2 domain in complex with SMURF2. Since my expertise is in the area of molecular modeling I will entirely focus on this part of the manuscript:

The molecular modeling is intended to support the experimental results and to provide a possible molecular view of the proposed mechanism.

Both the molecular docking and MD-simulations are well described and follow established protocols. The results of the simulations are well documented and support the experimental data. The inclusion of experimental data in the docking protocol greatly improves the quality of the prediction. I think the whole modeling section overall improves the quality of the paper.

My only concern and request is to emphasize that a docked model is a model and not necessarily a “real” structure. This could be discussed a bit further in the MS. Hence, the authors may want to indicate in the text (e.g. discussion) that the model suggests a possible binding mechanism that is not necessarily the only possible binding geometry.

c-Met activation leads to the establishment of a TGF β receptor regulatory network required for bladder cancer invasion.

Responses to reviewer's comments:

We thank the reviewers for the thoughtful and thorough review of the manuscript. We have revised the manuscript according to the reviewers suggestions and have addressed all the concerns brought forward by the reviewers. Thanks to their insights and comments our manuscript has greatly improved. We are glad to report that we have addressed the majority of concerns and comments raised by the reviewers. We have now **66** new figure panels and tables (**Figs. 2D, 2F, 3D, 4G, 5E, 5F, 5G, 6A, 6B, 6E, 6G, 7A, 7B, 8A, 8B, 8C, 8D, 8E, 8F, 8G, 8H, S1E, S2A, S2E, S2F, S5C, S5D, S5E, S5F, S7E, S8A, S8B, S8C, S9A, S9C, S9D, S9E, S9F, S10A, S10B, S10C, S10D, S10E, S10F, S10G, S10H, S11A, S11B, S11C, S11D, S11E, S12A, S12B, S12C, S12D, S12E, S12F, S14B, S14C, S14D, S14E, S14G, S14H, S14I, S14J and new Table 4.**

Reviewer #1. Overall, this is a very compelling study that describes a novel way in which HGF induces epithelial cell scattering, EMT and cell motility. Also, of note the mechanistic studies linking Src-dependent phosphorylation to Smurf2 functions are particularly elegant. Overall, I think a an exciting study...

We thank the reviewer for his/her kind comments.

Reviewer #1. The cell line used in these studies is an odd choice as it is derived from a chemically induced rat bladder cancer. It's unusual for a high grade bladder cancer to display a epithelial phenotype as these cells do. Also, the entire model is based on NBT-II. It is important to test key aspects of the model in different cell lines, in particular the key lines pursued in Figure 6.

We thank the reviewer for their excellent comments. As part of our original screening process we decided to use NBT-II cells as they reflect the nearly full spectrum of EMT transition. This would allow us to accurately analyse cell plasticity with transcriptional responses. Out of all the cell lines we have tested only NBT-II cells display this spectrum. We now indicate in the text that NBT-II cells are derived from a chemically-induced rat bladder carcinoma. Following our mechanistic analysis of HGF in NBT-II cells we subsequently analysed a panel of bladder cancer cell lines against our previously established EMT signature. Our original reasoning for using the following cell lines is as follows; as correctly pointed out by the reviewers it is highly unusual that high grade bladder cancer displays an epithelial phenotype, for this reason we choose to focus on cell lines which displayed the highest EMT score (most mesenchymal) specifically T24, J82, and UMUC3 to accurately reflect what is observed in clinical settings. In addition, we tested if these cell lines would form tumours following either intramural or intraluminal injection. As discussed in detail below we observed that while UMUC3 cells were able to form tumours through intraluminal injection, T24 were not able to form tumours through this method. Next, we wanted to explore the observed effects of TGF β Receptor inhibitor treatment in a more clinically relevant setting.

To do so we wanted to inject bladder cancer cells into the bladder wall and permit tumour formation, similar to what is observed in the clinic. We initially attempted this with UMUC3 cells but due to the aggressive nature of this cell line the mice died within days hindering our ability to accurately assess tumour formation following inhibitor treatment. To perform these experiments accurately we utilised the less aggressive cell line T24.

In relation to the reviewers comments above we have now also further analysed the role of HGF-induced EMT in bladder cancer. The following experiments were performed and are discussed in order to the reviewers comments below:

- Analysis of p-c-Met in T24, J82, and UMUC3
- Analysis of MET expression in our panel of bladder cancer cell lines.
- Utilisation of the c-MET inhibitor JNJ38877605 to determine the role of c-MET in T24, J82, and UMUC3.
- IF analysis of E-cadherin following HGF treatment in the absence or presence of JNJ38877605 in UMUC3 cells
- Confirmation of the feedback loops following the addition of the MEK inhibitor PD0325901 and the TGF β Receptor inhibitor LY2157299 in T24 cells.
- Validation of intercellular signalling following c- SRC inhibitors in T24, J82 and UMUC3 cells.
- Analysis of TGF β Receptor stability following HGF treatment in the presence or absence of a c-SRC inhibitor in UMUC3 cells.
- Invasion assay utilising UMUC3 and T24 cells.
- Anchorage independent assay in UMUC3 cells.

Reviewer #1. *The pathways proposed by the authors focus on TGFbeta. But based on the close similarity of Activin signaling and the fact that Activin is expressed in these cells, it's entirely possible this is mediated by Activin signaling or a combination of the two. To allow the conclusion that HGF is signaling through TGFbeta specifically, the authors should test if activin receptors are expressed and if knockdown of ALK5 and/or ALK4 interferes with HGF-induced EMT and migration.*

We thank the reviewer for his/her suggestion. To delineate the effects of the ALK4 and ALK5 in HGF induced TGF β Receptor signalling we have now performed 2 individual experiments using activin ligand trap ACVR2B-Fc and pan TGF β 1,2,3 neutralizing antibodies or utilised ectopic expression of kinase dead mutants of ALK4 and ALK5. We observe that while inhibition of the Activin receptor with either a blocking antibody or ectopic expression of ALK4 catalytically inactive mutant minimally decreases HGF induced SMAD2 phosphorylation, inhibition of ALK5 with either a blocking antibody to TGF β or ectopic expression of ALK5 mutant nearly completely abolishes this effect. These results suggest that TGF β /ALK5 signalling is the primary target of HGF induced SMURF2 downregulation and resulting in rapid upregulation of the pSMAD2. Nevertheless ALK4 does appear to play a minimal role in HGF induced pSMAD2 as well. It is important to note that during these studies we utilized both the TGF β Receptor inhibitors A83-01 and LY2157299. A83-01 targets ALK4, ALK5, and ALK7 at IC₅₀ values are 12, 45 and 7.5 nM respectively. Notably in figure 3B we note that A83-01 was more effective at downregulating HGF induced pSMAD2 than LY2157299 when treated at similar concentrations, once again suggesting a partial role for ALK4 in our observed effects. (**New figure S2E and S2F**).

Reviewer #1. *The interaction of Src with Smurf2 is intriguing. It is important to demonstrate this with endogenous proteins in NBT-II cells.*

We thank the reviewer for his/her suggestion. We have performed these experiments in both NBT-II and 293T cells. All experiments were performed at endogenous levels. (New figure S5E and S5F).

Reviewer #1. *Also they need to test if HGF stimulation leads to endogenous Smurf2 tyrosine phosphorylation and ideally at the mapped sites given that the authors have p-specific antibodies.*

We thank the reviewer for his/her excellent comment. We attempted to analyse endogenous tyrosine phosphorylation of SMURF2 following HGF treatment. As observed below we noted an increase in Y314 phosphorylation following HGF treatment at 60 and 90 minutes. However, we were unable to reproduce these results and do not feel these results are conclusive enough to warrant publication. However, we now demonstrate the induction of the c-SRC/SMURF2 complex formation following HGF treatment at endogenous levels. Indicating that c-SRC potentially phosphorylates SMURF2 following HGF treatment. (New figure 3D).

Reviewer #1. *The structural modeling is very nice.*

We thank the reviewer for his/her kind comments.

Reviewer #1. *In Fig. 4a, testing if Src suppression of auto-Ub is rescued by the Y314F mutant is an important control to include.*

We thank the reviewer for his/her comment. Similar to figure 4A we now include new data indicating that mutation of both tyrosine sites to phenylalanine (FF) completely inhibits the ability of c-SRC to downregulate SMURF2 activity. (New figure 5G).

Reviewer #1. In Fig. 4F the demonstration that Src interferes with Smad7-Smurf2 interaction is not convincing in particular as Smurf2 steady state levels drop upon Smad7 co-expression. The authors need to explore this more thoroughly by using catalytically dead Smurf2 which should be more stable, as well as comparing to a Y->F mutant which should resist Src-dependent antagonism.

We thank the reviewer for this excellent comment. We agree. However, we note that as equal levels of SMURF2 was immunoprecipitated and c-SRC had no effects on SMAD7 levels we feel that this demonstrates that c-SRC phosphorylation of SMURF2 inhibits the ability of SMAD7 to bind to SMURF2. Furthermore, as importantly suggested by the reviewer we have now performed the experiment with a catalytically inactive version of SMURF2(C/A) and the “dominant active” mutant SMURF2 YY314-434FF (FF). As expected co-expression of c-SRC inhibited the ability of SMAD7 to bind to catalytically inactive SMURF2 but not in the presence of the FF mutant.(New figure 4G)

Reviewer #1. Results at line 404 should be Fig. 4K I presume.

This has now been corrected.

Reviewer #1. Here the evidence that Y434E inhibits activity needs to be improved. Given the very strong effect of Y434F, the activities should be compared separately so that the conclusion that Y434E indeed suppresses Smurf2 activity can be better documented. To better solidify the model the authors could also test the C2 mutant or C2 deletion that fails to inhibit the HECT domain.

We agree. We now present new data in separate figures highlighting the effect of SMURF2 glutamic acid mutations and phenylalanine mutations on SMURF2 activity. As can be seen in the left panel below mutation of tyrosine 314 and 434 to glutamic acid which mimics the phosphorylation by c-SRC completely abolished SMURF2 catalytic activity. In contrast mutation of tyrosine 434 to phenylalanine, which blocks the ability of c-SRC to phosphorylate SMURF2, enhanced the autocatalytic activity of SMURF2 to levels similar with a dominant active mutant FF29/300AA (FF29/300AA inhibits the ability of complex binding between C2 and HECT domains rendering SMURF2 functionally active). Curiously, mutation of Y314 to a phenylalanine did not increase the activity of SMURF2 potentially due

to levels of SMAD7 in the cells or the continued presence of the Y434 wildtype residue. (New figures 5E and 5F).

Reviewer #1. Figure 5A and 5B tests effects of Smurf2 mutants and Src on TBRI stability. Here the experimental design could be improved substantially because the experiment does not assess turnover of the TGFβ heteromeric receptor which is what the Smurf2-Smad7 complex targets. It just looks at TBRI and so the reported effects are modest at best when compared to effects on Smad2 activation. Minimally the authors need to confirm that stability is indeed affected by testing Smurf2 catalytic mutants and showing that TBRI loss is blocked by proteasome/lysosome inhibitors.

We thank the reviewer for his/her comment. We have now performed a new experiment analysing the degradation of the TGFβ receptor with both SMURF2 and dominant active SMURF2, SMURF2(FF) in the presence or absence of MG132. As observed treatment with the proteasome inhibitor completely inhibited the degradation of the TGFβ receptor by both isoforms of SMURF2. (New figure 6E)

Reviewer #1. The activation of Erk by expressing Smurf2 is quite nice. This would be substantially improved and provide convincing support for the model if the authors showed that the Smurf2 mutants regulated beta-TRCP turnover as predicted from the model.

We agree with the reviewer. We now demonstrate that both SMURF2 and SMURF2(FF) decrease the overall endogenous levels β -TRCP. An effect annulled in cells transfected with a dominant negative isoform of SMURF2 (EE). (New Figure 6G).

Reviewer #1. The studies in the bladder cancer models are important but a lot of the data needs substantive improvement. In particular:

-correlation of Met expression is not convincing. There is almost no change in expression and only one cell line has epithelial character. Assessment using a more accurate method across all the cell lines or at least a good chunk of them is necessary to make this point.

We agree. To this end we analysed the RNA datasets of the bladder cancer cell lines (**new Table 4**). Furthermore, as correctly pointed out by the reviewer earlier it is highly unusual that high grade bladder cancer displays an epithelial phenotype, for this reason we choose to focus on cell lines which displayed the highest EMT score (most mesenchymal) specifically T24, J82, and UMUC3. Utilising these cell lines we now demonstrate that UMUC3 has the highest c-MET activity in absence of HGF treatment. Importantly, T24 has no detectable c-MET activity in the absence of HGF exposure. All three cells displayed increase c-MET phosphorylation following HGF addition, an effect annulled in cell lines co-treated with the HGF inhibitor JNJ38877605. (New figure S9A)

Reviewer #1. it was difficult to follow the rest of this section as the references to figures and panels was jumbled.

We apologise for the confusion on our part. We have now corrected these errors.

Reviewer #1. the “enhanced” E-cadherin referred to for Fig. 6e was completely unsupported by the figure.

To further analyse the role of HGF signalling in our cells we now demonstrate that HGF addition decreases E-cadherin staining in UMUC3 cells. Furthermore, we demonstrate that the addition of the c-MET inhibitor JNJ38877605 significantly increases E-cadherin staining. Notably, we now also demonstrate by IF that treatment of UMUC3 cells with the TGF β inhibitor A83-01 enhances the levels of cortical actin. Taken together these results suggest that inhibition of HGF mediated TGF β signalling utilising either a HGF inhibitor or a TGF β inhibitor switches UMUC3 cells to a more epithelial phenotype. (New figure S10G, S10H)

Reviewer #1. The pharmacologic interventions in Figure 6 show loss of invasive phenotype by inhibiting TBRI. This is interesting but it's unclear any of this relates to the mechanisms proposed above and specifically with respect to HGF signaling. Does Met inhibition also inhibit invasive capacity and most importantly, does a Smurf2 Src phosphorylation site mutant also inhibit this phenotype. This would provide strong support that the proposed molecular mechanism is important for the *in vivo* invasive phenotype.

We thank the reviewer for this excellent comment. We now demonstrate in UMUC3 and T24 cells that HGF enhances the invasive capacity of these cell lines while treatment with the MET inhibitor JNJ38877605 significantly inhibits this effect. Furthermore we show that treatment of UMUC3 cells with JNJ38877605 diminishes anchorage independent growth of these cells. Moreover, we now show in UMUC3 and T24 cells that ectopic expression of SMURF2(EE) (which enhances TGF β signalling) increases the invasive capacity of these cell lines in the absence or presence of HGF, while SMURF2(FF) (which decreases TGF β signalling) abolishes this effect. (New figures S10A, S10B, S10C, S10D, S10E S10F, S11A, S11B, S11C, S11D).

Reviewer #1. Similar concerns arise in Figure 7 where the combined treatment data provides impressive results but is it linked to the mechanisms proposed in the earlier part of the paper? Expressing Smurf2 mutants that affect TBR1 versus pERK would really increase the impact of these studies. For example, tumours harbouring Smurf2YY-EE (which prevents Erk activation, but allows for Smad2 activation) would presumably be highly sensitive to just TBRI inhibitors.

We now demonstrate that ectopic expression of UMUC3 cells with SMURF2(FF) (which enhances MAPK activation but downregulates TGF β signalling) increases the sensitivity of cells to the MEK inhibitor PD0325901 as measured by growth in soft agar. In contrast, ectopic expression of UMUC3 cells with SMURF2(EE) (which enhances TGF β activation but downregulates MAPK activation) increases the sensitivity to the TGF β inhibitor LY2157299. Notably, expression of SMURF2(EE) or SMURF2(FF) decreased the ability of UMUC3 cells to grow in soft agar highlighting the importance of both the TGF β and MAPK pathways in this regard. Furthermore we demonstrate that SMURF2(FF) mediated pERK signalling is sensitive to MEK inhibition while SMURF2(EE) mediated pSMAD2 is sensitive to TGF β receptor inhibition (New figures S12A, S12B, S12C, S12D, S12E, S12F).

Reviewer #1. Use of standard “ μm ” for distances (eg Fig. 1) would be less confusing than μM which stands for micromolar.

This has now been corrected.

Reviewer #1. The plotting of results of the p-SMAD2C studies in Fig. 2E are problematic as the treatment of the cells with A83-01 leads to significant loss of total SMAD2 in the nucleus, so the ratio plot of 2E doesn't tell us anything. Plotting nuclear pSMAD2 to total SMAD2 (n+c) is a more meaningful analysis.

We agree. We have now replotted the data from figure 2E to incorporate total SMAD2 as a

combination of total nuclear and cytoplasmic SMAD2 (New Figure 2F).

Reviewer #1 In Figure 2I, what be a specific timepoint based on experiment.

does 9 h mean? This should the description of the

We thank the reviewer for identifying this critical error. We have now corrected this mistake.

Reviewer #2. The authors have made the interesting and intriguing observation that HGF stimulation leads to the activation of the TGFBR pathway, including SMAD2 phosphorylation and early gene expression, leading to EMT.

We thank the reviewer for his/her critical reading of our manuscript and for finding our work interesting.

Reviewer #2. does HGF stimulation increase smurf2 phosphorylation on Y314 and 434?

Please see comments above.

Reviewer #2. Is HGF dependent EMT (in vitro assay) affected in cells transfected with smurf2 Y314/Y434 various mutant?

We have now analysed the effects of SMURF2 mutants on EMT. We note that ectopic expression of SMURF2(FF), which would enhance SMURF2 activity leading to TGFB receptor degradation, enhances β -catenin expression at cell-cell contacts and decreases overall Vimentin expression compared to either SMURF2 WT or SMURF2(EE) expression. SMURF2(EE) expression did not significantly enhance mesenchymal markers probably due to the mesenchymal nature of these cells. (New Figure S11E).

Reviewer #2. Also, if the hypothesis presented here is valid, HGF should lead to an increased stabilization of TBRI and this should be inhibited by src inhibition. This has not been investigated.

We thank the reviewer for his/ her comment. To demonstrate the role in TGF β receptor stability we biotin labelled cells in the presence of increasing concentrations of HGF. As can be seen in new figure 6A, HGF enhances plasma localisation of TGF β receptor. Next we analysed endogenous levels of TGF β receptor in the presence of HGF and c-SRC inhibitor AZD0530.

Once again we demonstrated that HGF enhances the stability of TBRI an effect which is inhibited following the addition of AZD0530. Similar results were observed in UMUC3 cells. (New figure 6A, 6B, S7F).

Reviewer #2. Control with c-met inhibition (or knock down) is missing in the in vitro and in vivo experiments using UM-UC3-3 cells to prove that the EMT phenotype of these cells results from c-met activity and that the tumorigenesis is c-met dependent.

Please refer to reviewer 1 comments above related to figures S9A, S10A, S10B, S10C, S10D S10E, S10F, S10G.

Reviewer #2. Some of the experiments should be repeated in presence of c-Met pharmacological inhibition / siRNA knock down to verify the requirement of c-Met in this system, at least in Figure 2.

We agree. We now demonstrate that chemical inhibition of c-MET using the c-MET inhibitor JNJ38877605 reverts HGF induced EMT in NBT-II cells and diminishes HGF induced pSMAD2 at both the c-terminal and linker phosphorylation sites. (New figures 2D and S1E)

Reviewer #2. The EMT figures are not clear (S1A, S2B, S4A): The methods indicate that cells were grown for 4 days and then treated and imaged by time lapse for 24h. It is therefore assumed the IF pictures are taken at 5 days. How was the confluence of the cells at 4 days? The untreated cells appear very confluent on the pictures. In the case cells were already quite confluent at day 4, how can the cells be so sparse upon growth factor stimulation? Have some cells died? In the case the cells were subconfluent at day 4, then, the control cells have continued proliferating until confluence at day 5 while the cells treated with the growth factor did not proliferate but would have gone EMT instead? It would be good to clarify this.

We thank the reviewer for his/her comment and apologise for the confusion in our figures. As originally stated in our materials and methods NBT-II cells were plated onto a 12 well plate at a low density of 200 cells per well in 1 ml of medium. Cells were allowed to grow and form epithelial colonies for a period of 4 days. The cultures were then refreshed with medium

containing the desired treatment. Time-lapse video microscopy was used to analyse individual cell colonies. It is important to note here that HGF induces cell dispersion as can be partially observed in figures 1A, S1E, S5C, S5D. We also invite the reviewer to look up our previous work on HGF induced cell dispersion (Bellusci et al. Journal of Cell Science 1994 107: 1277-1287). Please note HGF was previously referred to as scatter factor-like factor.

Reviewer #2. Line 210: the statement “while unexpectedly decreasing the levels of pERK” is not correct according to Figure S2A. HGF does increase ERK1/2 phosphorylation, strongly, although the level goes down with time (but sustained activation is still seen).

In figure 2A NBT-II cells were treated for 2 hours with HGF and protein lysate analysis was performed using a phospho-specific antibody microarray. Using this methodology a partial decrease in phosphorylated ERK levels was observed compared to relevant controls. In figure 2C we observe an expected upregulation pERK at earlier time points with a compensatory downregulation at later time points following HGF treatment. Why we observe a slight difference between these two methodologies in analysing pERK levels may be due to detection quality of the microarray. We have now changed the sentence accordingly describing the observed results from the microarray panel, “while unexpectedly levels of pERK were moderately down compared to controls at 2 hours post HGF treatment”

Reviewer #2. Line 225: The requirement of ERK1/2 for SMAD2 phosphorylation should be assessed with MEK inhibitor treatment.

This was originally performed in NBT-II cells (figure 3B) where we demonstrate that co-treatment with MEK inhibitors PD325901 and MEK162 inhibited HGF induced SMAD2 linker phosphorylation. We now also demonstrate this in T24, J82, and UMUC3 cells in **new figures S9C, S9D, S9E and 8F**. Note that SMAD2 linker phosphorylation can also occur through CDKs and JNK.

Reviewer #2. Line 240: specify what is A83-01

We agree and have now included this following sentence: “A83-01 is a small molecule inhibitor specifically targeting T β RI(ALK-5), ALK-4 , and ALK-7 the three of which contain highly structurally related kinase domains. “

Reviewer #2. Line 421: clarify that it is co-transfection of SMURF2 with TBRI and define clearly what is TBRI.

We have now rewritten this sentence accordingly, "Correspondingly, co-transfection of wild-type SMURF2 significantly decreased TBRI stability...". TBRI was previously defined in the introduction.

Reviewer #2. Figure 1C: statistics (from at least 3 independent experiments) is missing.

We apologise for the confusion and the lack of information given on our part. The data shown are representative of three independent and reproducible experiments. This has now been stated in the figure legend and relevant statistics have now been included. **(Revised figure 1C)**

Reviewer #2. Figure 2B: quantification from at least 3 independent experiments and statistical analyses should be provided.

This has now been done. (New figure S2A).

Reviewer #2. Figure 2D: is this really laminin beta1 or rather lamin beta?

Laminin B1 as suggested by the manufacturers' protocol.

Reviewer #2. Figure 2E: statistics (from at least 3 independent experiments) is missing.

We apologise for the confusion and the lack of information given on our part. In our original experiment the bars represent mean \pm SD of three independent experiments. Due to the suggestion of reviewer 1 (see above) we have altered the graph to depict nuclear pSMAD2 to total SMAD2 (nuclear + cytoplasmic). **Now figure 2F**. As per reviewers suggestion we now include details pertaining to the number of times the experiment was performed and relevant statistics.

Reviewer #2. *Figure 2G: quantification from at least 3 independent experiments and statistical analyses should be provided.*

We thank the reviewer for his/her comment. **Figure 2G now Sup. Figure 2C** contain representative images from two independent experiments taken at 0 hours and 24 hour time points. These figures were extrapolated from a bright field microscope image and not from the Axiovert-200M live image microscope which tracks individual cellular displacement. Therefore, we feel any statistical significance given to this experiment would be subjective. Nevertheless, we believe that the figure accurately displays our conclusions that knockdown of SMAD2 diminishes HGF-mediated cellular dispersion.

Reviewer #2. *Figure S4A: EGF, HGF and IGF should appear more clearly on the figure. AZD0530 is more effective on EMT with increasing doses. What does this mean? Are doses of 2 and 8 micromM still specific to src?*

We thank the reviewer for his/her comment. We have now edited the figure to more clearly display the HGF, EGF, IGF lanes. AZD0530 is a pan SRC family kinase inhibitor targeting c-SRC(2.7nM), LCK- YES (4nM), and Lyn(5nM) in cell free assays. As both c-SRC and LCK target SMURF2 for phosphorylation it is unsurprising that higher concentrations of AZD0530 decrease the epithelial mesenchymal transition induced by HGF.

Reviewer #2 *Controls should be provided including siRNA src.*

We now demonstrate that genetic inhibition of c-SRC utilising siRNA completely abolishes HGF induced EMT, similar to chemical inhibition with AZD0530. (New figure S5C, S5D)

Reviewer #2. *Figure 3B: quantification from at least 3 independent experiments and statistical analyses should be provided. The drugs (MEKI and SRCI) should have been added on cells + and – HGF, to clearly monitor the influence HGF dependent effect on the*

signalling analysed.

We thank the reviewer for his/her comment. We have now included quantification values for for figure 3B. Values were quantified by ImageJ and are representative of two individual reproducible experiments. (**Revised figure 3B**)

Reviewer #2. Figure 3C: can the co-IP be done with the endogenous proteins to show it is not an “artificial” interaction?

This has now been done. Please refer to comments made reviewer 1 related to figures S5E, and S5F.

Reviewer #2. Figures 3K and L: quantification from at least 3 independent experiments and statistical analyses should be provided.

We apologise for the confusion and the lack of information given on our part. The data shown are representative of three independent and reproducible experiments. This has now been stated in the figure legend and relevant statistics have now been included. (**Revised figure 3K and 3L**)

Reviewer #2. Figure 5A and 5B: quantification from at least 3 independent experiments and statistical analyses should be provided.

We thank the reviewer for his/her comment. We have now included quantification values for for both figures 5A and 5B (now figure 6C and 6D). Values were quantified by ImageJ and are representative of two individual reproducible experiments. (**Revised figures 6C and 6D**)

Reviewer #2. Figure 6A: pictures should be provided, at least for the cells shown in Figure 6C.

We thank the reviewer for his/her comment. The epithelial-mesenchymal transition (EMT) score was computed using a method developed previously {Tan and Thiery et al, EMBO 2014} to quantitate the EMT spectrum. Briefly, a generic EMT signature was derived from weighted rank of epithelial and mesenchymal genes identified by microarray analysis and correlated with known EMT markers, published genesets, or E-cad, N-cad immunofluorescence staining. Two-sample Kolmogorov-Smirnov-based method was then applied to estimate the degree of enrichment using the generic EMT signature. Unfortunately, no pictures are available for this data as the results are geneset enrichment-based.

Reviewer #2. Figure 6C: the blot quality is not good. Cells should have been treated with a c-met inhibitor to control for the P-c-met band. Moreover, additional experiments to show that c-met inhibition / siRNA knock down reverts the EMT phenotype should have been done. Which c-met P-site was investigated?

We agree. Utilising these cell lines we now demonstrate that UMUC3 has the highest c-MET activity in the absence of HGF treatment. Importantly, T24 has no detectable c-MET activity in the absence of HGF exposure. All three cells displayed increase c-MET phosphorylation following HGF addition, an effect annulled in cell lines co-treated with the HGF inhibitor JNJ38877605 (**new Sup. Figure 9A**). The phospho c-MET antibody detects Y1234-1235 which represents activated forms of the receptor. Please see above comments for c-Met inhibition and EMT reversal.

Reviewer #2. Figure 6E: quantification from at least 3 independent experiments and statistical analyses should be provided. Confocal pictures should be provided.

Please see reviewer 1 comments above related to figures S10G, S10H.

Reviewer #2. Page 12 of the manuscript: there are errors in figure numbers. It is not Figure 6 but Figure 4

We thank the reviewer for noting our errors. These have now been corrected.

Reviewer #3. Reference 21 hardly justifies studying c-Met in bladder cancer. Better references include: PMID: 25816892.

We thank the reviewer for his/her thorough review of our manuscript. We agree we have now included the following references. PMID: 11896103, 19121849, 16400012, 26225770 and the reference suggested by the reviewer 25816892

Reviewer #3. Also, the following paper suggests MET is down-regulated in aggressive bladder cancer: PMID: 24853099. How do the authors reconcile this?

Thank you for highlighting this article. Upon analyses of this article, the methodology used and results appear conclusive. As part of the reviewers suggestion we now include 4 articles that indicate that c-MET correlates with bladder cancer progression (see above). However, we cannot conclusively reconcile the contradictory results presented from this group and those of others. Nevertheless, we now clearly demonstrate that activated c-MET correlates

with EMT and that HGF leads to poor prognosis in bladder cancer patients (see comments below).

Reviewer #3. *In the Results (Line 440) the authors state that c-Met correlates with tumor grade and poor prognosis– please provide reference for this.*

The following references have been added: PMID: 11896103, 26225770, and 19121849.

Reviewer #3. *How does HGF activate TGF-B pathway? There is a section heading “c-MET activation drives TBR-dependent SMAD2 phosphorylation” – but c-MET has not really been studied here. We know only that HGF induces TGFB signaling, but how the two pathways link is not determined. c-MET activation is even in the title, but the actual role of c-MET is not established. It is likely a correct assumption, but should be demonstrated. Assuming that it is indeed through c-MET, how is TGFB pathway activated by c-Met? This key question is not addressed.*

We thank the reviewer for his/her thoughtful comment. As c-Met is the receptor for the HGF ligand we sought to validate that indeed c-Met was required for HGF induced TGFβ receptor signalling. We now demonstrated that chemical inhibition of c-Met by the c-MET inhibitor JNJ38877605 reverts HGF induced EMT in NBT-II cells and diminishes HGF induced pSMAD2 at both the c-terminal and linker phosphorylation sites. Please refer to reviewer comments above related to figure 2D, S1E, S9A, S10A, S10B, S10C, S10D, S10E, S10F, S10G.

We would like to address the reviewers comment,” how is TGFB pathway activated by c-Met?” and their following comment together.

Reviewer #3. *This paper is like two papers in one. After establishing a link between HGF and TGFB signaling, the investigators jump abruptly (Fig 3-5) into c-SRC signaling and SMURF2 without any logical transition. The rationale for these experiments needs to be clearer in the results (building on text in Introduction), and the transition needs to be smoother. The work on c-SRC and SMURF2 is elegant, innovative and valuable – but it does not appear to have anything to do with the rest of the paper, and especially nothing to do with the bladder cancer models. Can the investigators demonstrate relevance of SMURF2 in bladder cancer? The authors jump equally abruptly back to HGF-induced EMT/invasion in bladder cancer (Figs 6 and 7) in the latter part of the paper, picking up the story started in Fig 1 and 2.*

.....*Assuming that it is indeed through c-MET, how is TGFB pathway activated by c-Met? This key question is not addressed.*

We thank the reviewer for his/her comment and apologise for any confusion in regards to the flow of our study. The underlying goal of our study was to find potential mechanisms of HGF induced invasion in bladder cancer. Following on from the setup of our HGF induced EMT platform where we demonstrate that HGF enhances cellular dispersion we then compared transcriptomics to cellular plasticity and identified the activation of the TGFβ receptor in this regard (**Figure 2A and 2B**). Furthermore, we demonstrate that HGF induces c-SRC (**Figure 2C**). To uncover novel repressors of HGF induced EMT we previously performed a chemical compound screen, where one of the hits which inhibited HGF-mediated EMT was the c-SRC

inhibitor AZD0530 (**Figure 3A**). Furthermore, under these conditions c-SRC inhibition diminished phosphorylation of SMAD2 (**Figure 3B**). Therefore we concluded from these two points that c-SRC is indispensable for HGF induced EMT and that TGFβ receptor signalling may function downstream of c-SRC to permit this.

Recently, it has been published that c-SRC can phosphorylate the HECT E3 ligase NEDD4 leading to NEDD4 activation. As SMURF2 is a negative regulator of TGFβ receptor activity and a member of NEDD4 family of HECT E3 ligases we tested to see if indeed c-SRC phosphorylates SMURF2 as well. Our further analysis revealed that these phosphorylation sites are critical to SMURF2 activity, whereby phosphorylation of SMURF2 by c-SRC inhibits SMURF2 ligase activity resulting in activation of the TGFβ pathway.

In short, we describe mechanistically how the HGF/c-MET axis modulates SMURF2 activity through c-SRC to enhance the TGFβ receptor pathway and induce EMT and invasion. An effect we see in the bladder cancer models as well. An outstanding question remains what is the mechanism behind the feedback loops activating TGFβ and ERK signalling following MEK and TGFβ inhibition, respectively.

Reviewer #3. *There is no correlation in this paper to patient tumors - it is primarily in vitro work with some basic cell line xenograft work. There needs to be some link back to patient samples to provide clinical context.*

We thank the reviewer for this critical comment. To further analyse the role of the HGF-cMET axis in relation to EMT and bladder cancer we compared the TCGA dataset (n=408) to our established EMT signature. We now demonstrate that HGF, c-MET and p-c-MET correlate with EMT in bladder cancer and that high levels of HGF correlate with poorer overall survival (**New figures 7A, 7B**). We also now demonstrate a clear correlation between p-c-SRC and pSMAD2 in a panel of bladder cancer patients by IHC. (**New figures S8A, S8B, S8C**). We feel this conclusively demonstrates the relationship between active c-SRC

Reviewer #3. *There is no logic to the use of cell lines in this paper. One major underlying flaw is the use of a carcinogen (BBN) – induced rat bladder tumor cell line NBT-II without any mention that it is a rat model. This undermines the entire foundation of the HGF-MET signaling. This work needs to be carried out in a panel of human bladder cancer cell lines. UC3 and T24 are introduced later, but there is random use of one or the other, with no rhyme or reason. Fig 7 mixes UC3, T24 and NBT-II arbitrarily. In Figure 6 EMT is scored in a nice panel of cells lines, but it would be important to show c-Met expression and phosphorylation in a broader spectrum of these cell lines (not just the four selected lines in Fig 6b).*

We agree. We now mention in the text that NBT-II is a rat bladder carcinoma cell line. Furthermore, as mentioned above we analysed the RNA datasets of the bladder cancer cell lines. Please refer to reviewer 1 and 2 comments above concerning p-c-MET status of T24, J82, and UMUC3 and importantly the reasoning behind the specific use of each cell line used in our experiments.

Similar to our analyses of NBT-II in figure 3B we now show the effect of TGF β inhibitors, MEK inhibitors, and c-SRC inhibitors on T24, UMUC3, and J82 cells. In line with our previous results we observe that in T24, J82, and UMUC3 cells c-SRC inhibition completely abolishes HGF induced pSMAD2. Further we noted that c-SRC inhibition with AZD0530 enhanced pERK in cells. It is extremely important to note that the effect of these inhibitors on intercellular signalling and the intrinsic feedback loops differs greatly at different time-points between these cell lines. The results demonstrated below are performed at 2 hours similar to our experiments done in NBT-II cells (**New figures S9C, S9D, S9E, 8F**).

Reviewer #3. The investigators use A83-01 and LY2157299 to inhibit TGF β signaling – without justification of differences. Why one versus the other for different assays? They do not use one to validate the other, which would make sense, but instead interchange the two arbitrarily. In the middle of Fig 7 they switch between two TGF β inhibitors.

We thank the reviewer for his/her excellent comment. We now demonstrate new data exploring the activity of LY2157299 in both invasion assays and soft agar growth (**New figures 8A, 8B, 8C, 8D, 8E**). For our in vivo experiment analysing the effects in the combination with TGF β and MAPK inhibitors we choose to analyse the effects of LY2157299 rather than A83-01 as LY2157299 is presently being used in early phase clinical trials while A83-01 is not.

Reviewer #3. Fig 6a: Need to define in legend how EMT scores were measured.

We thank the reviewer for his/her comment. The epithelial-mesenchymal transition (EMT) score was computed using a method developed previously {Tan and Thiery et al, EMBO 2014} to quantitate the EMT spectrum. Briefly, a generic EMT signature was derived from weighted rank of epithelial and mesenchymal genes identified by microarray analysis and correlated with known EMT markers, published genesets, or E-cad, N-cad immunofluorescence staining. Two-sample Kolmogorov-Smirnov-based method was then

applied to estimate the degree of enrichment using the generic EMT signature. We refer to the following reference in the text (Tan. EMBO Mol. Med. 2014)

Reviewer #3. *In Fig 6d, what happens to c-Met phosphorylation under same conditions?*

Please see reviewers 1 comments above.

Reviewer #3. *Fig 6F-G – This is not really IHC but rather H&E (same in Fig 7F).*

This has now been corrected.

Reviewer #3. *It is easy to find a location of muscle invasion in one tumor and an area of no invasion in another – this hardly establishes an effect of the inhibitors. Please include number of mouse tumors established for each condition and categorize T-stage for each mouse (ie Ta/T1/T2/T3 etc). Comment if there were any nodal metastases. In Fig 6G one can see the entire bladder which is more convincing – in F one sees only part of interface with bladder wall. It is unclear how the investigators determined in lower right panel of F that the tumor is separate from the urothelium. The black line appears to be drawn arbitrarily through tumor.*

We thank the reviewer for his/her suggestion. This information has now been included. Daily administration of vehicle control only (Control) or A83-01 (50mg/kg) (n=5). T-stages at end of experiment in control treated mice indicated T3-3 mice, T2-2 mice. All A83-01 treated mice were T1. Daily administration of vehicle control only (Control) or LY2157299 (80 mg/kg) (n=3). T-stages at end of experiment indicated T3-1 mice, T1-2 mice. All LY2157299 mice were T1. We now also include new images of A83-01 treated mice displaying the entire bladder (previous figure 6F, now figure 7F). To determine invasion, all tissue sections were analysed by a trained pathologist blinded to the study.

We now also include data on the inhibition of lung metastasis with both the inhibitors A83-01 and LY2157299. As shown in **new figure S14G, S14H, S14I, S14J** intraluminal injection of

UMUC3 cells resulted in metastasis to lungs in 4 out of 8 mice. An effect which was abolished in the presence of either A83-01 or LY2157299.

Reviewer #3. The difference in pSMAD2 in panel H is subtle – much less than the summary numbers indicated in bar graph below (looks like 80-90% positive staining).

Tissue sections were quantified by a trained pathologist blinded to the results of the experiment. The percentage of pSMAD2 staining based on his assessment is correct.

Reviewer #3. Were results with both inhibitors (bar graphs for F and G) really identical?

Although the values are not indicated in the text the values are 13 and 15% respectively.

Reviewer #3. In Fig 6 (lines 460-462) the authors claim: “Western blot analysis in UM-UC-3 cells indicates that loss of TBR signaling through treatment with A83-01 results in enhanced E-cadherin expression indicating that TBR suppression reverts UM-UC-3 cells to a more epithelial state (Fig. 6e)” - but there is no E-cadherin expression in either condition on this Western blot.

Please see comments to reviewer 1 above.

Reviewer #3. In Fig 7F in the LY panel, the blow up should be of the middle of the tumor front where invasion would be more likely – and not way off at the edge of the tumor.

This has now been included. **(Previous figure 7F now figure 8I).**

Reviewer #3. Fig 7G: “tumour invasion” implies depth of invasion (lamina propria, muscularis propria, etc), but the variables entered are G1, G3 and G4. This implies “grade”. But it would be unusual for the grade of a tumor to change with drug therapy – especially the drastic changes described here. Also, bladder cancer is usually graded G1, G2 and G3 – there is no grade 4. The images in F do not reflect any difference in grade. Why is there a single tumor size for each condition – only a single mouse?

This data has now been removed. Please see comment below for the number of mice inoculated. Please see comments above related to tumour stage following treatments.

Reviewer #3. The investigators jump around between intravesical and subcutaneous mouse models. Why use any subcutaneous model if able to do orthotopic? In 7F the authors state T24 injected into the bladder wall of nude mice – but this implies intramural and not intraluminal inoculation. Which was used? Only the intravesical inoculation of bladder cancer cells is described in the methods for the orthotopic model. This is a very tricky model and it is hard to get good tumor take. It is surprising that the authors have made this work for both T24 and UC3. Have the authors verified that both cell lines are genotypically what they are supposed to be? This is not indicated in the methods – T24 is actually not even listed in the methods. Please indicate how many mice were inoculated, what proportion had tumors, and please show growth curves over time. The main advantage of this model is that drugs can be administered intravesically. Here they are administered systemically, so intramural injection would have been a more robust model.

We thank the reviewer for his/her excellent comment and to give us the opportunity to explain our rationale in using these cell lines. Our initial thought was to demonstrate that HGF-mediated EMT and invasion could be inhibited by treating mice with a TGF β receptor inhibitor. To this end we injected the highly aggressive bladder cell line UMUC3 into the bladder and treated the mice systematically with either TGF β receptor inhibitors A83-01 or LY2157299. As observed mice treated with either the T β R inhibitors A83-01 or LY2157299 tumour formation remained superficial with no apparent invasion of the submucosal layer. We choose to treat the mice systematically and not intravesically as this would have meant anaesthetizing the mice and using a catheter to inject the tumour with each treatment. The less aggressive T24 cells were not able to form tumours using this protocol.

However, we noted during these experiments that the mitotic index was not significantly reduced. Furthermore, western blot analyses revealed that addition of A83-01 and LY2157299 induced MAPK activity as determined by an increase in pERK. Therefore we sought to analyse what effect the combination of a TGF β receptor inhibitor and a MEK inhibitor would have on tumour growth and invasion. We now present data demonstrating the effect of A83-01 or LY2157299 and PD0325901 in UMUC3 cells (**New figure 8E**). Furthermore, we tested this combination in invasion assays in both UMUC3 and T24 cells (**New figure 8A, 8B, 8C, 8D**). Next, we tested this combination in vivo by subcutaneously injecting UMUC3 cells into mice and treating with A83-01 and PD0325901. Again we demonstrate that the combination of TGF β receptor and MEK inhibition decreased tumour size compared to either treatment alone. This data has now been removed.

We then explored this combination in a more clinically relevant form. To do so we wanted to inject bladder cancer cells into the bladder wall and permit tumour formation, similar to what is observed in the clinic. We initially attempted this with UMUC3 cells but due to the aggressive nature of this cell line the mice died too quickly to effectively treat them. To perform these experiments accurately we utilised the less aggressive cell line T24. As correctly indicated by the reviewer T24 cells were directly injected into the bladder wall and the mice were treated with LY2157299 or PD0325901 or a combination of the two. Importantly our rationale in choosing this combination is that both of these compounds are in clinical trials. A83-01 is not.

The following sentence has been added in the main body of the text.

We then wanted to gain insight into the potential anti-tumour activity of MEK and T β R inhibition in vivo. In most bladder cancers initial tumourigenesis originates within the urothelial layer lining the inner surface of the bladder with tumours eventually invading the surrounding smooth muscle. Therefore, we sought to generate a clinically relevant orthotopic bladder cancer model by injecting bladder cancer cells intramurally. We initially attempted these experiments with UMUC3 cells. However, due to the aggressive nature of these cells tumours rapidly invaded surrounding tissues with mice succumbing to the disease within a few days after detectable tumour formation. However, intramural injection using the less aggressive T24 bladder cancer cell line permitted tumour formation without affecting the overall fitness of the mouse. T24 cells were generated to stably express a mammalian codon-optimized firefly luciferase2 permitting bioluminescence detection. Following tumour formation mice were paired and treated by oral gavage with either vehicle or 80 mg/kg LY2157299 or 25 mg/kg PD0325901 or a combination of the two. Similar to our previous observations using our intravesical inoculation model the addition of LY2157299 decreased tumour size and inhibited the invasion of established tumours through the submucosa and muscular layers.”

Reviewer #3 Please indicate how many mice were inoculated, what proportion had tumors, and please show growth curves over time.

Reviewer #3. Figs 6 and 7: The legend says that tumor volume was measured twice weekly for the T24 orthotopic xenografts – but these results are not shown. Since this is a luciferase-transduced cell line, presumably the authors mean that bioluminescence was measured. Please show the growth curves rather than sample mice, which can be quite deceiving. It is again critical to show how many mice were used.

We agree. We now include data on the number of mice inoculated and overall tumour formation for both experiments in the figure legends and materials and methods. In brief (**previous figures 6F,G now figures 7F, G**) for the A83-01 experiment 50 mice were inoculated with 10 mice forming tumours. These were then segregated into two prongs, vehicle treated and A83-01 treated. For the LY2157299 experiment 45 mice were inoculated and 6 formed tumours. Once again these were separated into 2 equal prongs and treated accordingly. We now state in the materials and methods that, "For the first experiment (A83-01) 50 mice were inoculated with 10 mice forming tumours. Mice were then separated into vehicle control (n=5) or treated with A83-01 (n=5) in gavage solution (DMSO:Saline, 3:2, v/v) (80mg/kg daily for two weeks). Similarly, in the second experiment (LY2157299) 45 mice were inoculated with 6 mice forming tumours. Mice were then separated into vehicle control (n=3) or treated with LY2157299 (n=3) in gavage solution (0.5% hydroxyl propylmethyl-cellulose +0.2% Tween 80) (80mg/kg daily for two weeks). For bladder wall injection: 0.5 cm slit was made on the abdomen of mice under anaesthesia to expose the bladder. The mice were catheterized with PBS to maintain the bladder at about 85% full. 10 μ l of 1×10^6 T24-luc2 cells prepared as described for UMUC3 were injected into the bladder wall using a 31 gauge insulin syringe (BD) under stereoscope (Leica) with 3.5x magnification and the slit was sutured after procedure. Tumour implantation was assessed by bioluminescence imaging with IVIS Spectrum Imaging System 200 (Xenogen). Imaging was performed 10 minutes after an intraperitoneal injection of 150 mg/kg firefly D-Luciferin in 200 μ l of PBS. Signal intensity was quantified in photons per second per region of interest. Respective drugs were administered when bioluminescence intensity reached $\sim 2 \times 10^8$ photons/second. A83-01 administered in gavage solution (DMSO: Saline, 3:2, v/v) (80 mg/kg daily for two weeks). LY2157299 administered in gavage solution (0.5% hydroxyl propyl methyl-cellulose + 0.2% Tween 80) (80 mg/kg daily for two weeks). PD0325901 administered in gavage solution (1% polysorbate 80) (25 mg/kg daily for 2 weeks)." We also include the original bioluminescence evaluation and corresponding growth curves. (**new figures 8 G, 8H; Sup. Fig. 14B, 14C, 14D, 14E**).

Reviewer #3. The main endpoint reported in the

mouse model is depth of bladder wall invasion. The authors show sample images, but do not quantify this adequately. How many were muscle invasive? What about pT1 or pT3? Showing only part of the bladder wall can also be deceiving. Invasion into lamina propria is still early and not late stage bladder cancer, as the authors suggest. Please show an example of lymphovascular invasion – was this determined with IHC for a vessel wall marker (e.g. CD31) or just H&E?

We thank the reviewer for his/her comments. Please see figures and quantification above. Lymphovascular invasion was determined by H&E and evaluated by a trained pathologist blinded to the study.

Reviewer #3. *The authors emphasize the clinical relevance of their orthotopic bladder cancer model. They should remove this text. This model is not clinically relevant. It conforms to the “seed and soil” hypothesis, so it is biologically relevant, but it is far from clinically relevant.*

This has now been corrected.

Reviewer #3. *Would it not be useful to knockout SMAD2 to show its importance in HGF effects?*

These experiments were previously performed Fig. 2 G, H. now Sup fig 2C,D

Reviewer #3. *Figure 2: There is not enough evidence to support the conclusion that SMAD2L but not SMAD2C is likely to be mediated by ERK. The differences are subtle.*

We apologise for the confusion in our statement. The C-terminal tail of pSMAD2 can only be phosphorylated by the TGF β receptor, while pERK has been shown to phosphorylate the linker region of SMAD2. The sentence has been edited as follows. “**In addition, analysis of NBT-II cells treated with HGF in a time-dependent manner demonstrates a rapid induction of pERK at 5 mins with an analogous increase in the phosphorylation of the SMAD2 linker region indicating that SMAD2 linker phosphorylation may be mediated by ERK.**”

Reviewer #3. *Fig 2E: A83-01 reverted only a small proportion of the pSMAD2 activation.*

We agree. The statement has been rewritten as follows, “HGF treatment enhanced nuclear pSMAD2 localisation an effect that was **partially** reverted upon the co-addition of A83-01.”

Reviewer #3. *Figure 3 – Please indicate what pY is in legend.*

This has now been corrected.

Reviewer #3. *Fig 7a-c: There is no convincing evidence here that TGFB inhibitor leads to increased ERK phosphorylation in either cell line.*

This statement has now been edited.

Reviewer #3. *Line 458-490: Note errors in this section referring to Fig 3 rather than Fig 6.*

This has now been corrected.

Reviewer #3. *Fig 7: Why are tumors decreased in size with inhibitors if same or increased proliferation?*

We thank the reviewer for this excellent comment and to allow us to further explain our observations. As noted by the reviewer treatment of mice with LY2157299 and A83-01 (S14B, S14C, S14D, S14E) and later on with PD0325901 resulted in decreased overall tumour size. However, analysis of the remaining tumour in TGF β receptor inhibitor treated mice indicated that the remaining tumours had a high mitotic index. We hypothesized that these observations might be the result of an adaptive response in a sub population of the tumour. As we had observed that treatment with TGF β receptor inhibitors enhances MAPK pathway activation (Fig. 8F) we further analysed the combination with both TGF β inhibitors and MAPK inhibitors.

Reviewer #4. *My only concern and request is to emphasize that a docked model is a model and not necessarily a "real" structure. This could be discussed a bit further in the MS. Hence, the authors may want to indicate in the text (e.g. discussion) that the model suggests a possible binding mechanism that is not necessarily the only possible binding geometry.*

We thank the reviewer for their critical reading of our manuscript. We agree and have added/edited the following sentences to the manuscript.:

“Thus, the proposed structural model of the C2-HECT complex suggests a possible molecular mechanism for the observed effect of Tyr434 phosphorylation; however, this may not be the only optimum binding geometry and there could be other similar modes of interactions between the two domains in the ensemble of structures.”

“Subsequently, the complex structure was refined by energy minimization to obtain the final representative docked HECT-C2 complex state structure of Smurf2 (Figure S7E). “

Reviewers' comments:

Reviewer #1, Expertise: Tgfbeta

(Remarks to the Author):

I commend the authors for their detailed and constructive response to my critiques. I am strongly supportive of publishing.

Minor comment:

In revised figures S5E and S5F the labelling on the left of the blots I think should be IB not IP.

Reviewer #2, Expertise: HGF, metastasis (Remarks to the Author):

This revised version contains a number of additional data. Some improvement has been made and some of my requests have been addressed adequately such as quantifications. However, a number of issues remain. Some of my questions have not been addressed in full or adequately. In particular, whether HGF triggers TGFβ signaling through a src-smurf2 pathway remains unclear. I provide below a few examples.

Figure S11 A-D: No statistic is provided comparing + to – HGF. But it seems that the invasion is poorly (B) or even not (D) increased upon HGF stimulation. Therefore, the differences seen with the construct expression is independent of HGF. Why this assay was used instead of the EMT live imaging?

Figure S11: This figure is unrelated to HGF.

Figure 6B: The stability of TBR appears unchanged with HGF

Figure S5C,D: This figure is descriptive, with no quantification. How does Src siRNA influence HGF dependent cell motility, velocity upon time is unclear.

Reviewer #3, Expertise: Bladder cancer

(Remarks to the Author):

Reviewer #1

It is not correct to suggest that all high grade bladder cancer has undergone EMT. Bladder tumors in patients and human bladder cancer cell lines span the spectrum from epithelial to mesenchymal. The authors show this themselves in Fig 7.

Reviewer #3

The authors need to justify adding so many authors.

The authors have added an almost overwhelming amount of new data (66 new figures!). Most of the signaling work is much improved. But I have concerns with the xenograft experiments.

T24 is a highly aggressive cell line and it should not be emphasized that it is "less aggressive".

The authors describe that UM-UC3 injected into the bladder wall kills mice within days. This implies either that these cells are not really UM-UC3 – or the investigators are doing something wrong. The authors do not respond to the request for genotyping of cell lines. This is absolutely essential.

The description of the orthotopic model being clinically or biologically relevant remains erroneous. Please just label them as orthotopic models – there is nothing special that warrants particular designation as “relevant”. The addition of the statement that bladder cancer usually originates from the urothelium and the authors are therefore injecting into the bladder wall is contradictory (lines 712-714). Tumor cells are usually injected below the mucosa, so that they are already invasive from the start. Depth of invasion is difficult to interpret when the cells are injected into the bladder wall – this really only makes sense for intraluminal inoculation. The growth curves should be the main outcome of the animal experiments – but it remains difficult to align growth inhibition when no effect on proliferation.

A tumor engraftment rate of 6 in 45 and 10 in 50 suggests a highly inadequate xenograft model. T24 and UM-UC3 cell lines both grow well after intramural injection – with near 100% engraftment rates. The authors need to seek collaboration with a group that masters these techniques. The treatment arms are too small – especially considering the variability observed with bioluminescence. The animal work is not convincing, and it is disturbing that the authors did not reveal some of these numbers initially. The growth curves now are perfect – so why not include them in the first draft of the paper? These curves give the impression of being too good to be true.

Figure 7 – It is misleading to indicate % of muscle invasive tumors when the group sizes are so small – this would be better summarized as simple ratios in text. The sample sizes are not adequate to conclude much about T stages.

It remains unclear why the authors have randomly interchanged LY2157299 and A83-01. They say that LY2157299 was preferred because it is being tested in clinical trials – so why use A83-01 at all? And especially why in first xenograft study? Is there a difference between the two with respect to off-target effects?

Line 567 – what is an “epidermoid” bladder cancer? This is not conventional terminology. It compares to sarcomatoid and papillary – implying that the authors mean non-papillary conventional urothelial carcinoma.

Reviewer #4, Expertise: molecular modelling and dynamics (Remarks to the Author):

I carefully read the revised version of the manuscript and the response to reviewers. The authors extensively modified the manuscript and responded successfully to my concerns and made the necessary changes to the manuscript.

c-Met activation leads to the establishment of a TGF β receptor regulatory network required for bladder cancer invasion.

Responses to reviewer's comments:

We thank the reviewers for the thoughtful and thorough review of the manuscript. We have revised the manuscript according to the reviewers' suggestions and have addressed all the concerns brought forward by the reviewers.

Reviewer #1. I commend the authors for their detailed and constructive response to my critiques. I am strongly supportive of publishing.

Reviewer #1. In revised figures S5E and S5F the labelling on the left of the blots I think should be IB not IP.

We thank the reviewer for taking the time to review our manuscript. In regards to figures S5E and S5F the figures are labelled correctly as IP refers to the two immunoblots which were analysed following immunoprecipitation.

Reviewer #2. Figure S11 A-D: No statistic is provided comparing + to - HGF. But it seems that the invasion is poorly (B) or even not (D) increased upon HGF stimulation. Therefore, the differences seen with the construct expression is independent of HGF.

We thank the reviewer for his comment. We now include statistics comparing +/-HGF in both S11B and S11D. Furthermore, as demonstrated in Sup figure 10A, HGF-induced migration in both UMUC3 and T24 cells indicating that migration is indeed dependent on HGF signalling. However, for Sup figure 11B and D, we would expect that if indeed HGF induces TGFbeta-mediated migration, ectopic expression of SMURF2 would mitigate these effects as SMURF2 is the "target" of HGF/c-SRC. Therefore, we would expect minor changes between +/- HGF. However, we still see a slight increase in HGF-induced migration potentially due to transfection efficiency of UMUC3 and T24 cells. Nevertheless, the purpose of the experiment is validated as SMURF2 FF significantly decreases migration compared to SMURF2 EE in the presence of HGF. It is also extremely important to note that T24 and UMUC3 cells are highly mesenchymal cell lines and therefore any further increase in HGF-induced EMT may be limited.

Reviewer #2. Why this assay was used instead of the EMT live imaging?

We thank the reviewer for their comment. Analysis of the mesenchymal marker vimentin was performed in Sup figure 11E. Again, we clearly show that in the presence of HGF, SMURF2 FF hinders the expression of vimentin compared to SMURF2 EE suggesting that SMURF2 FF reverts these cells to a more epithelial phenotype even in the presence of HGF.

Reviewer #2.*Figure S11: This figure is unrelated to HGF.*

Please see comments above.

Reviewer #2.*Figure 6B: The stability of TBR appears unchanged with HGF.*

We thank the reviewer for his comment. Although we note that HGF only slightly induced TGFbeta receptor I expression in NBT-II cells, we observe a significant increase in TGFbeta receptor I expression in UMUC3 cells (sup fig. 7F), an effect significantly downregulated by AZD0530 in both cell lines. More importantly however, HGF addition enhanced TGFbeta receptor membrane expression. As only a small proportion of TGFbetaR expression is at the cell surface prior to being ubiquitinated by SMURF2 and directed towards caveolin positive vesicles leading to degradation, this is undisputable proof that HGF enhances TGFbeta receptor stabilisation and downstream TGFbeta signalling.

Reviewer #2.*Figure S5C,D: This figure is descriptive, with no quantification. How does Src siRNA influence HGF dependent cell motility, velocity upon time is unclear.*

In respect to the reviewers original comments,“ Figure S4A: EGF, HGF and IGF should appear more clearly on the figure. AZD0530 is more effective on EMT with increasing doses. What does this mean? Are doses of 2 and 8 microM still specific to src? Controls should be provided including siRNA src.“, we clearly demonstrate that similar to AZD0530, c-SRC siRNA inhibits HGF-induced cell scattering and blocks vimentin expression. We have previously shown that c-SRC inhibition with AZD0530 decreases HGF-dependent cell motility (Fig 3A). Taken together we feel we have conclusively demonstrated that c-SRC inhibition by chemical or genetic means mitigates HGF-induced EMT.

Reviewer #1. *It is not correct to suggest that all high grade bladder cancer has undergone EMT. Bladder tumors in patients and human bladder cancer cell lines span the spectrum from epithelial to mesenchymal. The authors show this themselves in Fig 7.*

We thank the reviewer for their comment. We have now added the following sentence, “As certain bladder cancer subtypes display a mesenchymal-like phenotype we focused our attention on cell lines displaying the highest EMT scores to accurately reflect what is observed in clinical settings.”

Reviewer #3. *The authors need to justify adding so many authors. The authors have added an almost overwhelming amount of new data (66 new figures!). Most of the signaling work is much improved. But I have concerns with the xenograft experiments.*

We thank the reviewer for reading our manuscript. During the revision of this manuscript, the corresponding author moved labs from CSI Singapore to Curtin University with a number of the original authors moving to other labs (A.P.K. and P.T.D.). Both A.P.K. and P.T.D. participated in experimental design and interpretation. As an expert of Ras and BRAF

signalling A.S. performed the experiments on B-TRCP. M.R., H.K.H, C.R., were involved in tumour collection and analysis.

Reviewer #3. T24 is a highly aggressive cell line and it should not be emphasized that it is “less aggressive”.

We agree but we wish to note that in our hands T24 is less aggressive than UMUC3. Nevertheless the sentence has been changed, “ However, intramural injection using the T24 bladder cancer cell line permitted tumour formation without affecting the overall fitness of the mouse.”

Reviewer #3. The authors describe that UM-UC3 injected into the bladder wall kills mice within days. This implies either that these cells are not really UM-UC3 – or the investigators are doing something wrong. The authors do not respond to the request for genotyping of cell lines. This is absolutely essential.

We thank the reviewer for their comment. As a clarification intramural injection of UMUC3 cells into the bladder wall resulted in the death of the mouse within 10-14 days. Limiting our ability to observe significant changes between TGFBRi and MEKi treatment. We now include genotyping of both T24 and UMUC3 cells and the STR profiles found on the ATCC website.

[REDACTED]

[REDACTED]

[REDACTED]

Reviewer #3. *The description of the orthotopic model being clinically or biologically relevant remains erroneous. Please just label them as orthotopic models – there is nothing special that warrants particular designation as “relevant”.*

This sentence has now been removed to respond to this criticism. However, we believe that future studies will help clarifying the usefulness of this model for clinical trials.

Reviewer #3. *The addition of the statement that bladder cancer usually originates from the urothelium and the authors are therefore injecting into the bladder wall is contradictory (lines 712-714). Tumor cells are usually injected below the mucosa, so that they are already invasive from the start. Depth of invasion is difficult to interpret when the cells are injected into the bladder wall – this really only makes sense for intraluminal inoculation. The growth curves should be the main outcome of the animal experiments – but it remains difficult to align growth inhibition when no effect on proliferation.*

A tumor engraftment rate of 6 in 45 and 10 in 50 suggests a highly inadequate xenograft model. T24 and UM-UC3 cell lines both grow well after intramural injection – with near 100% engraftment rates. The authors need to seek collaboration with a group that masters these techniques. The treatment arms are too small – especially considering the variability observed with bioluminescence. The animal work is not convincing, and it is disturbing that the authors did not reveal some of these numbers initially. The growth curves now are perfect – so why not include them in the first draft of the paper? These curves give the impression of being too good to be true.

We thank the reviewer for his comments. We would like to respond to these comments together. For clarification we performed two sets of mouse experiments. For the intraluminal experiments, we utilised trypsin instead of acid to get carcinoma cell implantation in the mucosa limiting tumour formation but also limiting tissue degradation allowing for a more clinically relevant scenario. Under these conditions, we showed an engraftment rate of 6 in

45 and 10 in 50. Treatment of these mice with either A83-01 or LY2157299 resulted in significant tumour reduction and more importantly complete loss of tumour invasion. No significant variability was observed in bioluminescence as demonstrated by the significant p value. The relevant growth curves were not included in the original manuscript, as we felt that invasion rather than proliferation is not the main parameter to evaluate here.

The second set of experiments performed were based on intramural injection as we wanted to clearly demonstrate that TGFb receptor inhibition limited the invasion of these tumours in the surrounding smooth muscle layer. In these experiments, both T24 and UMUC3 displayed a near perfect engraftment rate of 95% in line with engraftment rate suggested by the reviewer. Under these conditions, we again noted TGFb receptor inhibition significantly limited T24 invasion. However, we did not directly measure depth of invasion in our intramural model as these tumours are injected directly into the tumour wall, as correctly pointed out by the reviewer. Depth of invasion was only measured in our intraluminal experiments. Taken together we believe these experiments were performed and evaluated correctly.

Reviewer #3. *Figure 7 – It is misleading to indicate % of muscle invasive tumors when the group sizes are so small – this would be better summarized as simple ratios in text. The sample sizes are not adequate to conclude much about T stages.*

As suggested by the reviewer we have taken these data out and included them as ratios in the text.

Reviewer #3. *It remains unclear why the authors have randomly interchanged LY2157299 and A83-01. They say that LY2157299 was preferred because it is being tested in clinical trials – so why use A83-01 at all? And especially why in first xenograft study? Is there a difference between the two with respect to off-target effects?*

We thank the reviewer for their comment. We have consistently used A83-01 and LY2157299 for all the experiments performed to demonstrate the effectiveness of both of these compounds that target the TGFb receptor 1. LY2157299 was used instead of A83-01 in our intramural model as LY2157299 is presently being used in clinical trials. As far as we know, there is no difference between them with respect to off-target effects (they both target ALK4, ALK7 to a weak extent). Instead of using one here, in our study we used two different (with respect to chemical structure) and pharmaceutically relevant inhibitors that target TGFBR1 to demonstrate our point.

Reviewer #3. *Line 567 – what is an “epidermoid” bladder cancer? This is not conventional terminology. It compares to sarcomatoid and papillary – implying that the authors mean non-papillary conventional urothelial carcinoma.*

Epidermoid tumors is still a term employed by French pathologists including our co-author from Foch hospital (largest urology clinic in France) we have now replaced it by the conventional terminology : urothelial carcinoma with squamous differentiation.

Reviewer #4. *I carefully read the revised version of the manuscript and the response to*

reviewers. The authors extensively modified the manuscript and responded successfully to my concerns and made the necessary changes to the manuscript.

We thank the reviewer for taking the time to review our work.

Reviewers' comments:

Reviewer #2 (Remarks to the Author):

I have read this new revised manuscript carefully and the answers to reviewer comments. Although the manuscript contains an obvious high amount of work, is overall of high quality, and presents original and interesting results, some concerns remain:

1) I am still not convinced that HGF triggers a stabilization of TBR protein. Considering the results presented on HGF>MET>src phosphorylation>SMURF phosphorylation etc..., one would indeed expect a stabilization of TBR. However, this needs to be clearly demonstrated.

Figure 6A: That TBR1 expression is increased at the cell surface upon HGF stimulation does not prove it is more stable. It could indicate it does not internalize or rather recycles more. Why the total lysate from this experiment is not displayed? TBR1 should then also increase.

Figure 6B displays a potential, modest, increase. This is just one blot with no quantification.

Thus the authors would really need to clarify whether or not HGF leads to a clear TBR stabilization. Otherwise, the authors need to nail down the mechanism if different.

2) It is unclear whether the tumour growth and invasion in the orthotopic mouse model is actually MET dependent. The authors wrote Line 646 "we used an orthotopic mouse model of bladder cancer whereby we could analyze the role of TBR-mediated HGF-induced invasion". In fact, HGF produced by the nude mice should not be able to activate MET in the injected human bladder cancer cells. However, the authors have shown that UMUC3 cells express phosphorylated MET, giving the possibility that the tumours could then be MET dependent. However, this needs to be demonstrated through treating the mice with a specific MET inhibitor in parallel to the treatments performed (Figure 8H).

Reviewer #3 (Remarks to the Author):

Two issues not completely resolved.

1. Need statement in manuscript about genotyping of cell lines. Referring back to ATCC is only appropriate if very low passage numbers used. Cell lines need to be genotyped on a regular basis. Presumably these projects took multiple years to complete, so genotyping would be necessary at some point. Also, the UM-UC3 in this study are not behaving as one would anticipate, so particularly important to check. There is precedent for UM-UC3 being contaminated with HeLa in the urologic research community.

2. I remain skeptical about the orthotopic xenograft work. Depth of invasion of intraluminally inoculated tumors is highly subjective and likely not reproducible. Were the investigators blinded to treatment group? Only a small proportion of mice had tumor take which further undermines utility of model. The authors describe metastatic disease from intraluminal inoculation which is highly unusual.

Line 715 UM-UC3 should not kill animals "within a few days". In their rebuttal the authors state 10-14 days, but this is not "a few".

The following comment has not been addressed:

"The addition of the statement that bladder cancer usually originates from the urothelium and the authors are therefore injecting into the bladder wall is contradictory (lines 712-714). Tumor cells are usually injected below the mucosa, so that they are already invasive from the start."

The original text is:

710 In most bladder cancers, initial tumorigenesis originates within the

711 urothelial layer lining the inner surface of the bladder with tumours eventually invading the

712 surrounding smooth muscle. Therefore, we sought to generate an orthotopic bladder cancer
713 model by injecting bladder cancer cells intramurally.

The BLI images in Fig 14 B and D would indicate quite modest drug efficacy – yet massive
difference in the growth curves in Fig 14 C and E.

Fig S8 remove “epidermoid” from figure label.

c-Met activation leads to the establishment of a TGF β receptor regulatory network required for bladder cancer invasion.

Responses to reviewer's comments:

We thank the reviewers for the thoughtful and thorough review of the manuscript. We have revised the manuscript according to the reviewers' suggestions. We have addressed all the concerns brought forward by the reviewers.

Reviewer #2: I have read this new revised manuscript carefully and the answers to reviewer comments. Although the manuscript contains an obvious high amount of work, is overall of high quality, and presents original and interesting results..

We thank the reviewer for his/her overall assessment of our manuscript.

Reviewer #2: I am still not convinced that HGF triggers a stabilization of TBR protein. Considering the results presented on HGF>MET>src phosphorylation>SMURF phosphorylation etc..., one would indeed expect a stabilization of TBR. However, this needs to be clearly demonstrated. Figure 6A: That TBRI expression is increased at the cell surface upon HGF stimulation does not prove it is more stable. It could indicate it does not internalize or rather recycles more. Why the total lysate from this experiment is not displayed? TBRI should then also increase. Figure 6B displays a potential, modest, increase. This is just one blot with no quantification. Thus the authors would really need to clarify whether or not HGF leads to a clear TBR stabilization. Otherwise, the authors need to nail down the mechanism if different.

We thank the reviewer for this comment. We have now included the total lysate for TBRI for figure 6A, demonstrating conclusively stabilisation to TBRI following HGF treatment. Furthermore, we have again performed the experiments shown in figure 6B and Sup figure 7F multiple times and quantified our results. As shown below, HGF significantly induces TBRI expression in NBT-II cells at the 15 min time point ($p=0.002$), representing the transient early activation by c-SRC. Although TBRI is still upregulated compared to control at later time points we do observe an expectant TBRI decrease at 30 and 60 minutes. Most importantly, this effect is abrogated upon the addition of the c-SRC inhibitor AZD0530 at the earlier 15 min time point ($p=0.02$). Conclusively, demonstrating that the upregulation of TBRI by HGF is mediated by c-SRC. Similar, results have also been performed in UMUC3 cells (Sup figure 7F). Again we conclusively demonstrate the upregulation of TBRI following the addition of HGF.

Figure 6A

Figure 6B (Representative image)

Figure 6B (Quantification of three independent biological replicates)

		First	Second	Third	Average	StDev	Pvalue	Pvalue
	(-)	1	1	1	1	0	0	0
	15 mins	1.21314784	1.32350397	1.37142339	1.30269173	0.07259559	0.00295786	
	30 mins	0.99933945	1.23077127	1.32163806	1.18391626	0.14863551	0.12772419	
hgf	60mins	0.9284006	1.06389917	1.38591473	1.1260715	0.2101977	0.40539697	
	(-)	0.58537271	0.83242888	1.05574605	0.82451588	0.21044663	0.26600606	0.26600606
	15 mins	0.56358877	0.63000422	1.04391005	0.74583435	0.23279106	0.16601267	0.02406147
(+) AZD	30 mins	0.3719678	0.87307548	0.88974093	0.71159474	0.26317944	0.16480166	0.07269309
	60 mins	0.26738178	0.66722131	0.89757359	0.61072556	0.28520765	0.10197603	0.08731657

Sup Figure 7F. (representative image)

Concerning the comments regarding the increase of TβRI expression at the cell surface, these experiments are in line with our overall conclusions that HGF induces TGFβ receptor pathway activation through its receptor and subsequent downstream phosphorylation of SMAD2. We have conclusively demonstrated through genetic and chemical means that the TGFβ receptor is the critical node for this effect. However, we do note the concerns raised by the reviewer. It has been widely recognized in the field that the SMURF2/SMAD7 complex leads to increased ubiquitination and degradation of the TGFβ receptor complex. However, recent results have also demonstrated in SMURF2 null mice that loss of SMURF2 only results in altered ubiquitination of SMAD3. Therefore, as the reviewer correctly points out the role of SMURF2 in TGFβ receptor regulation is not truly known. However, we feel that this is completely outside of the scope of this article to determine the precise effect of SMURF2 on TGFβR. Nevertheless, we clearly demonstrate that HGF regulates a c-SRC/SMURF2 axis which results in enhanced expression of TGFβR at the cell surface and an overall increase in TGFβR expression both of which conclusively lead to increase TGFβ receptor pathway activity.

Reviewer #2: *It is unclear whether the tumour growth and invasion in the orthotopic mouse model is actually MET dependent. The authors wrote Line 646 “we used an orthotopic mouse model of bladder cancer whereby we could analyze the role of TβR-mediated HGF-induced invasion”. In fact, HGF produced by the nude mice should not be able to activate MET in the injected human bladder cancer cells. However, the authors have shown that UMUC3 cells express phosphorylated MET, giving the possibility that the tumours could then be MET dependent. However, this need to be demonstrated through treating the mice with a specific MET inhibitor in parallel to the treatments performed (Figure 8H).*

c-MET is likely to be activated through both autocrine and paracrine effects in our mouse model. However, the focus of this study is to unravel the contribution of the TGFβ receptor in invasion. Possibly other tyrosine kinase receptors aside from cMET could also contribute to the activation of the TGFβ receptor. Inhibiting cMet or other RTKs would be interesting but this is beyond the scope of the present study.

Reviewer #3: *Need statement in manuscript about genotyping of cell lines. Referring back to ATCC is only appropriate if very low passage numbers used. Cell lines need to be genotyped*

on a regular basis. Presumably these project took multiple years to complete, so genotyping would be necessary at some point. Also, the UM-UC3 in this study are not behaving as one would anticipate, so particularly important to check. There is precedent for UM-UC3 being contaminated with HeLa in the urologic research community.

Genotyping analysis confirms that all the cells lines tested are accurate. Please see below. This has also been acknowledged in the materials and methods. Furthermore, to ease the reviewers concerns we tested both our original batch of UMUC3 cells from 6 years ago and the batch we used in responding to reviewers.

FLDO-dossiernr: X19-007; ontvangen op 21 februari 2019, uitslag doorgegeven op 6 maart 2019

Omschrijving	UMUC3 - Human bladder cancer	UMUC3 (2) - Human bladder cancer	T24 - Human bladder cancer	J82 - Human bladder cancer	NBT-II - cell line derived from rat	HEK293 - Human embryonic kidney
SIN	UMUC3	UMUC3 (2)	T24	J82	NBT-II	HEK293
Extractienr.	X19-007-1	X19-007-2	X19-007-3	X19-007-4	X19-007-5	X19-007-6
Opmerkingen	match	match	match	match	no profile	partial match
Amel.	X	X	X	X Y	-	X
D1S1656	15-17.3	15-17.3	12-15	14	-	14-15-17.3
TPOX	10	10	8-11	11-12	-	11
D2S441	11	11	11-15	10-13	-	11-15
D2S1338	23	23	20-23	19	-	19-20
D3S1358	17-18	17-18	16	16-18	-	-
FGA	20-21	20-21	17-22	20-24	-	22-23-24
D5S818	12	12	10-12	12-13	-	8-9
CSF1PO	10-11	10-11	10-12	10-11	-	11-12
D7S820	8-9	8-9	10-11	9-11	-	9-11-12
D8S1179	13	13	14	8-13	-	-
D10S1248	14-15	14-15	14	13	-	14
TH01	6-9	6-9	6	9.3	-	7-9.3
VWA	16-17	16-17	17-19	17-18	-	-
D12S391	22	22	17-18	24	-	19-20-21
D13S317	8	8	12	10-12	-	12-13-14
Penta E	12	12	7-10	12-15	-	7-15
D16S539	8-9	8-9	9	11-12	-	9-13
D18S51	14	14	16-18	10-12	-	-
D19S433	14-15	14-15	13-14	12-13	-	-
Penta D	13	13	11-15	9	-	9-10
D21S11	31	31	29	30-31	-	28-30.2
D22S1045	16	16	16	16	-	15-16-17
DYS391	-	-	-	10	-	-

Additional remarks to:

- X19-007-1 100% match to known profile
- X19-007-2 100% match to known profile
- X19-007-3 100% match to known profile
- X19-007-4 100% match to known profile
- X19-007-5 no human DNA detected
- X19-007-6 profile fits HEK293. However, HEK293 is extremely variable and 100% matches are nearly impossible

Reviewer #3:*I remain skeptical about the orthotopic xenograft work. Depth of invasion of intraluminally inoculated tumors is highly subjective and likely not reproducible. Were the investigators blinded to treatment group? Only a small proportion of mice had tumor take which further undermines utility of model. The authors describe metastatic disease from intraluminal inoculation which is highly unusual.*

We thank the reviewer for his/her comments. All analysis regarding tumour invasion was performed by a trained pathologist in a double-blind manner. As indicated clearly in our previous response to reviewers we felt it appropriate to utilise trypsin instead of acid to permit tumour formation, allowing for a more clinical valid method for tumour formation. However, we expected from the start that this would severely limit tumour formation in all mice. As only 4 out of the 8 mice demonstrated lung metastasis; we agree that intraluminal inoculation may not be the best lung metastasis model.

Reviewer #3:*The following comment has not been addressed:*

“The addition of the statement that bladder cancer usually originates from the urothelium and the authors are therefore injecting into the bladder wall is contradictory (lines 712-714). Tumor cells are usually injected below the mucosa, so that they are already invasive from the start.”

We thank the reviewer for their comment. We agree. The sentence has been edited as follows.

‘As most bladder cancers eventually invade the surrounding smooth muscle, we sought to generate an orthotopic bladder cancer model by injecting bladder cancer cells intramurally to gain insight into the potential anti-tumour invasive activity of MEK and TβR inhibition *in vivo*.’

Reviewer #3:*The BLI images in Fig 14 B and D would indicate quite modest drug efficacy – yet massive difference in the growth curves in Fig 14 C and E.*

In both of the mouse models tested we observe a decrease of 1:3 or 1:4 consistent with the BLI images presented.

Reviewer #3:*Fig S8 remove “epidermoid” from figure label.*

Epidermoid has been replaced with the correct nomenclature Urothelial carcinoma with squamous differentiation

Reviewers' comments:

Reviewer #2 (Remarks to the Author):

The additional data and quantifications provided indicate indeed an increase of TBR1 expression level upon HGF stimulation. The increase is however very modest (1,3 folds). The quantification of panel 6B should be provided with statistics. Equally for SupFigure7.

My concern regarding the MET dependency of the in vivo model remains. Figure 8 has only one panel related to HGF/MET (in vitro). The authors answer that "the focus of this study is to unravel the contribution of the TGF β receptor in invasion". However, the title of the paper is "c-Met activation leads to the establishment of a TGF β regulatory network required for bladder cancer invasion". Accordingly, as I already commented in the last round of review, the authors can not write this: "Based on these results we used an orthotopic mouse model of bladder cancer whereby we could analyze the role of T β R-mediated HGF-induced invasion". The authors vaguely respond that "c-MET is likely to be activated through both autocrine and paracrine effects in our mouse model". Again, in the mice, murine HGF is unlikely to be able to activate human MET in UMUC3 cells, thus no paracrine activation. And there is no proof in this manuscript that the observed constitutive activation of MET in UMUC3 cells results from an autocrine loop. However, MET is activated without HGF treatment in these cells (Figure S9A). Furthermore, the soft agar experiment Figure S10E,F indicate that the anchorage independent growth of these cells is significantly reduced upon MET inhibitor treatment. These data indicate (although do not prove) the possibility that UMUC3 cells' tumorigenesis in vivo could be under control of MET activity, at least in part. The authors should modify the text introducing their in vivo experiment accordingly (high potential to be MET dependent, but not HGF dependent).

To keep the authors' conclusion that "These set of experiments confirm that the combination of both drugs is required to effectively inhibit HGF-induced invasion and proliferation", my request remains that the in vivo experiment should be performed with a MET inhibitor in parallel. If the authors are not willing to do this experiment, they should be more careful in their conclusions about the involvement of MET (and even more HGF) in their in vivo experiments. This includes the abstract "In vivo we show that TGF β receptor inhibition prevents HGF-induced bladder cancer invasion".

Reviewer #3 (Remarks to the Author):

-

Reviewer #5 (Remarks to the Author):

I found the overall study very interesting at the molecular level, elegantly and thoroughly supported by Figures 1-6. Second, there are clinical relevance (Figures 7-8), rendering the overall manuscript high impact and therefore justify publication at Nature Communications.

Regarding specific comments from Reviewer 3. I understand his/her technical concern regarding Supp Fig 14. On the other hand, since the experiments were performed as n=5 and n=3 respectively, I would also understand that it is possible the authors did not pick the representative images carefully? I would recommend the following, is it reasonable to either:

1. Request the authors to provide all the IVIS images for all the n numbers to satisfy reviewer 3's request?
2. Alternatively, the authors can provide a more representative image for the final figures? The BLI values are generated by the instrument, it is also possible that the scale for control and treatment group are different (and simply a technical oversight of the authors, which is reasonable but not currently presented in the Figure as different scales).

Finally, I found Figures 8i and 8s particularly well done and informative, compared to other studies in the field, which is another highlight of the current study.

c-Met activation leads to the establishment of a TGF β receptor regulatory network required for bladder cancer invasion.

Responses to reviewer's comments:

We thank the reviewers for the thoughtful and thorough review of the manuscript. We have revised the manuscript according to the reviewers' suggestions and have addressed all the concerns brought forward by the reviewers.

Reviewer#2: *The additional data and quantifications provided indicate indeed an increase of TBR1 expression level upon HGF stimulation. The increase is however very modest (1,3 folds). The quantification of panel 6B should be provided with statistics. Equally for SupFigure7.*

We thank the reviewer for his/her comment. We now include statistics for Figure 6B comparable to Sup. Figure 7 G,H. (New figure Sup Figure 7F).

Reviewer#2: *My concern regarding the MET dependency of the in vivo model remains. Figure 8 has only one panel related to HGF/MET (in vitro). The authors answer that "the focus of this study is to unravel the contribution of the TGF β receptor in invasion". However, the title of the paper is "c-Met activation leads to the establishment of a TGF β regulatory network required for bladder cancer invasion". Accordingly, as I already commented in the last round of review, the authors can not write this: "Based on these results we used an orthotopic mouse model of bladder cancer whereby we could analyze the role of T β R-mediated HGF-induced invasion". The authors vaguely respond that "c-MET is likely to be activated through both autocrine and paracrine effects in our mouse model". Again, in the mice, murine HGF is unlikely to be able to activate human MET in UMUC3 cells, thus no paracrine activation. And there is no proof in this manuscript that the observed constitutive activation of MET in UMUC3 cells results from an autocrine loop. However, MET is activated without HGF treatment in these cells (Figure S9A). Furthermore, the soft agar experiment Figure S10E,F indicate that the anchorage independent growth of these cells is significantly reduced upon MET inhibitor treatment. These data indicate (although do not prove) the possibility that UMUC3 cells' tumorigenesis in vivo could be under control of MET activity, at least in part. The authors should modify the text introducing their in vivo experiment accordingly (high potential to be MET dependent, but not HGF dependent). To keep the authors' conclusion that "These set of experiments confirm that the combination of both drugs is required to effectively inhibit HGF-induced invasion and proliferation", my request remains that the in vivo experiment should be performed with a MET inhibitor in parallel. If the authors are not willing to do this experiment, they should be more careful in their conclusions about the involvement of MET (and even more HGF) in their in vivo experiments. This includes the abstract "In vivo we show that TGF β receptor inhibition prevents HGF-induced bladder cancer invasion".*

We thank the reviewer for his/her comment. Note, as part of the revision process it was indicated to us by the editor that no new mouse experiments were required. Although we agree with the reviewer that this would be an interesting experiment to perform we believe our overall results conclusively demonstrate, in vitro, the role of the TGF β receptor pathway

in HGF mediated EMT. Furthermore, we have now performed a new experiment further demonstrating the role of the TGF β receptor in HGF-mediated invasion (New figure Sup Fig 3 B,C).

In regards to the reviewer's concerns about the conclusions involving MET and HGF in our *in vivo* experiments, we agree. In line with the reviewer's suggestions we have now modified the text for the *in vivo* experiment, the manuscript title, and altered the abstract accordingly.

Title:

c-Met activation leads to the establishment of a TGF β regulatory network required for bladder cancer invasion.

c-Met activation leads to the establishment of a TGF β -receptor regulatory network in bladder cancer progression

Abstract:

‘This upregulation of the TGF β pathway by HGF leads to TGF β -mediated Epithelial-Mesenchymal Transition (EMT) and invasion. *In vivo* we show that TGF β inhibition completely prevents HGF-induced bladder cancer invasion.’

‘This upregulation of the TGF β receptor pathway by HGF leads to TGF β -mediated EMT and invasion. *In vivo* we show that TGF β receptor inhibition prevents bladder cancer invasion.’

‘However, the precise mechanism through which HGF induces EMT remains undetermined. Here we elucidate the mechanism through which HGF activates downstream T β R signalling enhancing EMT and invasion.’

‘However, the precise mechanism through which c-MET activation induces EMT remains undetermined. Here we elucidate the mechanism through which activated c-Met promotes downstream T β R signalling enhancing EMT and invasion *in vitro*, and tumour progression in bladder carcinoma orthotopic transplantation model.’

‘Based on these results we used an orthotopic mouse model of bladder cancer whereby we could analyze the role of T β R-mediated HGF-induced invasion.’

‘Based on these results we used an orthotopic mouse model of bladder cancer whereby we could analyze the role of c-MET-mediated invasion. We choose to utilize UMUC3 cells in this setting as they expressed the highest levels of activated c-MET.’

‘First, SMURF2 inhibition by c-SRC upregulates the TGF β pathway by stabilising TGF β R complex a function we define to be critical for HGF-induced EMT and invasion in bladder cancer.’

‘First, SMURF2 inhibition by c-SRC upregulates the TGF β receptor pathway by stabilising the TGF β receptor complex, a function we define to be critical for HGF-induced EMT and invasion *in vitro* and tumour progression *in vivo* in bladder cancer.’

Reviewer #3:- We thank the reviewer for his/her effort in reviewing our article.

Reviewer #5 : *I found the overall study very interesting at the molecular level, elegantly and thoroughly supported by Figures 1-6. Second, there are clinical relevance (Figures 7-8), rendering the overall manuscript high impact and therefore justify publication at Nature Communications.*

We thank the reviewer for his/her comments and the opinion that our manuscript is suitable for publication in Nature Communications.

Reviewer #5 *Regarding specific comments from Reviewer 3. I understand his/her technical concern regarding Supp Fig 14. On the other hand, since the experiments were performed as $n=5$ and $n=3$ respectively, I would also understand that it is possible the authors did not pick the representative images carefully? I would recommend the following, is it reasonable to either:*

- 1. Request the authors to provide all the IVIS images for all the n numbers to satisfy reviewer 3's request?*
- 2. Alternatively, the authors can provide a more representative image for the final figures? The BLI values are generated by the instrument, it is also possible that the scale for control and treatment group are different (and simply a technical oversight of the authors, which is reasonable but not currently presented in the Figure as different scales).*

We thank the reviewer for his/her excellent comment and now recognise the previous concerns raised by reviewer number 3. After a thorough review of our mouse data in Sup Fig 14 we identified a number of errors in the presentation of our data. We sincerely apologise for these inconsistencies. The quantification of this data is calculated by the IVIS spectrum imaging system and is represented as total flux (photon/seconds). This calculation was used to generate the representative graphs in figures Sup Fig 14C and 14E. These calculations are also shown in the source data. This data is correct and has now been re-validated. However, as correctly pointed out by this reviewer the scale bars for both Sup Fig 14B and Sup Fig 14D incorrectly represent the data. We sincerely thank the reviewer for picking up this important error on our part.

Pertaining to the reviewer's suggestions we now demonstrate all of the IVIS images for each mouse in both experiments. This data has now been included in the source data. Furthermore, we have now generated new figures for Sup Fig 14B and Sup fig 14D demonstrating the specific scale bar for each image. As correctly predicted by the reviewer the scales bars for controls and treatment groups are indeed different.

Reviewer #5: *Finally, I found Figures 8i and 8s particularly well done and informative, compared to other studies in the field, which is another highlight of the current study.*

We thank the reviewer for his/her comments concerning the quality of our work.

REVIEWERS' COMMENTS:

Reviewer #2 (Remarks to the Author):

The authors are responded to all my comments

Reviewer #5 (Remarks to the Author):

The authors has made the appropriate changes including corresponding scales for the IVIS images (at the final format, it would be recommended that they use the same scale for all images to demonstrate the temporal change of tumors over time and the relative difference between control and drug treatment).

Likewise, we would highly recommend them to provide the scales for Figure 8g as well, with the current presentation, Figure 8g does not correspond with 8i-h.

c-Met activation leads to the establishment of a TGF β -receptor regulatory network in bladder cancer progression

Responses to reviewers' comments:

We thank the reviewers for the thoughtful and thorough review of the manuscript. We have revised the manuscript according to the reviewers' suggestions.

Reviewer#2: *The authors are responded to all my comments.*

We thank the reviewer for his continued support during the revision of this manuscript.

Reviewer#5: *"..we would highly recommend them to provide the scales for Figure 8g as well, with the current presentation, Figure 8g does not correspond with 8i-h."*

We thank the reviewer for his comment. We have now included scale bars for this figure.

Reviewer#5: "...it would be recommended that they use the same scale for all images to demonstrate the temporal change of tumors over time and the relative difference between control and drug treatment."

We attempted to provide the same scale bar for all figures. Unfortunately, if we use the same scale bar we are unable to see significant tumour formation at the earlier time points. We ask the reviewer to kindly accept the figure as it is presented.